# DNA methylation and lncRNA control asynchronous DNA replication at specific imprinted gene domains

Yui Imaizumi [1,2,5], François Charon [3,5], Caroline Surcis[1,2], Christel Picard [1,2], Pol Arnau-Romero[1,2], Jean-Christophe Andrau [1,2], Daan Noordermeer [3], Benoit Moindrot [3] ✉, Jean-Charles Cadoret[4] ✉ & Robert Feil [1,2] ✉

Besides genome-wide patterns of replication timing (RT), some genes display allelic replication asynchrony in stem cells, brought about by stochastic events and genetic polymorphisms. Whether epigenetic modifications control asynchronous replication remains unclear. Here, we explore domains controlled by genomic imprinting, where parental DNA methylation imprints mediate allele-specific gene expression. Our genome-wide and locus-specific assays in monoparental and hybrid mouse ESCs reveal pronounced RT asynchrony—which is parent-of-origin dependent and lost upon neural differentiation—at the *Dlk1-Dio3* and *Snrpn* domains, which both comprise lncRNA polycistrons. Generating a range of mutant lines, we find that asynchronous replication at *Dlk1-Dio3* is mediated by differential DNA methylation, and that the lncRNA Meg3 controls early replication across parts of the domain on the maternal chromosome. Moreover, we find no evidence that RT and organisation into TADs are linked in this domain. The combined replication timing, DNA methylation, 3D chromatin structure, and gene expression data highlight how parental methylation imprints and lncRNA expression control replication and can override RT domain organisation.

Eukaryotic genomes are organized in 400–800-kb regions that replicate early or late in S-phase named CTRs (Constant Timing Regions), separated by transition zones called TTRs (Transitional Timing Regions)[1,2]. Replication timing (RT) domains have been associated with gene expression states and GC content[3], and their boundaries coincide with the boundaries of Topologically Associating Domains (TADs)[4,5]. Gene-rich and transcriptionally-active regions tend to replicate early and are within the structurally-defined A-compartment of the genome. Regions with low transcriptional activity are mostly in the B-compartment and replicate mostly late in S-phase[6–10]. Although RT is maintained along a substantial part of the genome, specific regions alter their RT during stem cell differentiation[11,12]. These RT switches correlate with changes in gene activity, sub-nuclear localization, and chromatin organisation[3,9,13–16]. Despite these emerging links, the molecular regulation of differential RT remains poorly understood.

One powerful technology for RT consists of determining the copy number of DNA along the genome in individual cells[9,10,17]. Another, more broadly used approach quantifies the incorporation of chemically-tagged nucleotides into newly-synthesized DNA, in early-versus late S-phase fractions[18], or compares the DNA content between S-phase and G1-phase cells[19]. With such approaches, asynchronous replication has been detected at a few percent of genes in clonally-derived mouse embryonic stem cells (mESCs)[15,20]. This kind of random asynchrony, in which RT is different between alleles and the early-

[1]CNRS, Institute of Molecular Genetics of Montpellier (IGMM), Montpellier, France. [2]Université de Montpellier, Montpellier, France. [3]Université Paris-Saclay, CEA, CNRS, Institute for Integrative Biology of the Cell (I2BC), Gif-sur-Yvette, France. [4]Université Paris Cité, Institut Jacques Monod, CNRS, Paris, France. [5]These authors contributed equally: Yui Imaizumi, François Charon. ✉e-mail: benoit.moindrot@i2bc.paris-saclay.fr; jean-charles.cadoret@ijm.fr; robert.feil@igmm.cnrs.fr

replicating allele is different between cells, is often associated with mono-allelic transcription states[16,20,21]. Diverse studies on haematopoietic stem cells have shown stochastic replication asynchrony at specific gene loci as well, including at immune system genes[22,23]. During X-chromosome inactivation, similarly, the inactive X chromosome becomes reorganised to replicate in late-S[24].

Imprinted gene domains provide an attractive RT paradigm. They are controlled by parental DNA methylation imprints at differentially methylated regions (DMRs), which are up to several kilobases in size, but there is no ubiquitous canonical mechanism through which these imprints mediate in-cis the allelic expression of genes. At imprinted domains, the parental chromosomes display opposite patterns of DMR methylation, gene expression, TAD structure, and binding of the CTCF architectural protein[25], features that can all potentially influence RT[23].

DNA fluorescence in situ hybridisation (FISH) studies on mESCs have shown a 'doublet plus singlet' signal in a relatively high percentage of interphase nuclei at the imprinted *Igf2-H19, Snrpn, Igf2r,* and *Dlk1-Dio3* domains, suggestive of asynchronous replication[26–29]. However, FISH monitors chromatid separation and does not always reflect the timing of DNA replication[30,31]. Furthermore, recent genome-wide studies on hybrid mESCs did not report asynchronous replication at imprinted domains[20], but here the emphasis was not on imprinted genes, and maintenance of methylation imprints was not monitored. It remains therefore uncertain to what extent imprinted domains might replicate asynchronously between the parental chromosomes. To strictly prevent aberrant alterations in DNA methylation at imprinted loci, here we explored mono-parental and hybrid mESCs grown under serum-free conditions in the presence of ascorbic acid[32]. In agreement with genome-wide studies[20], we find that the parental chromosomes replicate synchronously along most of the genome, including at most imprinted domains. Pronounced replication asynchrony was detected at specific imprinted loci; however, most strongly along the *Dlk1-Dio3* domain on chromosome 12. Using CRISPR-based approaches to test candidate mechanisms, for the first time, our study reveals an essential role for differential DNA methylation and for lncRNA expression in asynchronous replication. These data evoke a model in which differential DNA methylation in conjunction with developmental factors confers replication asynchrony in pluripotent cells by overriding inherent mechanisms that dominate elsewhere in the genome.

## Results

### A genome-wide survey in mESCs reveals replication asynchrony at imprinted domains

To assess allele-specific RT, we compared mESCs with two maternal genomes (parthenogenetic line PR3) with mESCs with two paternal genomes (androgenetic line AK2). Biparental male mESCs that were hybrid between *M. m. domesticus* strain C57BL/6J and *M. m. molossinus* strain JF1 were studied as well. Specifically, we studied mESC line 'BJ', in which the maternal genome is C57BL/6J and the paternal genome JF1, and line 'JB', which has the reciprocal genotype.

Our assay consisted of a 1 hour exposure of asynchronous cells to BrdU, followed by cell-stage-dependent FACS, precipitation of BrdU-enriched DNA, and micro-array hybridisation[11,33] (Fig. 1a and Supplementary Fig. 1a). An 'early fraction' comprised of late-G1 and the first ~40% of S-phase, was compared with a 'late fraction' covering the last ~40% of S-phase and early G2 (Supplementary Fig. 1b). Previously, we reported that this approach gives similar results as multi-fraction Rep-seq[34]. Based on triplicate experiments, ratios between early- and late hybridisation signals [log-2 (Early/Late)] were plotted along the genome. Control developmental genes, including *Dppa2* [active and early-replicating[35]] and *Ptn* [repressed and late-replicating[36]], confirmed correct cell-cycle-stage fractionation (Fig. 1b and Supplementary Fig. 1c). Our assay has a resolution of ~13-kb, and yielded the expected genome-wide pattern of early and late replicating regions, as shown for chromosome 12 (Fig. 1c). The androgenetic and parthenogenetic

mESCs had a comparable genome-wide RT pattern (Fig. 1c), and showed a similar high overlap (~90% identity) with the biparental mESCs (Supplementary Table 1). The genome-wide RT patterns were similar to those recently reported in the inner cell mass (ICM) of the blastocyst[10,37], from which they derived (Supplementary Fig. 1d).

For each mESC line, we determined RT using START-R software[34], and focused on the imprinted gene domains controlled by parental DNA methylation imprints (Fig. 1d). We centred our analysis on the germline DMRs [*i.e.*, the imprinting control regions, ICRs] and close-by neighbouring sequences, using all data points from multiple sequential probes in triplicate experiments (Fig. 1d). In agreement with their gene-richness and, for most, their transcriptional activity in mESCs[38], a majority of imprinted domains showed early RT, with no apparent asynchrony between androgenetic and parthenogenetic cells (Fig. 1d). The strongest asynchrony was measured at the *Dlk1-Dio3* domain on distal chromosome 12 (Fig. 1d). This was apparent also from the RT profiles, with early replication in parthenogenetic and late replication in androgenetic cells, and 'intermediate RT' in biparental mESCs (Fig. 1e). This correlated with maintenance of normal DNA methylation levels at the domain's DMRs (Fig. 1f).

Although for several other imprinted domains (*Igf2-H19, Peg3/ Zim, RasGrf1*) putative RT differences were apparent (Fig. 1d), these loci displayed similar RT profiles in androgenetic and parthenogenetic mESCs and this was observed at the imprinted *Igf2r, Plagl1 and Grb10* loci as well (Supplementary Fig. 1e).

At the imprinted *Snrpn* locus, a tendency towards late RT was apparent in both mono-parental lines. However, this gene replicated significantly earlier in the androgenetic than in the parthenogenetic mESCs (Fig. 1d, g). This correlated with maintenance of a normal methylation level at the domain's gDMR (Fig. 1f), which suggested that differential methylation could potentially be involved.

### A large zone of asynchronous replication comprises the *Dlk1* and *Meg3-Rian-Mirg* imprinted genes

The asynchronous RT at the *Dlk1-Dio3* domain covered approximately 800 kb (Fig. 1e). It includes the paternally-expressed *Dlk1*–a developmental regulator of signalling–and the maternally-expressed *Meg3-Rian-Mirg* polycistron that produces a multitude of ncRNAs, including the lncRNA Meg3, C/D-box snoRNAs (*Rian*) and miRNAs (*Mirg*)[39] (Fig. 2a). The two paternally methylated DMRs showed the expected methylation levels. The intergenic IG-DMR–the gDMR/ICR of the domain[40]–and the *Meg3*-DMR were both highly methylated in androgenetic and barely at all in parthenogenetic cells, and had an expected intermediate level (40–60%) in the biparental mESCs (Fig. 1f, Supplementary Fig. 2c).

Next, we directly compared the parental alleles within the hybrid mESCs. Initially, we performed PCR across single nucleotide polymorphisms (SNPs) within *Meg3* (Fig. 2b). Both allele-specific PCR (top panels) and Sanger sequencing of PCR products (lower panels) showed early replication on the maternal and mostly late replication on the paternal chromosome, both in BJ and JB mESCs (Fig. 2c, d). Since our array-based studies suggested replication asynchrony at *Dlk1* as well (Fig. 1e), we also performed amplification across an SNP at this developmental gene (Fig. 2b). In both JB and BJ cells, Sanger sequencing of PCR products showed early replication on the maternal and late replication on the paternal chromosome (Fig. 2c, d). We conclude that at *Meg3* and *Dlk1*, the asynchrony reflects the parental origin, rather than genetic strain differences between the parental chromosomes.

To pinpoint the precise sequences that undergo asynchronous replication we developed 'Capture Repli-seq', a high-resolution replication-sequencing approach in which regions of interest were captured prior to DNA sequencing. SNPs between C57BL/6 J and JF1 [~4 per kb[41]] allowed us to distinguish the maternal- and paternal-chromosome signal at 1-kb resolution. Similar to the recent genome-

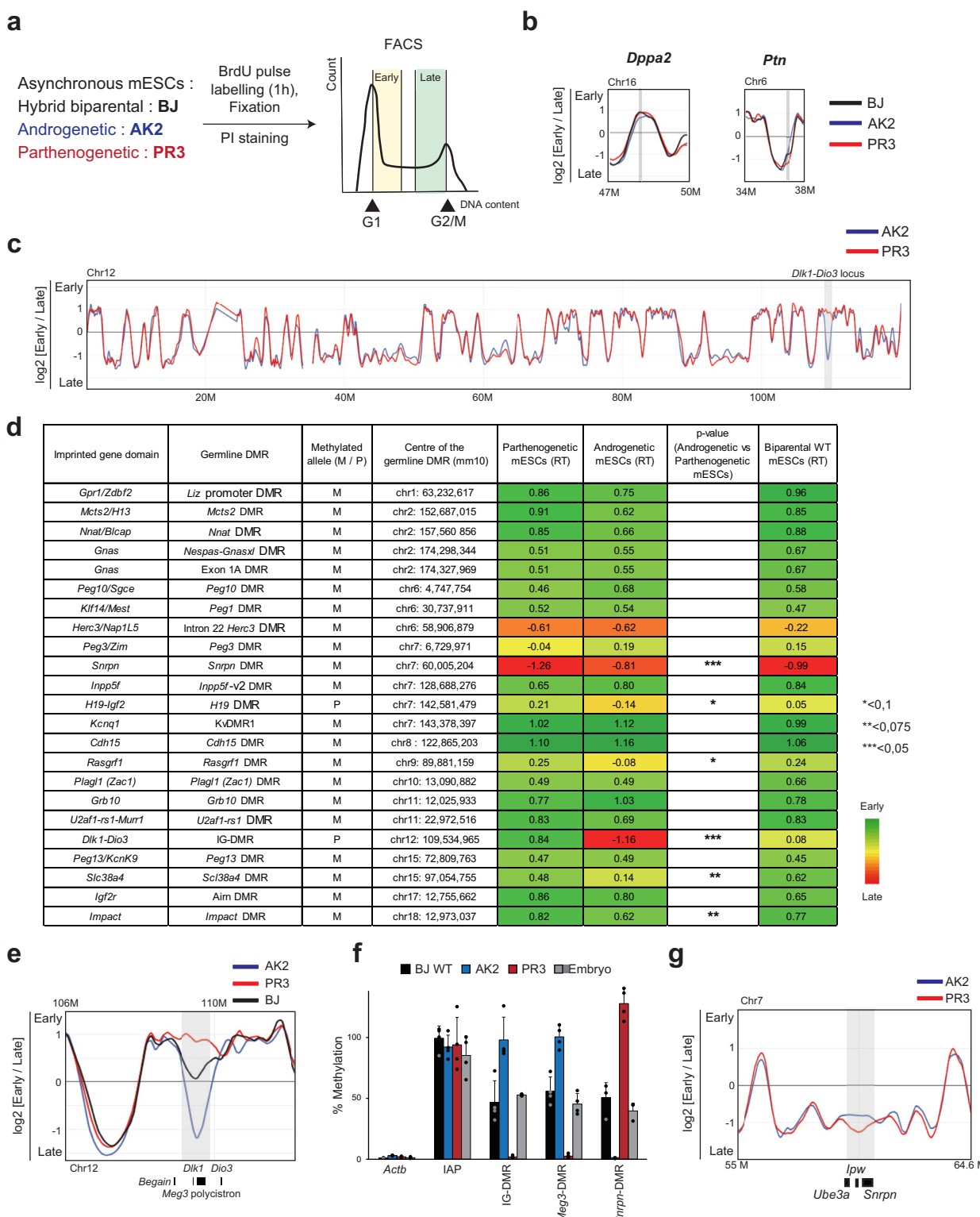

wide Repli-Capture-Seq assay[42], our approach relies on pools of custom oligonucleotides, but our design tiled specific regions, thereby allowing their enrichment at high resolution.

We first analysed a 0.6-Mb control region on chromosome 4, which showed a comparable RT pattern on the maternal and paternal chromosome (Supplementary Fig. 2a). At the captured *Dlk1-Dio3* locus, there was pronounced asynchrony with >60 fold allelic-sequence enrichment across ~750 kb. This large, continuous region of asynchrony includes the *Dlk1* gene and the entire 220-kb ncRNA

polycistron (Fig. 2a,e,f). RT profiles were comparable between the alleles with the same parental origin in BJ versus JB, which confirms that the RT asynchrony is parent-of-origin dependent. However, we noted a difference between the reciprocal lines downstream of the ncRNA polycistron. On the paternal chromosome, a late-replication zone includes this distal region in JB cells (paternal chromosome: B6), but not in BJ cells (paternal chromosome: JF1) (Fig. 2e, f). This difference is likely linked to genetic differences between the JF1 and C57BL/6 J mouse strains.

**Fig. 1 | Genome-wide RT assays on mono-parental mESCs pinpoint asynchronous replication. a** Schematic of the array-based RT analysis. Following BrdU incorporation in androgenetic (AK2), parthenogenetic (PR3), and biparental hybrid (BJ) mESCs, BrdU-precipitated DNA from early and late cell-cycle fractions is used for array hybridisation, followed by data analysis. **b** RT (log2 of early/late hybridization signals) at *Dppa2* (early replication in mESCs) and *Ptn* (late replication in mESCs) in BJ, AK2, and PR3 mESCs. Vertical bars indicate positions of the indicated genes. **c** RT along chromosome 12 in AK2 (blue) versus PR3 (red) mESCs. The grey rectangle indicates a distal region with pronounced asynchrony. **d** Heatmap of RT at imprinted domains in PR3, AK2, and BJ mESCs. For the domains' gDMRs (ICRs), M indicates methylation on the maternal allele, and P indicates methylation on the paternal allele. The colour range indicates RT−red being late and green being early−determined based on the earliest and latest values obtained from the smooth curves generated by START-R. Locus-specific values are indicated, determined using the Mean method, with a sliding window length of 60 and with an overlap size

of 30, as described[34]. Significance levels (AK2 versus PR3) are indicated. **e**, RT profile of PR3 (red line), AK2 (blue line) and BJ (in black line) mESCs at the *Dlk1-Dio3* locus (distal chr.12). The grey box indicates the region with asynchronous replication, determined by START-R software with the Mean method, with a p-value threshold = 0.05, a sliding windows length = 60 and an overlap size = 30, as described previously[34]. **f** Methylation levels in the *Dlk1-Dio3* and *Snrpn* domains, determined by methylation-sensitive qPCR at the domains' ICRs (IG-DMR at *Dlk1-Dio3*, *Snrpn*-DMR at *Snrpn* domain) and a secondary DMR (*Meg3*-DMR) in PR3, AK2, and BJ mESCs. Embryo (E13) DNA is included for comparison; negative (*B-Actin* promoter, no methylation) and positive control regions (IAPs, highly methylated) are included. Bars represent means ± SD from 4 independent experiments. **g** The *Snrpn* domain on central chromosome 7 shows a less pronounced asynchrony, with earlier replication in AK2 mESCs across a region comprising *Snrpn*. The light grey box indicates the region with putative asynchronous replication.

To determine more precisely when replication occurs, we sorted cells into four fractions: 'late-G1', 'early-S', 'late-S' and 'early-G2' (Supplementary Fig. 2d). This showed that *Dppa2* (active) and *Ptn* (inactive) replicated very early and in late-S, respectively (Supplementary Fig. 2e). At *Dlk1-Dio3*, replication of the maternal *Meg3* and *Dlk1* alleles was detected at late-G1 and the onset of S-phase (fraction A, Supplementary Fig. 2d); the paternal alleles showed replication in late-S predominantly (Supplementary Figs. 2f–h). These findings indicate that a ~750-kb domain comprising *Dlk1*, *Meg3-Rian-Mirg* and *Rtl1* replicates very early on the maternal chromosome, and mostly in late-S on the paternal chromosome.

### Allelic analysis of replication timing at the *Igf2-H19*, *Kcnq1*, and *Snrpn* domains

Using Capture Repli-seq, we analysed the imprinted *Igf2-H19* domain−at which an early study on serum-grown mESCs had reported differential RT[27] −and the neighbouring *Kcnq1* domain as well. Capture Repli-seq on this distal chromosome-7 region showed that the parental chromosomes had a similar RT (Supplementary Fig. 2b). At the *Kcnq1* domain, *Kcnq1* and the flanking *Cdkn1c* gene showed early replication on both the parental chromosomes, while the *Igf2-H19* domain showed an intermediate RT on both the parental chromosomes. This lack of asynchrony correlated with unaltered methylation levels (40-60%) at the *Igf2-H19* ICR and at KvDMR1, the ICR of the *Kcnq1* domain (Supplementary Fig. 2c).

Capture Repli-seq could not be applied with consistent coverage across the *Snrpn* domain because of its exceptionally high content in repeat elements (mm10, Chr7:59,400,000−59,950,000). Instead, we performed PCR across SNPs. This showed that at *Snrpn* (regions 1 and 2), the early fraction showed replication of the paternal chromosome predominantly, whereas the maternal chromosome was enriched in the late fraction. A third, further-downstream region, at *Ipw*, did not show allelic differences in RT (Supplementary Fig. 1f). Upon sorting into 4 fractions, similarly, the paternal *Snrpn* allele was detected in late-G1 and early-S mostly, whereas the maternal allele was enriched in early-G2 (Supplementary Fig. 2i, j). These data show that at *Snrpn*, the paternal chromosome replicates on average earlier than the maternal chromosome, similar as reported in human ESCs[43,44].

In the hybrid mESCs, we also analysed the imprinted genes *H19*, *Peg3*, *Rasgrf1*, and *Slc38a4*, which had shown putative differences between the androgenetic and parthenogenetic mESCs (Fig. 1d, Supplementary Fig. 1e). This did not show allelic enrichments in the early and/or late fractions, and therefore no evidence for replication synchrony (Supplementary Fig. 1g).

### Differential DNA methylation is essential for asynchronous replication at the *Dlk1-Dio3* and *Snrpn* domains

The *Dlk1-Dio3* locus encompasses two essential DMRs: the germline DMR called the IG-DMR[40], and the somatically-acquired *Meg3*-DMR,

both unmethylated on the maternal allele (Fig. 2a). On the maternal chromosome, the IG-DMR acts as an enhancer that drives the expression of the close-by *Meg3-Rian-Mirg* ncRNA polycistron[26,45]. To determine whether DNA methylation is important for the allelic RT across the domain, we analysed BJ-derived mESCs in which methylation was either biallelic (line Sh-1) or absent from both parental alleles (line *Zfp57*[-/-]) at the IG-DMR and *Meg3*-DMR (Fig. 3a). The Sh-1 mESCs were derived previously by directing shRNAs against enhancer RNAs expressed from the maternal IG-DMR, which induced biallelic DNA methylation at both the IG-DMR and *Meg3*-DMR[26]. mESCs deficient in ZFP57 (*Zfp57*[-/-] cells)−a KRAB-domain zinc finger protein essential for imprint maintenance[46]−were generated recently[47]. They showed some residual IG-DMR methylation and hardly any methylation at the *Meg3*-DMR (Fig. 3b). Concordantly, Sh-1 mESCs did not express *Meg3, Rian*, and *Mirg*, while the *Zfp57*[-/-] cells showed increased *Meg3, Rian* and *Mirg* expression (Supplementary Fig. 3a). In the Sh-1 mESCs, *Dppa2* and *Ptn* showed the expected early and late replication, respectively (Supplementary Fig. 3c). Both parental *Meg3* alleles showed late replication (Fig. 3c and Supplementary Fig. 3c), indicating an early-to-late switch on the maternal chromosome. In the *Zfp57*[-/-] mESCs, conversely, *Meg3* replicated early on both the parental chromosomes (Fig. 3d and Supplementary Fig. 3d), indicating a late-to-early switch on the paternal chromosome.

To determine where precisely RT depends on the imprinted DNA methylation, we performed Capture Repli-seq. In the Sh-1 and *Zfp57*[-/-] mESCs, RT was unaffected at the chromosome-4 control region (Supplementary Fig. 3e). In contrast, compared to WT mESCs (Fig. 3e), at *Dlk1-Dio3* the entire asynchrony zone had shifted to late replication on both parental chromosomes in the Sh-1 cells. In the *Zfp57*[-/-] cells, conversely, the entire zone replicated early on both parental chromosomes (Fig. 3e). Combined, these findings indicate that allelic DNA methylation at the IG-DMR and/or the *Meg3*-DMR is a distinguishing determinant of asynchronous replication across the entire ~750-kb zone.

Also the ICR of the *Snrpn* locus (Supplementary Fig. 1f) requires the KRAB-domain zinc finger protein ZFP57[47]. In the *Zfp57*[-/-] mESCs, hardly any methylation was left at this gDMR (Supplementary Fig. 3b). Allelic PCR showed loss of replication asynchrony and mostly early RT (Supplementary Fig. 3f, g). Also at *Snrpn*, therefore, allelic DNA methylation seems to control asynchronous replication in mESCs.

### LncRNA expression controls maternal chromosome-specific early replication at the *Dlk1-Dio3* domain

The RT switch in the *Zfp57*[-/-] cells could be attributed to the loss of DMR-methylation itself, or to resulting changes in gene expression (Fig. 3a). To determine the relative contribution of each, we generated ZFP57-deficient cells that no longer expressed the *Meg3* lncRNA polycistron. A transient CRISPR-Cas9 approach was used to delete a

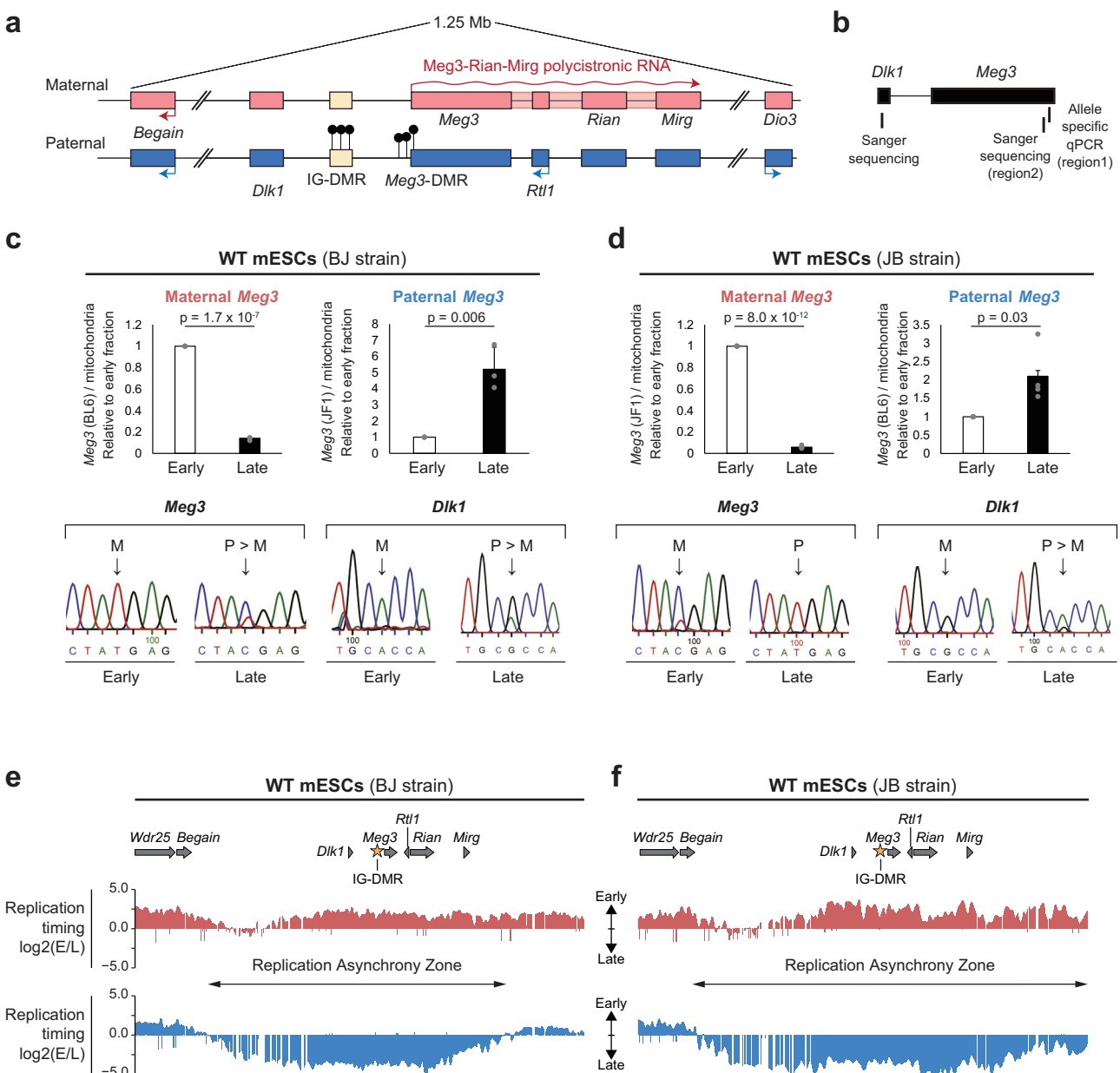

**Fig. 2 | Capture Repli-seq and PCR-based exploration of allelic RT at *Dlk1-Dio3* in hybrid mESCs. a** Schematic presentation of the *Dlk1-Dio3* domain on distal chromosome 12. Rectangles indicate genes on the maternal (red) and the paternal (blue) chromosome and their expression (arrows) in the embryo[78]. The allelic expression of the 220-kb *Meg3-Rian-Mirg* ncRNA polycistron is controlled by the IG-DMR (beige rectangle), which acts as an enhancer on the maternal, and is methylated (black lollipops) on the paternal chromosome. The *Meg3*-DMR is also methylated on the paternal chromosome. **b** Genomic locations chosen for allelic RT studies at *Dlk1* (Sanger sequencing) and *Meg3*-DMR (Sanger sequence analysis, allelic qPCR). RT at *Dlk1* and *Meg3* analysed by PCR in BJ (**c**) and JB (**d**) mESCs. Top panels, qPCR-based analysis of the maternal and paternal *Meg3* in the early and late samples. Signals were normalized against mitochondria, which replicate throughout the cell cycle[73]. For each experiment, values were further normalized to the early sample. Statistical significance between early and late samples was assessed using an unpaired two-tailed Student's t-test. Bars represent means ± SD from 3 biological replicates for BJ and 4 replicates for JB. Bottom panels, Sanger sequencing-based assessment of *Meg3* and *Dlk1* in early and late fractions. Arrows indicate SNPs that distinguish the parental chromosomes. At both genes, the maternal chromosome (M) replicates earlier than the paternal (P) chromosome. Capture Repli-seq at the *Dlk1-Dio3* locus across 1.1-Mb in BJ (**e**) and JB (**f**) mESCs. Maternal and paternal chromosomes are distinguished in the early and late fractions using sequencing reads that cover SNPs. Regions without reads correspond to repeat-rich sequences. Depicted is the log-2 early/late ratio across the domain (-1-kb resolution) on the maternal (in red) and the paternal (in blue) chromosome. Gene positions are aligned above. The asterisk indicates the IG-DMR position. Two-headed arrows indicate RT asynchrony between the maternal and the paternal chromosome. Genomic coordinates are for chromosome 12 (mm10).

198-bp region directly upstream of the *Meg3* TSS, to generate *Zfp57*[-/-];*Meg3*-pro[-/-] mESCs (Fig. 4a,b). Because of the lack of ZFP57, the IG-DMR remained lowly methylated, but the *Meg3*-DMR gained partial (~50%) de novo methylation on both the non-expressed alleles (Fig. 4c, d), similarly as seen in the pre-implantation embryo on the paternal *Meg3*[48]. As expected[49], the *Zfp57*[-/-];*Meg3*-pro[-/-] cells did not express *Meg3* or the other ncRNAs of the polycistron, as opposed to the *Zfp57*[-/-] mESCs from which they derived and that showed biallelic *Meg3* polycistron expression (Fig. 4e, f). As in the *Zfp57*[-/-] line (Fig. 3d), in the *Zfp57*[-/-];*Meg3*-pro[-/-] mESCs, the region carrying the (now

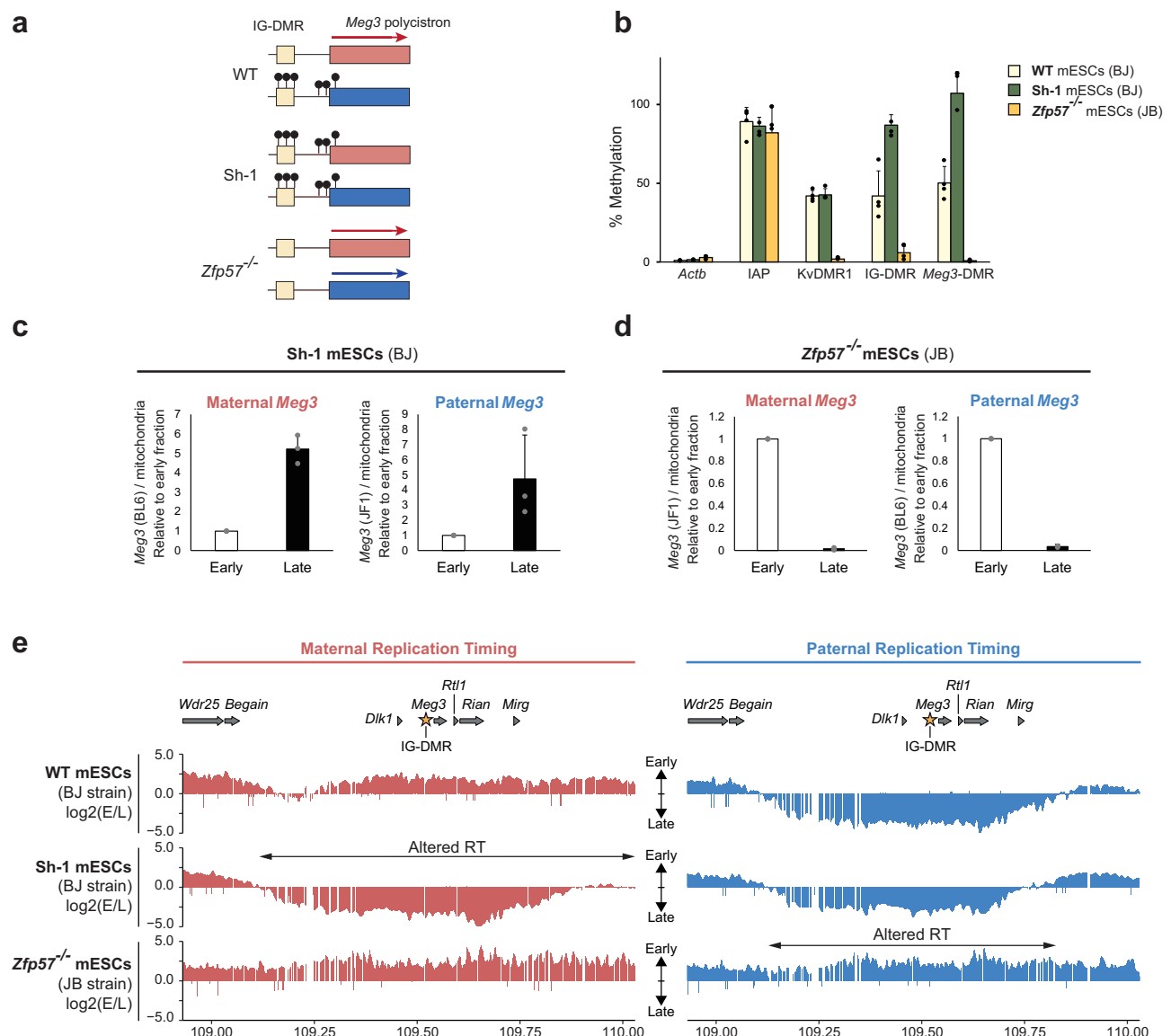

**Fig. 3 | Differential DNA methylation controls asynchronous replication at *Dlk1-Dio3*. a** Schematic presentation of IG-DMR and *Meg3*-DMR methylation (black lollipops) and *Meg3-Rian-Mirg* polycistron expression in WT, Sh-1 and *Zfp57*[-/-] mESCs. **b** Methylation-sensitive qPCR at the IG-DMR, *Meg3*-DMR, KvDMR1 (control ICR), *ActB* promoter (low-methylation control), and *IAP* elements (high-methylation control). Bars represent means ± SD from 4 experiments. Replication timing at *Meg3* on maternal and paternal chromosomes in Sh-1 (**c**) and *Zfp57*[-/-] (**d**) mESCs analysed by allele-specific qPCR. Bars represent means ± SD from 3 independent experiments. **e** Allelic Capture Repli-seq in WT (BJ), Sh-1 and *Zfp57*[-/-] mESCs at the *Dlk1-Dio3* domain. Depicted is the log-2 early/late ratio across the domain ( -1-kb resolution) on the maternal (in red) versus the paternal (in blue) chromosome. Gene positions are aligned above; the asterisk indicates the IG-DMR position. Two-headed arrows indicate altered RT in the Sh-1 or *Zfp57*[-/-] versus BJ mESCs.

inactive) *Meg3* and polycistron replicated early on both the parental alleles (Fig. 4g).

Using Capture Repli-seq, we assessed whether other parts of the replication zone could have become altered as a consequence of the loss of *Meg3* polycistron expression. This confirmed the observed early RT over the *Meg3* gene in *Zfp57*[-/-] and *Zfp57*[-/-];*Meg3*-pro[-/-] cells (Fig. 4h). However, at a proximal region located between *Dlk1* and *Begain*, spanning ~250 kb, the deletion of the *Meg3* promoter in the *Zfp57*[-/-] background caused aberrant late replication on both the parental chromosomes. RT in this region had adopted the RT pattern normally associated with the paternal chromosome. The *Meg3* polycistron expression therefore appears to control the early RT on the maternal chromosome at this '*Begain-Dlk1* region'. A less pronounced switch towards late replication was observed at an ~150-kb region comprising *Mirg* ('*Mirg* region') at the 3' side of the polycistron (Fig. 4h). At the

*Dock7* control locus, RT was unaltered in the *Zfp57*[-/-];*Meg3*-pro[-/-] cells (Supplementary Fig. 4). Combined, the above data evoke a model in which the paternal DMR controls differential RT in its surroundings on both the parental chromosomes, whereas the maternal expression of the large ncRNA polycistron in turn controls in cis early replication across proximal and distal parts of the asynchronous replication zone. We conclude that the asynchronous replication zone primarily depends on the action of ZFP57 ('maintenance of allelic DMR methylation') around *Meg3* and on the expression of the *Meg3* polycistron in the '*Begain-Dlk1* region' (strong effect) and the '*Mirg* region' (weaker effect).

To uncouple the role of Meg3 lncRNA from transcription over the polycistron, we generated hybrid mESCs that had a maternal 27.7-kb deletion from the 5'-end of intron-2 till directly downstream of *Meg3* exon-10 (Fig. 5a,b; Supplementary Fig. 5a−c). This line, called ΔMeg3-

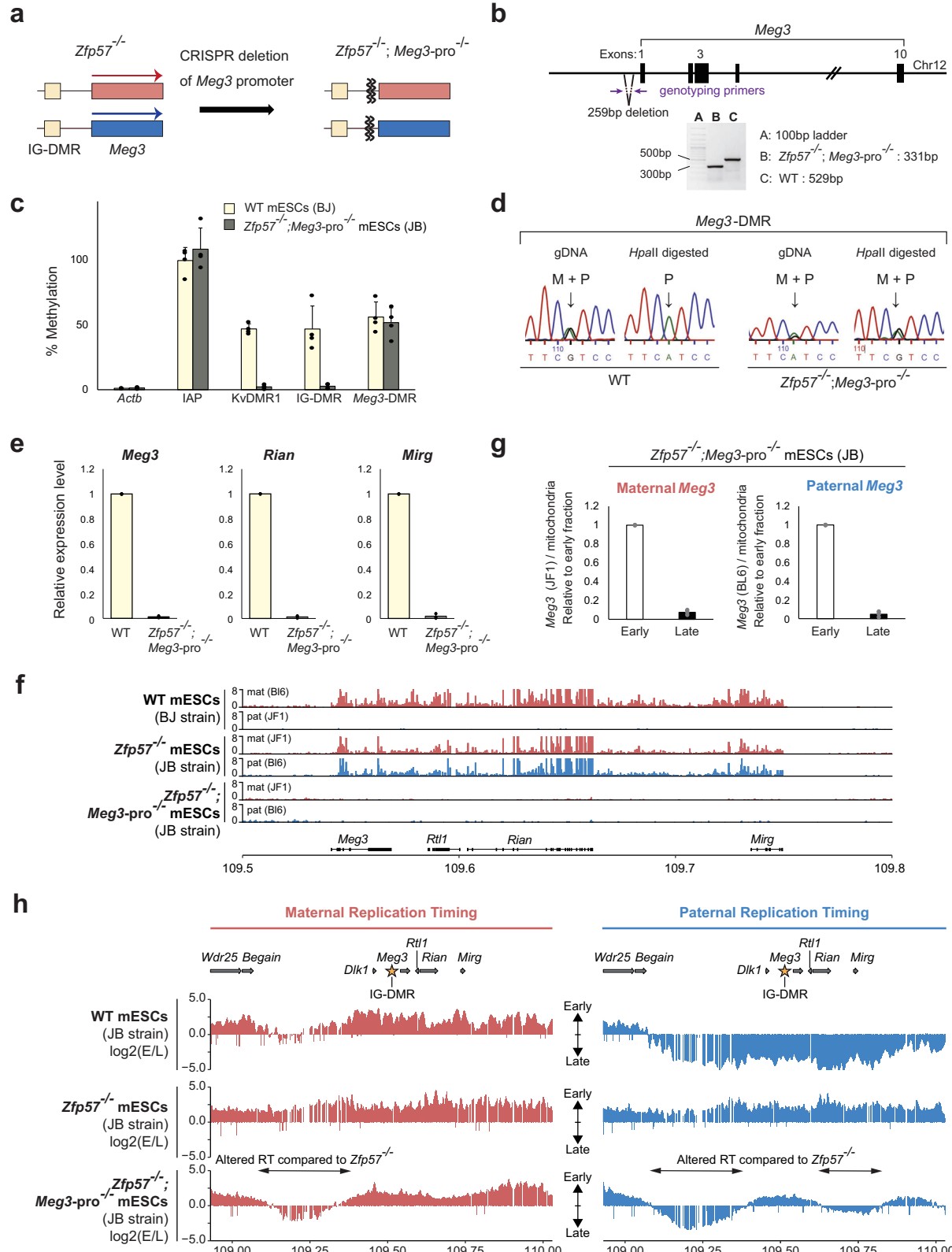

C1, had unaltered DNA methylation levels at the IG-DMR and *Meg3*-DMR (Fig. 5c). RT-qPCR analysis showed that besides the expected loss of *Meg3* expression, there was persistent *Rian* and *Mirg* expression (Supplementary Fig. 5d). This was confirmed by RNA-seq, which showed no reads across the deleted region, and a profile of expression elsewhere across the polycistron that was comparable to that in BJ WT mESCs (Fig. 5d). Using Capture Repli-seq, we assessed whether the

differential RT across the domain had become altered as a consequence of the loss of Meg3 lncRNA (Fig. 5e). A proximal region between *Dlk1* and *Begain* and a region distal to the polycistron showed aberrant late replication on the maternal chromosome. This finding implied that the early replication, normally observed across these regions on the maternal chromosome, requires the lncRNA Meg3 and is not a consequence of transcription across its gene body. In ΔMeg3-

**Fig. 4 | An lncRNA polycistron contributes to asynchronous DNA replication.** Generation of *Zfp57*[-/-]*;Meg3*-pro[-/-] mESCs by CRISPR-Cas9 technology (**a**), with a 198-bp deletion directly upstream of the TSS (**b**). Agarose gel electrophoresis of PCR products shows biallelic deletion. The same result was obtained in 2 independent experiments. **c** Methylation-sensitive qPCR analysis in *Zfp57*[-/-]*;Meg3*-pro[-/-] and WT mESCs at IG-DMR, *Meg3*-DMR, KvDMR1 (control ICR), *ActB* promoter (low methylation) and *IAP* elements (high methylation). Bars represent means ±S.D. from 4 independent experiments. **d** Sanger sequencing of *Meg3*-DMR PCR products obtained from genomic DNA (gDNA) or *Hpa*II-digested gDNA in WT and *Zfp57*[-/-]*; Meg3*-pro[-/-] mESCs. In the *Zfp57*[-/-]*; Meg3*-pro[-/-] cells, the partial methylation at the *Meg3*-DMR (panel d) is equally present on both the parental chromosomes. **e** Levels

of Meg3, Rian, and Mirg RNA relative to Gapdh determined by RT-qPCR in WT (BJ) and *Zfp57*[-/-]*; Meg3*-pro[-/-] mESCs, with WT values set at 1. Bars represent means ± SD from 4 independent experiments. **f** RNA-seq on BJ, *Zfp57*[-/-] and *Zfp57*[-/-]*;Meg3*-pro[-/-] mESCs. Reads on the maternal (in red) and the paternal chromosome (in blue) are depicted along the *Meg3-Rian-Mirg* polycistron. Gene positions are shown below. **g** Replication timing at *Meg3* analysed by allelic qPCR at *Meg3* in *Zfp57*[-/-]*;Meg3*-pro[-/-] mESCs. Bars represent means ± SD from 3 independent experiments. **h** Parental chromosome-specific RT determined by Capture Repli-seq in BJ, *Zfp57*[-/-] and *Zfp57*[-/-]*; Meg3*-pro[-/-] mESCs. Two-headed arrows indicate zones with altered RT in *Zfp57*[-/-]*; Meg3*-pro[-/-] mESCs relative to *Zfp57*[-/-] mESCs.

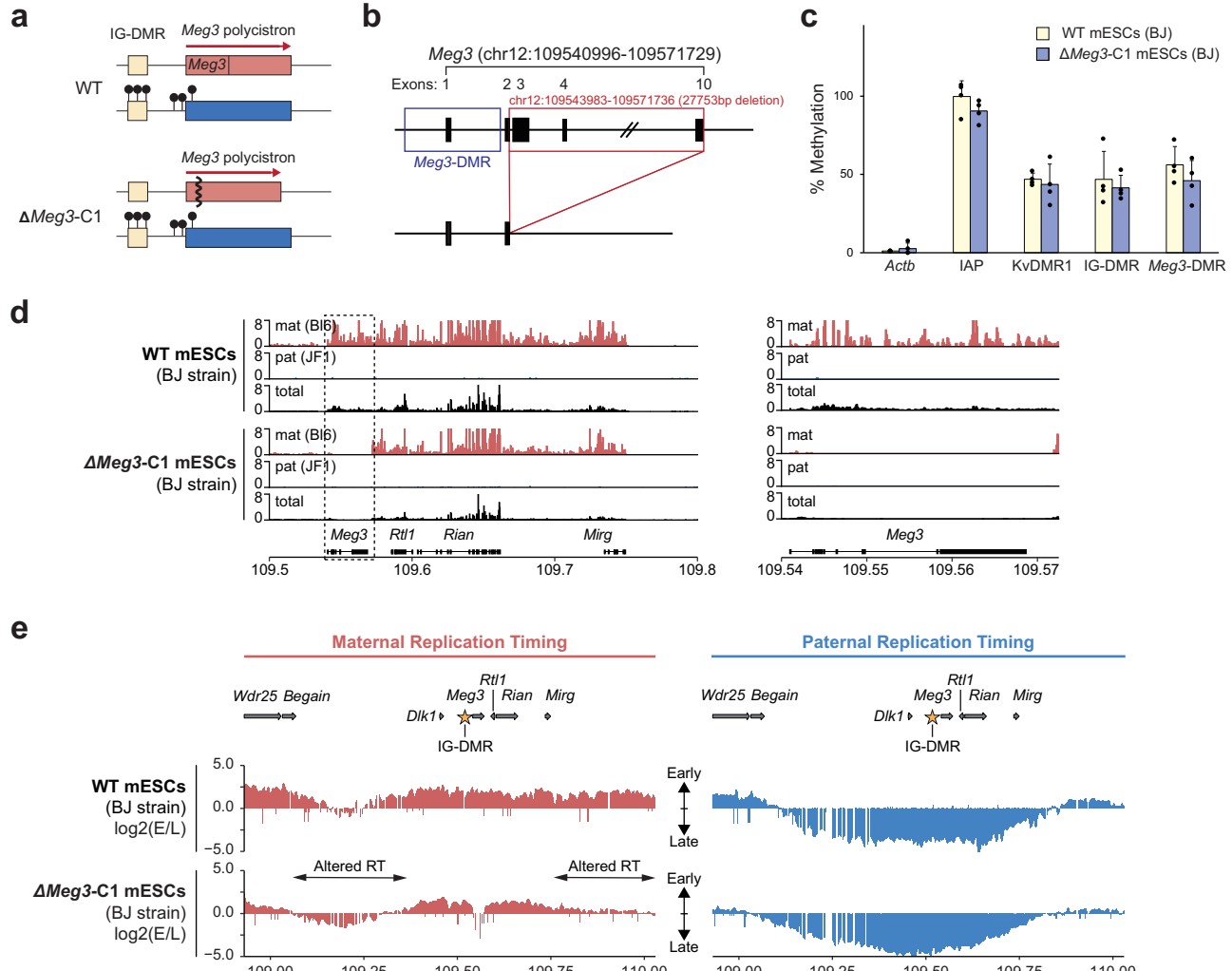

**Fig. 5 | Meg3 lncRNA mediates early DNA replication at proximal and distal regions. a**, **b** Generation by CRISPR-Cas9 technology of mESCs with a 27.7-kb deletion comprising intron-2 to exon-10 of *Meg3* (line Δ*Meg3*-C1). **c** Methylation-sensitive qPCR analysis in Δ*Meg3*-C1 and WT mESCs at the IG-DMR, *Meg3*-DMR, KvDMR1 (control ICR), *ActB* promoter (low methylation) and *IAP* elements (high methylation). Bars represent means ±S.D. from 4 independent experiments. **d** RNA-

seq analysis of the WT (BJ) and Δ*Meg3*-C1 mESCs, on the maternal versus the paternal chromosome. In the tracks with the total reads, levels were adjusted against genome-wide RNA expression levels. Boxes show genes; *Meg3* is enlarged in the right panel. **e** Allelic RT determined by Capture Repli-seq. Two-headed arrows indicate zones with altered RT in Δ*Meg3*-C1 relative to WT (BJ) mESCs. Gene positions are aligned above; the asterisk indicates the IG-DMR.

C1 cells, the *Dock7* control locus showed unaltered early replication (Supplementary Fig. 5e).

### Functional exploration of putative replication origins within the *Dlk1-Dio3* domain

Given that DMR methylation is essential for asynchrony replication at the *Dlk1-Dio3* domain, we wondered whether putative replication origin zones within or close to the DMRs could be involved. To test this

hypothesis, first, we refined the extent of the IG-DMR and *Meg3*-DMR differential methylation using published genome-wide bisulphite sequencing datasets[50]. This showed that the paternal allele-specific methylation covers ~10-kb at the IG-DMR and ~6-kb at the *Meg3*-DMR (Supplementary Fig. 6a). Next, we explored published data on replication origins in mESCs[51]. This pinpointed two zones enriched for putative replication origins within the DMRs: one flanking the IG-DMR on its 3' side, and one within intron-1 of *Meg3* (Supplementary Fig. 6a).

To explore the role of the replication origins located 3' of the IG-DMR, we deleted a 1.2-kb region in hybrid mESCs by CRISPR-Cas9 using two guide RNAs. One clone with biallelic deletion, called ΔOri$^{-/-}$, was selected for replication studies (Supplementary Fig. 6b,c). In these cells, the IG-DMR and *Meg3*-DMR had unaltered DNA methylation, but there was reduced *Meg3*, *Rian*, and *Mirg* expression, which is explained by this element's role in transcriptional regulation[52] (Supplementary Fig. 6d, e). qPCR across SNPs showed unaltered early replication at *Meg3* and *Dlk1* on the maternal, and unaltered late replication on the paternal chromosome (Supplementary Fig. 6f, compare with Fig. 2c, d). To explore the role of the origins within *Meg3* intron-1, we took advantage of a previously generated cell line that carries a 2.2-kb deletion in the first intron of *Meg3*[49]. These 'Δintron-1$^{-/-}$' mESCs did not express the *Meg3* polycistron, as previously reported[49], and have full methylation at the *Meg3*-DMR (and concordant loss of *Meg3*, *Rian* and *Mirg* expression) and unaltered methylation at the IG-DMR (Supplementary Fig. 6d, e). Despite the *Meg3*-DMR methylation change and the removal of the putative replication origins, *Meg3* and *Dlk1* remained early-replicating on the maternal, and late replicating on the paternal chromosome (Supplementary Fig. 6g). The above data suggest that the differentially methylated regions with putative origins, one by one, are not essential for the replication asynchrony at *Meg3* and *Dlk1*.

At several other imprinted loci (*Igf2r*, *Grb10*, *Rasgrf1*, *Cdh15*, *Peg1*, and *Gnas*) we find that the germline DMRs (Fig. 1d) comprise multiple replication origins in mESCs (data from ref. [51]). At these loci, however, there was no apparent replication asynchrony. In general, our study does not link putative replication origins activity at imprinted DMRs to asynchronous RT.

### Imprinted asynchronous replication and parental chromosome-specific TADs show distinct boundaries

To explore possible links between RT asynchrony and allele-specific chromatin structure, we performed genome-wide Hi-C on the monoparental mESCs (80-kb bin size). The genome-wide A (active) versus B (inactive) compartments—determined using Eigenvector decomposition of the Hi-C matrix—were similar in the androgenetic and parthenogenetic mESCs, as shown for chromosome 12 (Fig. 6a). As reported by others before[53,54], the A compartment (positive values) overlapped with early-replication domains, while negative values correlated with late-replication domains (Fig. 6a). Interestingly, the imprinted *Dlk1-Dio3* domain showed intermediate values on both parental chromosomes, which were however lower in the androgenetic cells that had late replication at the domain (Fig. 6b, bins are 40-kb). At the *Dlk1-Dio3* domain, asynchronous replication thus associates with minor differences in A/B compartment organization.

To explore whether parental-chromosome-specific replication zones could be linked to TAD architecture, we performed allele-specific Capture Hi-C, similarly as reported before[55], supplemented with allelic ChIP-seq for the CTCF insulator protein. In BJ mESCs, in agreement with our earlier studies[55,56], the maternal chromosome adopts a TAD organisation that is different from that on the paternal chromosome. Specifically, a maternal-specific sub-domain forms with boundaries, as defined by local minima in the insulation score[57], that aligned with a CTCF binding site at the *Meg3*-DMR on one side, and on the other side with a CTCF binding site downstream of *Mirg* that could not be assigned to either parental chromosome due to the lack of a discriminative SNP (Fig. 6c). Comparison with the 750-kb asynchronous replication zone did not reveal apparent concurrence with the positioning of TADs. Specifically, the maternal early-replicating domain was considerably larger than the structural sub-domains detected by Capture Hi-C. As a result, the early-replicating edges of Transitional Timing Regions (TTRs) on both sides were positioned >150 kb away from the sub-TAD boundaries (Fig. 6c, arrows), exceeding the very large majority of genome-wide TRR-TAD boundary pairs in earlier research[4]. We also explored a possible link between RT domains

and sites of CTCF binding. Again, there was no apparent co-localization between CTCF binding and RT domain boundaries, except for the biallelic CTCF binding site, upstream of *Dlk1*, which located at the extremity of the TTR on the maternal allele (Fig. 6c, grey shaded; Supplementary Fig. 7a–c).

Comparative Capture Hi-C on the *Zfp57*$^{-/-}$ mESCs --in which both the parental chromosomes showed early replication timing across the entire domain—showed that the TAD structure on the paternal chromosome had become similar to that on the maternal chromosome (Fig.6d). This finding implies that DMR methylation and lack of polycistron expression do not only contribute to differential RT, but also to allelic differences in TAD structure. Yet again, we observed no apparent link between RT domains and TAD organisation or overlap between their boundaries (Fig. 6d). Similar comparisons were performed for *Igf2-H19* and *Kcnq1*, which showed no overt overlap between RT domains and TAD structure in the WT and *Zfp57*$^{-/-}$ mESCs (Supplementary Fig 7a–f).

### Imprinted replication asynchrony resolves during neural differentiation

To explore whether replication asynchrony persists during differentiation, we generated neuronal progenitor cells (NPCs) using a previously reported protocol[55] (Fig. 7a). After 9 days of differentiation, most cells expressed *Nestin*, a marker of proliferative NPCs. Only a minority of cells expressed *Tubulin-B3*, an early marker of post-mitotic neurons (Fig. 7b). In agreement with our earlier study[55], NPC differentiation did not alter the methylation levels at ICRs (Fig. 7c, d). Following BrdU incorporation for 1-h, 'early' and 'late' fractions were sorted as for the mESCs (Supplementary Fig. 8a). In agreement with its reported repression[58], array hybridisation and qPCR showed that *Dppa2* replicated late in the NPCs, as reported before[35]. *Ptn1* becomes activated upon neural differentiation[36], and showed early replication (Fig. 7e and Supplementary Fig. 8b). Other marker genes showed correct RT as well (Supplementary Fig. 8c). In agreement with earlier studies on NPCs[11,20], about 37% of the genome significantly changed its RT upon neural differentiation (Supplementary Table 1), as exemplified by the RT profile of chromosome 12 (Supplementary Fig. 8d).

*Dlk1* becomes activated on the paternal chromosomes during NPC differentiation, whereas the maternal *Meg3* expression is maintained[26]. Array hybridization in the hybrid mESC-derived NPCs showed a shift towards early replication in the vicinity of *Dlk1* and the *Meg3* polycistron (Fig. 7f), which was confirmed by qPCR and Sanger sequencing (Fig. 7g, h). To determine the precise zone of altered RT, we performed Capture Repli-seq. The maternal chromosome remained early-replicating across the central part, from a LINE-rich segment[59] until the *Mirg* gene (Fig. 7i). On the paternal chromosome, conversely, the central part of the domain notably shifted towards intermediate RT in NPCs (Fig. 7i), as compared to its late replication in mESCs (Fig. 2e, f). This late-to-intermediate switch in the central part occurred despite the maintenance of the paternal methylation imprint (Fig. 7c, d). In addition, the proximal and distal parts of the domain were no longer earlier replicating on the maternal chromosome in the NPCs. Combined, these data show extensive developmental remodelling of allelic replication timing.

The parental chromosomes broadly retained their differential TAD organisation in the NPCs, despite the marked shifts in RT (Fig. 7i, compare with Fig. 6c). Like in mESCs, we explored whether apparent co-localization could be noted between RT and TAD organization. Although the observed changes in RT upon differentiation towards NPCs could not globally be linked to TAD organization, on the paternal chromosome, a larger sub-TAD may be observed whose boundaries more closely coincide with the extent of the domain with intermediate RT. The absence of obvious valleys in the insulation score indicates a less discrete boundary organization, though. Nonetheless, this may suggest that on the paternal chromosome, upon differentiation

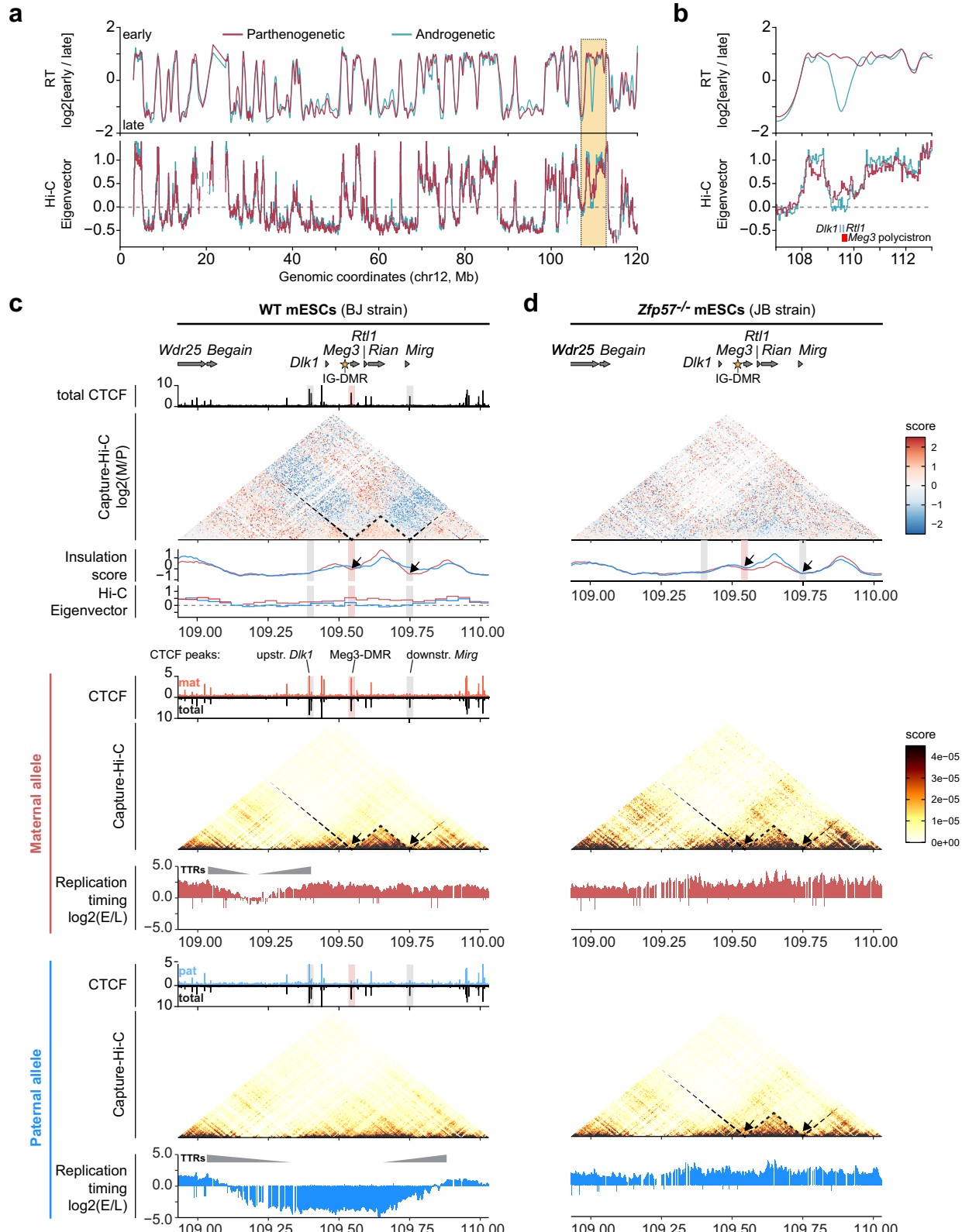

towards NPCs and the associated activation of the *Dlk1* gene, a certain coincidence between RT and TAD structure may arise.

At the *Snrpn* domain, the NPCs showed a shift towards earlier replication compared to the mESCs (Supplementary Fig. 8e), which could correlate with gene activation during in vitro differentiation into NPCs[60,61]. Allelic qPCR and Sanger sequencing no longer showed allelic differences in the NPCs. The parental chromosomes were equally

represented in late S, during which most of the replication occurred, suggesting an attenuation of the RT asynchrony (Supplementary Fig. 8f).

Capture Repli-seq indicated that *Igf2-H19* replicated early on both the parental chromosomes (Supplementary Fig. 8b, compared to 'middle' in mESCs, Supplementary Figs. 2b, 1d). The adjacent *Kcnq1* domain remained early replicating on both parental chromosomes in

**Fig. 6 | Parental chromosome-specific 3D chromatin architecture is disconnected from RT at the *Dlk1-Dio3* domain in mESCs. a** Eigenvector-based assessment of the A versus B compartment in androgenetic and parthenogenetic mESCs (bins are 80-kb) on chromosome 12, compared to the RT profiles determined in Fig. 1. The yellow box shows the *Dlk1-Dio3* region, depicted at higher resolution (bins are 40-kb) (**b**). **c** Comparison of the 3D-chromatin organization between the parental chromosomes using allelic Capture Hi-C in hybrid mESCs (BJ). Bins are 5-kb. Stronger signal on the maternal chromosome (log2 ratio) is shown in red, a stronger signal on the paternal chromosome in blue. The dashed line outlines the maternal-specific sub-TADs described before[55,56]. Capture Hi-C data are aligned with insulation scores on the maternal (red) and the paternal chromosome (blue), with Eigenvector values as a measure of A/B compartments (inferred from Hi-C in monoparental cells; red/blue signal), with allelic (red/blue tracks) and total (black tracks) CTCF binding[56] and with RT for both the maternal (red) and the paternal (blue) chromosomes. Grey triangles highlight potential TTRs (Timing Transition Regions). Gene positions are indicated above the panels; the asterisk indicates the IG-DMR. The boundaries of the maternal *Meg3-Rian*-Mirg sub-TAD are indicated with dashed lines on the heatmaps. The red bar indicates the maternal chromosome-specific *Meg3*-DMR CTCF binding. Grey highlights indicate CTCF peaks upstream *Dlk1* and downstream of *Mirg* that are important for the locus' architecture[55]. Black arrows indicate insulation score minima and the linked TAD borders. **d** Comparison of the 3D-chromatin organization between the maternal and the paternal chromosome in *Zfp57*[-/-] mESCs. Bins are 5-kb. Capture Hi-C data are aligned with allelic insulation scores and RT for both the maternal (red) and the paternal (blue) chromosomes.

NPCs (Supplementary Fig. 8g). This was linked to unaltered DNA methylation levels at the KvDMR1 (Fig. 7c). Capture Hi-C across this 1.2-Mb region showed a comparable Hi-C pattern as in the hybrid mESCs; yet with stronger looping on the maternal chromosome originating from the *Kcnq1* gene body (Supplementary Fig. 8g, compared with Supplementary Fig. 7a–c). Although the entire region shows early replication, it is structured in several 3D-compartments and loops on both the parental chromosomes, with boundaries that do not co-localise with RT boundaries, like in mESCs (Supplementary Fig. 8g). At the control chromosome 4 region, both the parental chromosomes remained early replicating in NPCs.

We conclude that imprinted asynchronous replication, only observable at two imprinted domains in stem cells in this study, attenuates or resolves during neural differentiation, and these RT changes are not associated with concordant changes in TAD organisation.

## Discussion

Our genome-wide approaches combined with high-resolution targeted assays revealed pronounced replication asynchrony between the parental chromosomes at the *Dlk1-Dio3* and *Snrpn* imprinted domains, which both comprise a large lncRNA polycistron. For *Dlk1-Dio3*, we found that the RT asynchrony depended strictly on the paternal allele-specific DNA methylation at its DMRs. In addition, on the maternal chromosome, the DMR-activated Meg3 lncRNA mediates early replication in *cis* across proximal and distal parts of the domain (Fig. 8). Another key insight is that RT domains seem unlinked to parental-chromosome-specific TAD organization at imprinted domains, notably in mESCs, which presents an intriguing difference with genome-wide trends.

Using different RT assays, our study revealed pronounced replication asynchrony at the conserved, disease-associated *Dlk1-Dio3* and *Snrpn* domains, but not at other imprinted domains. In contrast to an earlier genome-wide study on mESCs, in which imprinted asynchronous replication was not reported[20], we used mESCs cultured under serum-free culture conditions with ascorbic acid complementation[32], to prevent aberrant DNA methylation at DMRs. This gave stable maintenance of methylation imprints in the various cell models and conditions used. Significantly, a recent study on human mono-parental ESCs also concluded that imprinted RT is limited to the *Dlk1-Dio3* and *Snrpn* domains, although the molecular mechanisms involved were not explained[44]. In our study, we successfully addressed the mechanism regulating asynchronous replication. Using multiple hybrid lines, our findings demonstrate that parental methylation imprints are clearly essential at both imprinted domains. This epigenetic mark, however, is insufficient on its own, since replication asynchrony resolved upon differentiation, and was not observed along other methylation-controlled imprinted domains (Fig. 1d). At the *Igf2-H19* and *Kcnq1* domains, for instance, despite maintained DNA methylation, Capture Repli-seq did not provide evidence for replication asynchrony between the parental chromosomes. The *Dlk1-Dio3* and

*Snrpn* domains share that they are both Mb-sized and uniquely comprise large polycistrons that express lncRNAs and multiple snoRNAs. Several of the polycistron-expressed ncRNAs are nuclear and retained in *cis* at these loci—including Meg3 lncRNA and partially processed forms of the Rian snoRNAs[26,49,62]—and might thus be functionally involved in RT.

Based on our findings and recent studies on human ESCs[43,44], we hypothesize that lncRNA in-*cis* retention controls RT, possibly through titration of nuclear factors that influence DNA replication. A similar mechanism has been suggested for human ASAR lncRNAs, which are essential for the maintenance of (synchronous) early replication on human chromosomes 6 and 15[63,64]. As a first step to test this hypothesis, we deleted part of the *Meg3* promoter in ZFP57-deficient hybrid mESCs such as to completely lose the expression of the polycistron in mESCs, which also no longer had differential DNA methylation. Importantly, the altered RT in these cells pinpointed a contribution of the polycistron expression primarily in the proximal "Begain-Dlk1 region", which could coincide with the zone of Meg3 cis-accumulation (Fig. 8)[49]. By deleting *Meg3*, while keeping the remainder of the polycistron (and methylation profiles) unaltered, we found that Meg3 lncRNA contributes to the early replication on the maternal chromosome at this proximal region, and to a lesser extent in a distal region close to *Mirg*. Although this shift from early-to-late replication along non-overlapping proximal and distal parts in the Δ*Meg3*-C1 cells pinpoints the involvement of the lncRNA, the precise mechanism(s) remain to be addressed in future studies.

One limitation of exploring deletions that ablate Meg3 lncRNA expression (Δ*Meg3*-C1 and *Zfp57*[-/-];*Meg3*-pro[-/-]) is that these may have comprised regulatory sequences that influence RT. This could be relevant particularly for the Δ*Meg3*-C1 line in which we deleted a large region, from exon 2 until exon 10 of *Meg3*. However, since these two independent, non-overlapping deletions both led to delayed replication timing within the imprinted domain, the most parsimonious model is that the lncRNA expression contributes to the early replication on the maternal chromosome (Fig. 7). Another limitation is that the deletion lines in our study were not all of the same hybrid genotype. As highlighted by the minor differences observed between the BJ and JB mESCs (Fig. 2e,f), also this may have influenced the observed effects on replication timing.

Replication domains statistically align with TADs in mammalian cells, while early and late replication correlate with A and B compartments, respectively[4,5,10,54,65], indicating interactions between chromatin architecture and the replication programme. This relationship is further illustrated by the enrichment for CTCF binding, a master regulator of TAD organization, at RT domain boundaries[5,66]. However, at the studied chromosomal domains with imprinted asynchronous replication, the situation seems to be different. The high-resolution Repli-seq on *Dlk1-Dio3* showed that both RT and TAD structure are different between the parental chromosomes in hybrid WT mESCs, with no evident overlap between TAD boundaries and RT. In agreement with

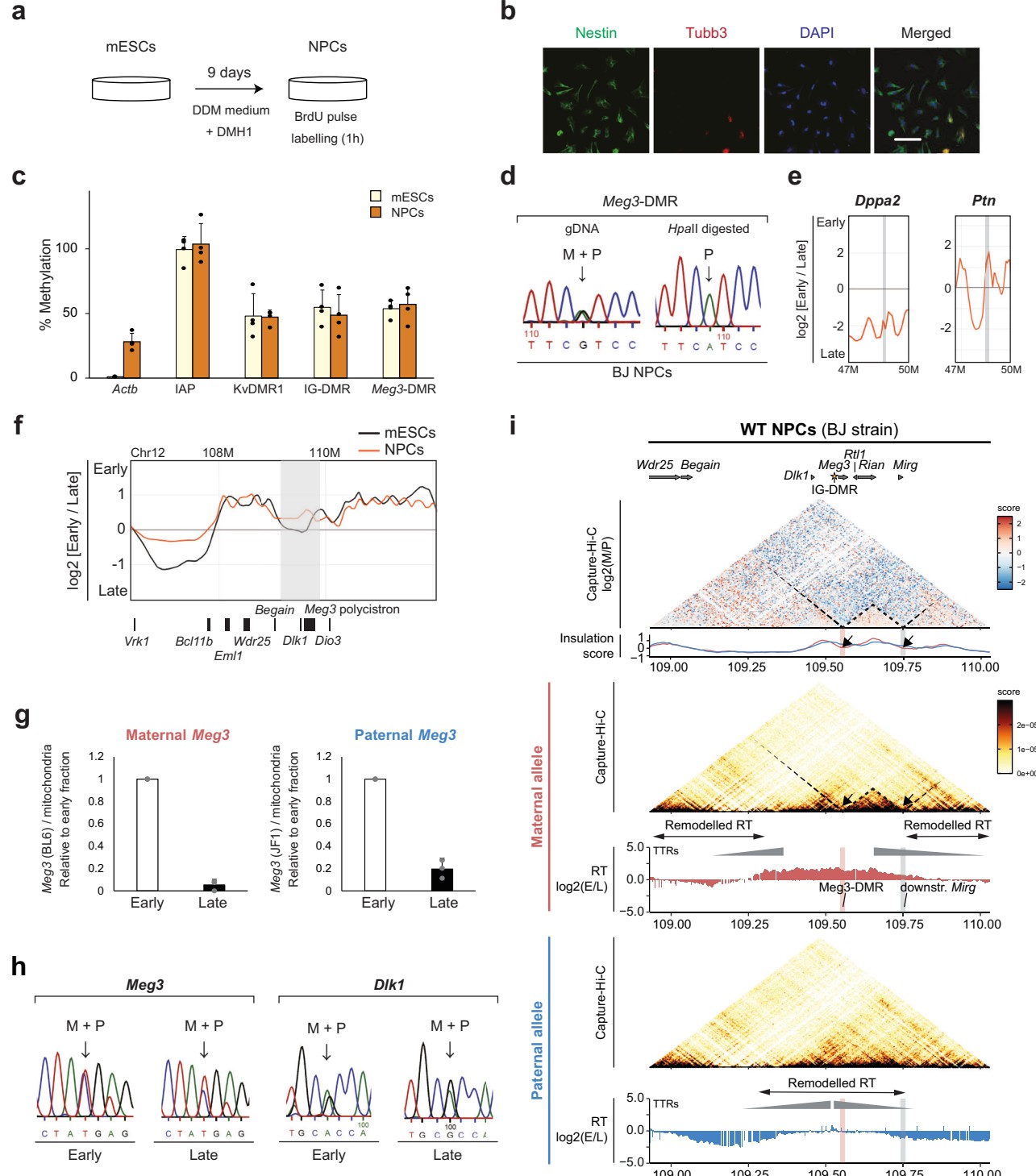

**Fig. 7 | Replication asynchrony resolves during neural differentiation irrespective of TAD organisation. a** Schematic of mESC differentiation into NPCs in the presence of DMH1[55]. **b** Immunofluorescence staining of Nestin and Tubulin-B3, with DAPI counterstaining, in the obtained day-9 NPCs. Scale bar is 50 μm. The same result was obtained in 2 independent experiments. **c**, Methylation qPCR analysis in mESCs and NPCs at the IG-DMR, *Meg3*-DMR, KvDMR1 (control ICR, *Kcnq1* domain), *ActB* (low methylation), and *IAP*s (high methylation). Bars represent means ±SD from 4 independent experiments. **d** *Hpa*II digestion indicates paternal methylation at the *Meg3*-DMR in NPCs. **e** Micro-array RT profiles in day-9 NPCs at the *Dppa2* (late) and *Ptn* (early) reference loci. Vertical lines indicate the gene positions. **f** Micro-array RT profiles of day-9 NPCs compared to mESCs at the *Dlk1-Dio3* region. Gene positions are shown underneath. Light grey indicates the region with asynchronous replication in mESCs. **g** RT at *Meg3* analysed by allelic qPCR in early and late fractions. Bars

represent means ± SD from 3 experiments. **h** Sanger sequencing-based allelic assessment of *Meg3* (left) and *Dlk1* (right) in NPC early and late fractions. Arrows indicate the SNPs used to distinguish the maternal (M) and paternal (P) alleles. **i** Capture Repli-seq in day-9 NPCs shows remodelled RT (indicated by two-headed arrows) compared to mESCs (Fig. 2). RT is aligned with 3D-chromatin organization maps, with the maternal-versus-paternal comparison matrix above (bins are 2.5-kb). The interrupted line outlines a maternal-specific sub-TAD, similarly as described before[55,56]. Gene positions are aligned above the panels; the asterisk indicates the IG-DMR. The red highlight indicates maternal chromosome-specific *Meg3*-DMR CTCF binding. The grey highlight indicates the CTCF binding site downstream of *Mirg* involved in the maternal chromosome-specific sub-TAD in both NPCs and mESCs[55]. Black arrows indicate insulation score minima in NPCs and the corresponding sub-TAD borders. Grey triangles highlight potential TTRs (Timing Transition Regions).

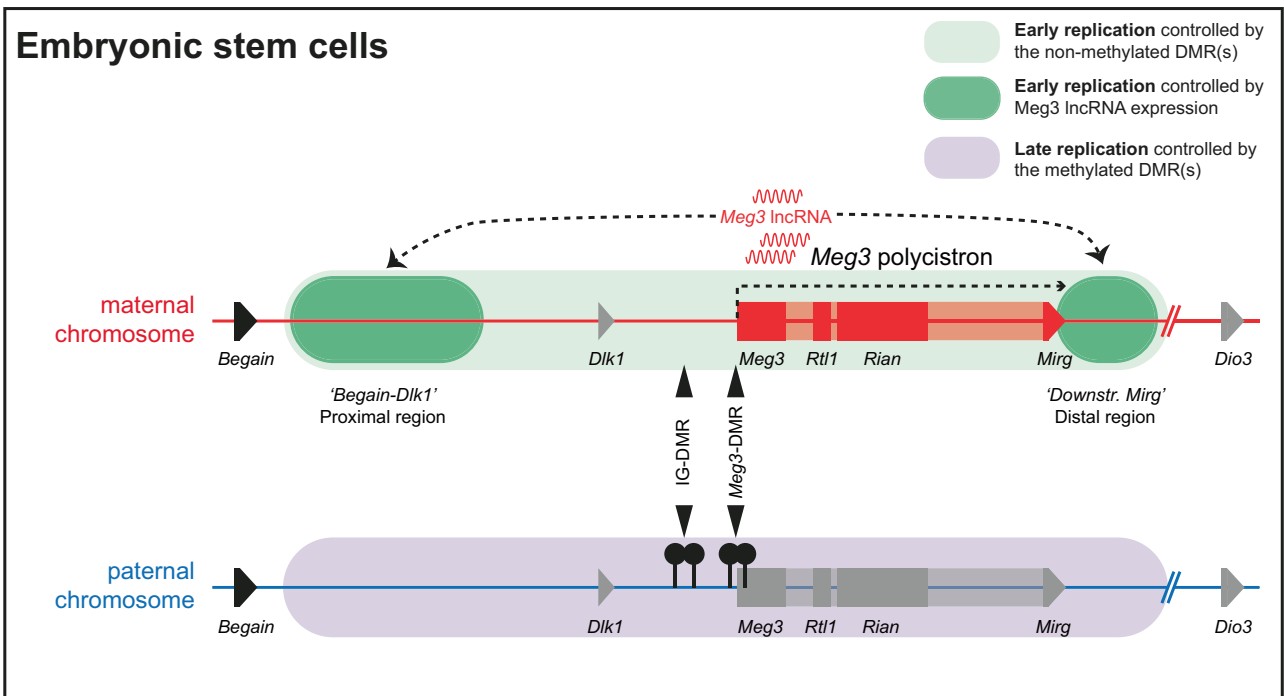

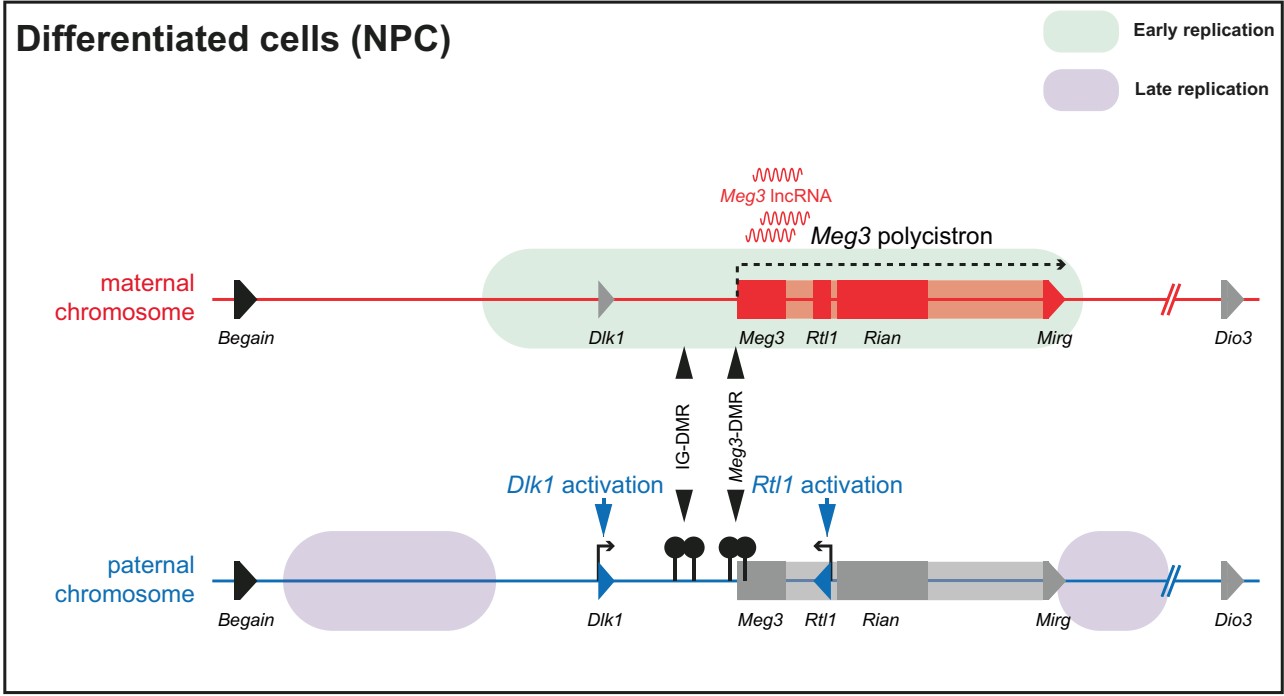

**Fig. 8 | Model for the regulation of asynchronous replication at the *Dlk1-Dio3* imprinted domain.** In mESCs, asynchronous replication is observed across ~750-kb. Methylation at the domain's DMRs confers late replication, possibly as a consequence of its resulting lack of gene expression. The unmethylated status of the DMRs, conversely, is essential for early replication. Early-replication is also conferred in the proximal and distal parts of the locus by the DMR-driven expression of the 220-kb *Meg3-Rian-Mirg* polycistron. This generates Meg3 lncRNA that accumulates in cis[26,55] and is important for the early replication at these parts. The RT boundaries appeared largely independent of CTCF binding and 3D chromatin

architecture. Upon differentiation into NPCs, despite the maintained DMR methylation, there is no longer late replication on the paternal chromosome in the central region encompassing *Dlk1* and the *Meg3-Rian-Mirg* polycistron, possibly because of the developmental gene activation. On the maternal chromosome, only the central part remained early-replicated, despite maintenance of *Meg3* lncRNA expression. This difference suggests that, in mESCs, in addition, pluripotency-associated factors are required for asynchronous replication at the proximal and distal regions.

earlier allelic studies[20,43], differentiation into NPCs resulted in the loss of asynchronous replication, whereas the allelic TAD structure was globally maintained. At the flanking *Igf2-H19* and *Kcnq1* imprinted domains on chromosome 7, we detected early replication across the

entire captured 1.3-Mb region on both parental chromosomes, despite being structured in different TADs. We conclude that, at the domains analysed, there is no apparent link between 3D chromatin architecture and replication timing. A compelling hypothesis could be that

epigenetic regulatory layers at play at imprinted domains may partially disrupt such structural links.

The question remains as to what dictates the specificity of the imprinted asynchronous replication (Fig. 8). Differential DNA methylation is clearly essential at the *Dlk1-Dio3* and *Snrpn* domains, but replication origin zones within the DMRs of the *Dlk1-Dio3* domain are not required for the replication asynchrony, at least not individually. On the maternal chromosome, we find, however, that part of the RT effect is mediated by the DMR-driven allelic expression of the lncRNA polycistron, at non-overlapping regions in *cis*. On the paternal chromosome, the domain's DMRs are methylated, and the associated late replication is less well understood, but could be linked to the low levels of gene expression on this parental chromosome, at least in mESCs[36]. Irrespective of the precise downstream mechanism(s), our studies provide the first demonstration that DNA methylation functionally controls imprinted asynchronous replication. Previous studies by others reported that DNA methylation does not play a major role in replication timing in general. In mESCs triple knock-out for *Dnmt1*, *Dnmt3a*, and *Dnmt3b*, for instance, there were no significant changes in replication timing at the genome-wide level[67]. In human colorectal cancer cells double-KO for *DNMT1* and *DNMT3A*—which induced strongly reduced DNA methylation levels—there was a reduced precision of RT, with more variability between cells, but only 3% of gene loci showed shifts in replication timing[68]. Further studies are required in mESCs and other model systems to better understand the role of DNA methylation in asynchronous replication and the nuclear factors involved, including ZFP57, some of which could be linked to pluripotency.

## Methods

### mESC derivation, maintenance, and differentiation
mESC lines hybrid between C57BL/6 J and *M. m. molossinus* strain JF1[69] were derived under serum-free conditions in ESGRO 2i medium (Sigma-Aldrich, SF016-200)[26,49], and were maintained on gelatine-coated dishes in ESGRO 1i medium (Sigma-Aldrich, SF001-500P), supplemented with 50 µg/ml L-Ascorbic acid to prevent acquisition of aberrant DNA methylation[32]. Sh-1[26] and Δintron-1[-/-] mESCs[49] are of (C57BL/6 J x JF1)F1 genotype, and *Zfp57*[-/-] mESCs[47] of (JF1 x C57BL6J)F1 genotype. Mono-parental mESC lines PR3 and AK2 were derived previously[70,71], and were cultured under serum-free conditions in ESGRO 1i medium (Sigma-Aldrich, SF001-500P) supplemented with 50 µg/ml L-ascorbic acid. mESCs were differentiated into NPCs in the presence of DMH1 on gelatine-coated dishes using a previously published protocol[55].

### CRISPR-Cas9 mediated gene targeting in mESCs
gRNAs were designed using the CRISPR-Cas9 guide RNA design checker (https://eu.idtdna.com/site/order/designtool/index/CRISPR_SEQUENCE/). They were flanked with *Bbs*I sticky ends and cloned into pSpCas9(BB)−2A-GFP plasmid (Addgene, #48138). 2.5 µg of each plasmid with sgRNA-insertion was electroporated into 5.0×10^6 mESCs using a Nucleofector™ Transfection 2b device (LONZA) and the Alexa™ Mouse ES cell nucleofector™ kit (LONZA). GFP-positive cells were purified 48 h post-electroporation by flow cytometry (FACS AriaII machine, Becton Dickinson), and individual cells were seeded onto 96-well plates. After 10–12 days of culture, colonies were transferred to 6-well plates to derive clonal mESC lines. Extracted genomic DNAs were subjected to PCR-based genotyping. The gRNAs are provided in the Supplementary Table 3.

### Immunofluorescence staining of cells
This was done as reported before[55]. Primary antibodies were directed against Nestin (Abcam, #ab81755, batch GR154015-3, 1:500) or Tubulin-B3 (Biolegend, #801201, batch B353040, 1:500). Secondary antibodies were goat anti-mouse Alexa fluor 488 (Thermo-Fisher, #A-11011,

1:1000) or goat anti-rabbit Alexa Fluor 594 (Thermo-Fisher, #A-11012; 1:1000). A minimum of 4 images were taken with Confocal Zeiss LSM880 FastAiryscan per experiment. Images were processed with Fiji[72].

### DNA methylation analysis
Methylation levels were analysed through digestion with the methylated-sensitive restriction endonuclease *Hpa*II, followed by qPCR analysis at loci of interest. Briefly, 1 µg of genomic DNA was pre-digested for 3 hours with 1 µl of *Eco*RI (New England Biolabs, #R3101) in a 100 µl reaction volume, which was split into two Eppendorf tubes subsequently. To one tube, 1 µl of *Hpa*II (New England Biolabs, #R0171) was added, but not to the other. Following 37 °C overnight incubation, DNAs were extracted and enzymes inactivated by incubation at 65 °C for 20 min. Samples were then subjected to qPCR; obtained values were normalized to the amplification levels of a region at *Col1a2* not containing an *Hpa*II site. The percentage of methylation for each region was calculated as [*Hpa*II + ] / [*Hpa*II-]. The PCR primers used are in the Supplementary Table 3.

### Gene expression analysis
Total RNA samples were extracted using Trizol LS reagent (Thermo Fisher, #10296010) and Phenol-Chloroform, and were reverse-transcribed into cDNA using random hexamers (Thermo Fisher, #SO142) and Superscript II (Thermo Fisher, #18064022). In subsequent RT-qPCR analyses, measured RNA quantities were normalized to *Gapdh*. Primer sequences are provided in the Supplementary Data file. For RNA-sequencing, total RNAs were extracted with RNeasy-Plus Mini-Kit (Qiagen, #74136), quantified by Qubit 4 Fluorometer (Thermo Fisher), and quality-checked with the Bioanalyzer RNA 6000 Assay kit (Agilent, #5067-1513). 210 ng RNA per sample were used for DNase I digestion (Merck, #AMPD1) and ribodepletion with NEBNext® rRNA Depletion Kit v2 (Human/Mouse/Rat) (New England Biolabs, E7400S). RNA was then processed with Next® Ultra™ II RNA Library Prep Kit for Illumina® (New England Biolabs, E7770) for library preparation, without prior fragmentation. Definitive libraries were obtained after 13 PCR cycles. These were quantified by Qubit, quality-checked with BioAnalyzer High Sensitivity DNA Kit (Agilent, #5067-4626), and paired-end (100-bp) sequenced by BGI Genomics (Shenzhen, Republic of China), with an average of 33 M reads per sample.

### Allele-specific analysis of RNA-seq data
Paired-end sequencing reads were trimmed with Trimmomatic and aligned to the mouse reference genome (mm10) with HISAT2. Aligned reads were sorted with samtools, and genome coverage tracks were generated with bedtools. For allele-specific analyses, trimmed reads were additionally aligned to the JF1/MsJ N-masked genome using HISAT2, and allele assignment was performed with SNPsplit. BAM files were further processed with deepTools bamCoverage to generate bigWig files, normalized as CPM in 100-bp bins.

### BrdU incorporation, cell cycle-dependent FACS, sonication
Asynchronous cell cultures were incubated in medium with 50 µM bromodeoxyuridine (BrdU) for 1 hour at 37 °C, followed by fixation in ice-cold 70% ethanol. For cell cycle analysis, cells were first stained with 80 µg/mL propidium iodide (Invitrogen, P3566) at RT, in the presence of 0.4 mg/mL RNaseA (Roche, #10109169001) for 1 h to degrade RNA in the cells. Flow cytometry was then performed using an AriaII FACS machine (BD Biosciences), and early and late fractions of ~250,000 cells were sorted according to the fluorescence strength of propidium iodide. Sorted cells were recovered in a lysis buffer (50 mM Tris-HCl pH8, 10 mM EDTA, 0.5% SDS, 300 mM NaCl) as described before[33]. Genomic DNA samples were extracted through incubation with 0.2 mg/mL Proteinase-K (Thermo Fisher, #EO0492) at 65 °C for 2 h,

followed by Phenol-Chloroform extraction and EtOH precipitation, and were sonicated into 100-500-bp fragments using a Bioruptor Pico machine (Diagenode), followed by denaturation at 95 °C for 5 minutes.

## Genome-wide analysis of RT by microarray hybridization

Denatured DNA samples were incubated with anti-BrdU antibody (10 µg mouse anti-BrdU (BD Biosciences, #347580, batch 3016583) for 5 h in IP buffer (10 mM Tris, pH 8, 1 mM EDTA, 150 mM NaCl, 0.5% Triton X-100, 7 mM NaOH), followed by a 6 h incubation with Dynabeads Protein G (Invitrogen; #10004D). The beads were then washed with Wash Buffer (20 mM Tris, pH 8.0, 2 mM EDTA, 250 mM NaCl, 1% Triton X-100) and elution was carried out at 37 °C for 2 h in a solution containing 1% SDS and 0.5 mg/mL Proteinase-K, followed by 6-h incubation at 65 °C after bead removal. Immuno-precipitated BrdU-labelled DNA was purified using phenol–chloroform and precipitated with cold ethanol. Control qPCRs were performed using oligonucleotides specific to mitochondrial DNA[73] as well as early and late replicating control regions. Whole-genome amplification was conducted using the SeqPlex Enhanced DNA Amplification Kit, according to the manufacturer's protocol (Sigma-Aldrich; SEQXE). The amplified DNA was purified using a PCR Purification Kit (Macherey-Nagel; #740609.50). Importantly, qPCR was performed again, as for the first step of quality control, to ascertain that the ratio between early and late replication regions was preserved. Purified early and late nascent-DNA fractions were labelled with Cy3-ULS and Cy5-ULS, respectively, using the ULS arrayCGH Labelling Kit (Kreatech; EA-005). Equal amounts of early- and late-labelled DNA were hybridized at 65 °C onto mouse DNA micro-arrays [SurePrint G3 mouse CGH Arrays (4x180K); Agilent Technologies, G4826A] that cover the whole genome with one probe every ~13 kb. The following day, the micro-arrays were scanned with an Agilent C-scanner using a resolution of 2 µm and the autofocus option. Feature extraction was performed with the Feature Extraction 9.1 software (Agilent Technologies). Data analysis was performed using the START-R suite[34]. Differential analysis, based on two independent experiments with two technical replicates each, was conducted using the START-R Analyzer and visualized with START-R Viewer[34].

## Allele-specific RT analysis by qPCR and Sanger sequencing

After BrdU-incorporation, sonication and denaturation, DNA samples were incubated overnight at 4 °C with 25 µg/mL anti-BrdU antiserum (BD Biosciences, #555627) in IP buffer (10 mM Tris-HCl, pH 8.0, 1 mM EDTA, 150 mM NaCl, 0.5% Triton X100), adjusted to 7 µM NaOH to keep the DNA molecules single-stranded. Samples were then incubated with protein-G Dynabeads (Thermo Fisher, #10004D) for 3 h at 4 °C, and washed subsequently with IP buffer, B buffer (20 mM Tris-HCl pH 8.0, 2 mM EDTA, 250 mM NaCl, 0.5% Triton X100), and 10 mM Tris-HCl pH 8.0, sequentially. Samples were then incubated with 0.2 mg/mL Proteinase-K (Thermo Fisher, #EO0492), for 2 h at 65 °C, and for 4 h at 37 °C. DNA was purified using Phenol-Chloroform, followed by EtOH precipitation. The BrdU-incorporated DNA samples from early and late fractions were subjected to Real-time qPCR using a Roche LightCycler 480 Real-Time PCR System and a LightCycler 480 SYBR-Green detection kit (Roche); the obtained data were analysed using LightCycler software (Roche). The intensity of the target gene, normalized to the intensity of a mitochondrial gene, was calculated for each early and late fraction. The relative values of the late fraction to the early fraction were calculated for more than 3 pairs of early versus late DNA samples. Allele-specific qPCR primers were designed in a region containing two SNPs. For each allele-specific primer, the two SNPs are positioned at the 3'-end of one primer in the pair, while the other primer is a common sequence. PCR primers are provided in the Supplementary Table 3.

## Capture Repli-seq

Enrichment of specific genomic regions was achieved using a custom panel of probes, designed by Agilent (SureSelect DNA Design), targeting the following genomic intervals chr12:108930000-110030000; chr7:142300000-143530000 and chr4:98670000-99260000 (mm10). Library construction was performed as described before[74]. BrdU-incorporated DNA was extracted from ~250,000 cells and sonicated into 100-500 bp fragments using a Bioruptor Pico (Diagnode). Using NEBNext Ultra II Library Prep Kit for Illumina (New England Biolabs, E7645S) and NEBNext Multiplex Oligo for Illumina (New England Biolabs, E7645S), adaptors were ligated following the manufacturer's protocol. DNA was then Phenol-Chloroform purified, denatured (95 °C, 5 min), and subjected to BrdU immunoprecipitation as described before[33]. DNA was purified subsequently and was indexed during the amplification stage, using multiplex oligos (New England Biolabs, E7645S) following the manufacturer's method. DNA was then purified with AMPure XP beads (Thermo Fisher, #10136224). The obtained libraries were quantified with Qubit and their quality was assessed with Bioanalyzer 2100 (Agilent Technologies). 4 to 12 Repli-seq libraries were pooled in equimolar quantities (1 µg in total) and captured simultaneously, using the Arima Capture Modules (Arima, #A311032, #A311033, #A311034) according to the SureSelect XT HS2 DNA Kits protocol provided by Agilent. Hybridization of Repli-seq libraries with the SureSelect probes was achieved in a thermal cycler with 60 cycles [1 min at 65 °C, then 3 sec at 37 °C] followed by overnight incubation at 21 °C. The hybridized pool of libraries was pulled down using Streptavidin beads, and libraries were then amplified following the manufacturer's instructions (Agilent) with 12 PCR cycles. Finally, libraries were purified using AMPure XP beads (Beckman Coulter; 1:1 beads:sample ratio), quantified using Qubit dsDNA HS kit (Thermo-Fisher Scientific; Q33230), and their size distribution measured using the TapeStation with the D1000 kit (Agilent). The material was sequenced on an Illumina NextSeq 2000 (paired-end, 2x50 cycles) at the High-throughput sequencing facility of the I2BC (CNRS, Gif-sur-Yvette).

## Capture Repli-seq data analysis

Capture Repli-seq reads were processed as described before[74], with only minor adjustments to accommodate allele-specific analyses. First, sequencing reads were mapped using bowtie2 to a modified N-masked GRCm38/mm10 genome in which all variants for *Mus musculus JF1* were replaced by the ambiguous base 'N' (N-masked genome prepared using SNPsplit tools; variant information obtained from the Mouse genome project). PCR duplicates were then removed using Samtools rmdup, and reads were assigned to each of the parental genotypes (Bl6 or JF1) using SNPsplit. For the rest of the analyses, we followed a published procedure[74], except for the span of the loess smoothing parameter, which was fixed to 0.01. Plots were generated using R's ggplot2 library.

## Hi-C analysis and A/B compartment allocation in monoparental mESCs

Hi-C experiment in mono-parental mESCs (AK2 and PR8) was performed using the Arima Hi-C+ kit (Arima Genomics), following the manufacturer's instructions. Hi-C libraries were sequenced on the Illumina NovaSeq 6000 system (2 × 150). The reads were processed using Hi-C pro[75]. Resulting ValidPairs tables were converted into mcool files, using Hi-C pro hicpro2higlass.sh script, using --res 10000 and --norm parameters. A/B compartments were then called at 80-kb and 40-kb resolution, using the cooltools eigs-cis module and phased using GC content. Eigen-vector values were plotted using R, at an 80-kb resolution for the whole genome view, and at a 40 kb resolution for local analyses at the *Dlk1-Dio3* domain.

## Capture Hi-C

Experiments and data analysis were performed as reported recently[55], using the Arima Hi-C+ kit (Arima Genomics), following the manufacturer's instructions. Sequencing libraries were prepared using the SureSelect XT HS2 DNA System kit (Agilent), and target enrichment was achieved on pools of 4 libraries using a custom panel targeting the genomic intervals chr7:142,240,000-143,530,000; chr12:108,840,000-110,050,000; chr4:98,670,000-99,260,000 (mm10). Captured Hi-C material was sequenced on the Illumina NextSeq 2000 system (2×60 bp). The read pairs were initially processed using AGeNT Trimmer to trim the adaptors and dark bases. Trimmed read pairs were then processed using Hi-C pro[75] to generate the Hi-C allelic matrices. 2-3 technical replicates were combined, and Hi-C matrices were displayed using R. Insulation scores were calculated using the GENOVA R package[76] with a 20-bins window.

## Analysis of ChIP-seq data

CTCF ChIP-seq raw data for BJ1 and JB1 ESCs were from Farhadova et al. 2024 (GEO record GSE207166). Reads were aligned with Bowtie 2 to a modified N-masked GRCm38/mm10 genome in which all variants for *Mus m. molussinus* JF1 were replaced by the ambiguous base 'N' [N-masked genome prepared using SNPsplit tools[77]] and variant information obtained from the Mouse genome project. PCR duplicates were removed using the Samtools markdup, and reads were assigned to their respective allele using SNPsplit. A bedgraph for uniquely mapped and paired reads was generated using bedtools for each allele. Bedgraphs were scaled to a total of 10 million uniquely mapped and paired reads on both JF1 and C57BL/6 J alleles. Bedgraph was visualized and displayed using the R ggplot2 library.

## Bio-informatic tools

The following bioinformatic tools were used in this study: Trimmomatic (version 0.39), HISAT2 (version 2.2.1), bamCoverage (version 3.5.1), Bowtie2 (version 2.4.2), Samtools (version 1.6), SNPsplit (version 0.3.4), Bedtools (version 2.31.0), Awk (version 5.1.0), R (version 4.04), ggplot (version 3.4.2), preprocessCore (version 1.52.1), Agent Trimmer (version 3-2), GENOVA (version 1.0.1).

## Reporting summary

Further information on research design is available in the Nature Portfolio Reporting Summary linked to this article.

## Data availability

The array-based replication data of this study are available on the NCBI's Gene Expression Omnibus database under GEO series accession number GSE287936. The Capture Repli-seq data of this study are under accession GSE289027. The Hi-C data (mono-parental mESCs) of this study are available under GEO series accession GSE306486. The RNA-seq data of this study are available under the GEO Series accession number GSE306487. Capture Hi-C data of this study are available under the GEO Series accession number GSE289022. The CTCF ChIP-seq data used in this study were retrieved using the GEO Series accession number GSE207166. Source data are provided with this paper.

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

## Acknowledgements
We thank members of our teams for discussion and comments on the manuscript, the FACS service at the IGMM, and the I2BC high-throughput sequencing facility supported by France Génomique (funded by the French National Program "Investissement d'Avenir" ANR-10-INBS-09). This work was supported by the Fondation pour la Recherche Médicale (FRM, grant EQU202103012763 to R.F.), the Agence Nationale de Recherche (ANR-18-CE12-0022-02 IMP-REGULOME to R.F. and D.N., ANR-22-CE12-0016-03 IMP-DOMAIN to R.F. and D.N.). Y.I. acknowledges Fellowship funding from the Japan Society for the Promotion of Science (JSPS). The funding agencies had no role in study design, data collection, and data analysis, or in the manuscript preparation and decision to publish.

## Author contributions
Y.I.: conceptualisation, investigation, formal analysis, data curation, writing–review & editing; F.C.: investigation, formal analysis, data curation, writing–review & editing; C.S.: investigation, formal analysis; C.P.: conceptualisation, investigation; P.A-R.: investigation; J-C.A.: supervision, writing–review & editing; D.N.: supervision, conceptualisation, funding acquisition, writing–review & editing; B.M.: supervision, conceptualisation, data curation, software, writing–review & editing; J-C.C.: conceptualisation, investigation, formal analysis, writing–review & editing; R.F.: supervision, conceptualisation, funding acquisition, writing–original draft, writing–review & editing. The manuscript was read and approved by all authors.

## Competing interests
The authors declare no competing interests.
