## [Transparent Peer Review file · Nature Communications]

DNA methylation and lncRNA control asynchronous DNA replication at specific imprinted gene domains

Corresponding Author: Dr Robert Feil

Version 0:

Reviewer comments:

Reviewer #1

(Remarks to the Author)

This study by Imaizumi et al. utilizes uni-parental and hybrid mouse ES cell lines to examine the asynchronous replication timing of imprinted gene domains using genome-wide replication timing assays and Capture Repli-seq. The authors successfully delineate detailed asynchronous replication patterns across the Dlk1-Dio3 and Snrpn domains, where lncRNA expression is regulated by DMR DNA methylation. The observed parent-specific replication timing was affected either by the DNA methylation status of the DMR or by a small deletion that abolished lncRNA expression within the domain. Overall, the experiments are well-designed and executed, and the obtained data are solid. However, the authors fail to provide mechanistic insights into why the local methylation status at the DMR has a domain-wide impact on replication timing. Additionally, it remains entirely unclear how lncRNA expression is involved in the early replication of the Begain-Dlk1 region, given that the lncRNA coding region does not overlap with the Begain-Dlk1 region. This study is interesting in that it identifies factors (DMR DNA methylation and lncRNA expression) that affect parent-specific replication timing patterns. However, an important question remains regarding how domain-wide asynchronous replication timing is established. Therefore, this study is considered preliminary.

I have the following comments:

Major comments:

1. The author mentioned that Igf2-H19, Peg3/Zim, and RasGrf1 show potential minor asynchrony (line #85), while Snrpn is considered to exhibit putative asynchrony. However, in Extended Data Figure 1 and Figure 1g, the levels of asynchrony between Igf2-H19, Peg3/Zim, RasGrf1, and Snrpn appear quite similar. It is unclear how the author determined which regions were categorized as asynchronous or synchronous. This is because the author did not clearly explain how many imprinted regions were identified as asynchronous or which specific regions met the criteria for asynchrony (as mentioned in Figure 1 legend, asynchronous replication was determined using START-R software with the Mean method and a p-value threshold of 0.05). This information, such as the p-values, should be included, possibly alongside Figure 1d for clarification. Including statistical values might help readers better understand these distinctions.
2. The authors mentioned in lines #89–91 regarding Snrpn that “This putative asynchrony between the parental chromosomes correlated with faithful maintenance of methylation levels at the domain’s gDMR (Fig. 1f).” However, it is unclear what message the authors intended to convey here. Does this imply that minor asynchrony regions, such as Igf2-H19, Peg3/Zim, or RasGrf1, do not show a similar correlation?
3. The authors should elaborate on “Capture Repli-seq” in more detail, including the experimental scheme. Is it similar to the Repli-Capture-seq method described by Rivera-Mulia et al. in Blood Advances (2019)?
4. The section titled “lncRNA expression controls maternal-specific early replication at the Dlk1-Dio3 domain” is somewhat confusing. The authors show that Zfp57^{-/-};Meg3-pro^{-/-} mESCs no longer express Meg3 (Fig. 4c). They then state that there are no RT changes at the Meg3 region but significant changes upstream and downstream of Meg3 (Fig. 4h). The authors conclude in line 194 that “DMR methylation, and not the expression of the Meg3 polycistron, controls RT.” However, this statement seems contradictory to both the section title and the later conclusion in lines 200–202, which states, “The Meg3 polycistron expression therefore appears to control the early RT on the maternal chromosome at this 'Begain-Dlk1 region.'” This section needs to be rewritten for better clarity and interpretation. In addition, the authors should highlight the regions

where changes do or do not occur to help readers understand the data more easily. Furthermore, although the authors mention that no RT changes occur near the Meg3 gene (Fig. 4h), I believe there may be subtle changes that could be observed if they carefully compares Zfp57^{-/-} mESCs (Fig. 4g) with Zfp57^{-/-};Meg3-pro^{-/-} mESCs (Fig. 4g). Further quantification of these potential changes should be done for more accurate interpretation.

5. The conclusion regarding Meg3 expression and RT changes at the Begain-Dlk1 region is based on experiments involving mutant backgrounds and DNA sequence manipulation (such as the deletion of the promoter), which are known to sometimes affect RT. For example, "early replication control elements" (ERCEs) have been shown to cause RT changes, as demonstrated by Sima et al. (Cell, 2019). Although the authors deleted only ~200 bp near Meg3 gene, this could possibly still have unintended effects on RT. An alternative method to inhibit transcription, such as α -Amanitin treatment, could help validate this conclusion without altering the genomic DNA. Furthermore, using Ori^{-/-} and intron1^{-/-} mESCs, in which Meg3 expression is reduced, would provide additional control groups to validate these results. Since Ori^{-/-} mESCs exhibit approximately half the expression level of Meg3, if transcription truly affects early RT, the authors might be able to observe corresponding changes in RT that are dependent on the level of transcription (i.e., dosage-dependent)

6. To examine the effect of gene knockout or CRISPR-mediated genomic deletion on gene expression, the authors performed qPCR analysis only for the Meg3 gene. Since there are several genes within the asynchronously replicating region of the Dlk1-Dio3 domain, it would be more appropriate to perform RNA-seq analysis to examine all possible transcripts within the domain.

7. While RT generally correlates well with A/B compartments, this is not always the case. It would be informative to include A/B compartment profiles alongside RT and TAD profiles for better comparison. For example, upon mESC differentiation into NPCs, the A/B compartments of the paternal Dlk1-Dio3 domain may rearrange in parallel with changes in RT.

8. In lines 235–236, the authors state, "The maternal chromosome adopted a TAD organization that was different from that on the paternal chromosome." However, the subtitle claims, "Imprinted asynchronous replication seems not connected to parental allele-specific TAD organization." This is again somewhat confusing. In general, while TAD structures differ between the two alleles, they do not always align precisely with RT profiles. So, it may be premature to conclude that RT is unrelated to allele-specific TAD structure. The differences in TAD structure could potentially contribute to RT, possibly in connection with A/B compartments, as discussed in comment #7.

9. In lines 245-247, it is stated that the upstream and downstream boundaries of the replication timing domain overlap with CTCF binding sites, but it is not clear how they determined these replication timing boundaries, nor where they are shown in Figures C and D (please indicate the position of the boundaries with arrows).

10. The title of this manuscript does not accurately represent the experimental results and is misleading, as the involvement of lncRNAs in the regulation of replication timing has only been suggested in a very limited number of imprinted gene regions.

Minor comments:

1. The explanation for Fig. 2b is a bit confusing. Adding "top panel" and "bottom panel" to the text might help clarify which figures to refer to. Also, whether the PCR assays are allelic-PCR, qPCR, or Sanger sequencing should be clearly stated, as it is sometimes unclear which method is being used.

2. The sentence, "This asynchrony included ~450-kb of sequences upstream of Dlk1, the Dlk1 gene itself, the entire 220-kb ncRNA polycistron, and Rtl1 (Fig. 2a, e, f)," is unclear in terms of where to focus. The lettering is also very small for the gene names. Using highlights or arrows would help the readers identify the relevant regions.

3. Typo in Extended Data Fig. 1d: "RT profiles at example imprinted domains and imprinted gene loci in AK2 and PR3 mESCs. Vertical rectangles indicate he extent of the imprinted loci." ("he" should be "the.")

4. The authors did not discuss why deleting the replication origin and intron-1 affected Meg3 expression.

5. In Fig. 5, the meaning of the dashed and solid lines should be clearly explained.

6. In Fig. 6b, there is no scale bar, just a number?

7. In the Fig. 6b legend, "Nestin" and "Tubulin-B3" should not be italicized.

Reviewer #2

(Remarks to the Author)

Imprinted genes have proven to be an excellent model system for studying epigenetic regulation of gene expression and

could therefore be a similar paradigm in studying epigenetic contribution to other DNA functions such as replication. It has long been known (early 1990's) that some imprinted genes replicate asynchronously – indeed Howard Cedar was the first to show that fluorescent in-situ hybridisation with probes for the critical PWS region showed split signals. To date we still don't know the extent of this phenomenon or the underlying causal mechanisms. Here Imaizumi and colleagues from the Feil lab in Montpellier show that asynchronous allelic replication differences is quite subtle at most imprinted loci in mouse embryonic stem cells, similar to previous genome-wide reports (Rivera Mulia et al Genome research 2018) and in contrast to Edwards et al Parent-of-origin-specific DNA replication timing is confined to large imprinted regions - ScienceDirect). The exception is the Dlk1-Dio3 locus which shows a large difference in replication timing. The strength of this study is the use of reciprocal hybrid mESCs as well parthenogenic and androgenic derived ESCs, and congenic (CRISPR derived) lines in which DNA methylation is perturbed and/or expression of the lncRNA Meg3 congenic line is disrupted. These tools enable dissection of the contribution of imprinted DNA methylation and lncRNA expression to replication timing.

Data are presented well and the conclusions are supported

1. Intuitively, based on what is well known about X-inactivation and overall high content of DNA methylation in late replicating regions, we'd expect that for imprinted genes the methylated allele would be the later replicating allele. From the heatmap in fig 1d, it seems that most of the imprinted loci studied were in the early replicating phases and not different between the mat and pat allele (using monoparental androgenic/parthenogenetic embryos). Exceptions include Herc3, and Snrp loci which replicate late, For the small number of imprinted gene loci (n=4) that do replicate asynchronously, three seem to have the methylated allele replicating late (Dlk1-Dio3, H19lgf2, Peg3) and one locus Rassgrf1 where it is the opposite – i.e the unmethylated allele that replicates late. In extended table 1 it is interesting that some loci have changes in replication timing after differentiation to neural cells – This extended table should be in the main text. Given that the methylation patterns don't change upon differentiation, replication timing should also be correlated to expression changes during differentiation.

2. Do the differences between cell cycle length of ESC and that of differentiated cells have an effect on early and late replication ?

3. Also there should be some mention in the discussion that different imprinted genes are regulated through different mechanisms and that there is no ubiquitous canonical mechanism at play at all the ICRs (some are boundaries, others are promoters etc).

4. In most cases gene expression is completed before replication starts but there are exceptions. Would it be possible to examine when transcription of the lncRNA (Meg3) other imprinted genes are completed in the cell cycle? This is further important because actively transcribed genes are early replicating, long transcripts could result in replicative stress.

5. More discussion is required highlighting the sensitivity of the assay compared to other methods for genome-wide replication timing

6. The putative replication origins should be mapped for all the imprinted loci. It would be interesting to see how many replication origins fall within DMRs.

7. Does the chromatin conformation change in the Znf57-/- cells? Although the Meg3 was only transiently deleted, would we predict that the HiC conformation would change when the lncRNA is no longer transcribed – if so then the altered replication pattern may well correlate with the Hi-C TAD structure.

8. Rather than correlating TAD structure, it would be useful to highlight A and B compartments in relation to replication domains.

9. It would be nice to include a model figure bringing the data together and explaining the contribution of Meg3 transcription and DMRs to asynchronous replication before and after differentiation.

Fonts on figures need to be increased

Reviewer #3

(Remarks to the Author)

Key results

The authors describe asynchronous replication timing at two parentally imprinted loci, Dlk1-Dio3 and Snrpn, and related the asynchrony to differential allelic parental DNA methylation and expression of lncRNA polycistrons. The authors conclude that differential DNA methylation functionally controls imprinted asynchronous replication.

Significance & Validity

The authors and others have established that expression, DNA methylation, histone modifications and 3D organization are associated with allelic asynchronous replication timing (Du et al. 2021; Rivera-Mulia et al. 2018; Kota et al. 2014; Edwards et al. 2024; Bickmore and Carothers 1995). Some of these studies including the authors have already shown that manipulation of DNA methylation, histone marks or expression can affect allelic asynchronous replication timing, including imprinted regions (Du et al. 2021; Kota et al. 2014; Bickmore and Carothers 1995). The main advance in this study is that the authors use genetic manipulation models to connect allelic replication timing to DNA methylation, 3D organization and expression of the same allele. However, I have major concerns over some of their methods and the conclusions the authors derive from the results. These are further described below.

Major comments

1. Result section “DNA methylation is essential for imprinted asynchronous replication.”

a. The authors themselves show that DNA methylation is not essential at all imprinted loci, and in their cell systems, methylation is specifically relevant to Dlk1-Dio3 and Snrpn. A more nuanced writing of their results is warranted.

b. The authors use two clonally derived cell lines, Sh-1 and Zfp57-/- lines, which have either methylation on both alleles or no methylation at all, to address the importance of DNA methylation for allelic replication timing at the Dlk1-Dio3 locus. As these lines are clonally derived, it is hard to conclude that there is a direct causative effect between DNA methylation and

allelic replication timing, even though there definitely appears to be an association. In my view this could be improved through use of a more transient disruption of DNA methylation in their BJ and JB cell systems. One possibility may be to assay allelic replication timing after removal of ascorbic acid from the culture media. The effect of ascorbic acid on DNA methylation in mESCs is reversible, whereby 5mC gradually returns over 9 days (Blaschke et al. 2013). Even better, a timecourse here could definitively prove causality if allelic asynchrony of RT gradually increases/decreases with gradual change in DNA methylation after ascorbic acid removal. An alternative is 5-azacytidine, however, 5-aza may have confounding side effects on mESC identity, I am not so familiar with the literature here.

2. Is it possible to add allelic PCR for replication timing or Capture Repli-seq for the Snrpn locus? As the Snrpn locus is the only other example of what the authors describe, it would be useful to see the Snrpn replication timing and changes in more detail.

3. Result section "lncRNA expression controls maternal-specific early replication at the Dlk1-Dio3 domain."

a. "A transient CRISPR-Cas9 approach was used to delete a 198-bp region directly upstream of the Meg3 TSS, to generate Zfp57^{-/-};Meg3-pro^{-/-} mESCs (Fig. 4a,b)." line 186. The authors use CRISPR to induce deletions in the Dlk1-Dio3 locus at the TSS of Meg3 and then examine replication timing and 3D organization. The purpose of the deletion is to change expression of the lncRNA. However, it is known that replication timing and 3D organization is highly influenced by promoter and enhancer regions due to binding of transcription factors like CTCF (Sima et al. 2019; Bonev and Cavalli 2016). Deleting the promoter region of Meg3 could be unintentionally affecting TF/CTCF sites that directly affecting RT and 3D organization regardless of expression changes (Chakraborty et al. 2025). The authors thus cannot conclude that any consequences are due to only expression loss.

b. Have the authors investigated potential transcription factor binding motifs in the deleted region? Even if CTCF itself is not disrupted, other TFs can affect where and how CTCF binds, potentially which CTCFs interact with each other which then can determine 3D organization.

c. The text says the Meg3 TSS deletion is 198bp (line 186), however, Fig 4b says 259bp. What is the discrepancy here?

4. Result section "Imprinted asynchronous replication seems not connected to parental allele-specific TAD organization." In general, I am concerned about the authors interpretation of their results.

a. "Similarly, neither on the maternal nor on the paternal chromosome, we could observe apparent co-localization between CTCF binding and RT domain boundaries (Fig. 5a-c)." line 241. There is a non-allelic CTCF site at 109.75 (Fig. 5a) that does appear to be at the rightmost boundary of both the maternal and paternal TADS. However, this does not appear in either the maternal or paternal CTCF tracks. Could this be missing due to lack of SNPs in this region to distinguish reads allelically?

b. "Here, the upstream boundary of the early-replicating RT domain over the IG-DMR coincided with a CTCF binding site that demarcates the boundary of the Dlk1-Meg3 sub-TAD on the maternal chromosome in WT cells (Fig. 5b,d)." line 245.

i. Can the authors please show the location of the IG-DMR in Fig 5?

ii. As far as I can tell, the upstream boundary of the early replicating RT domain over IG-DMR is right between 109.25 and 109.50, whereas the CTCF binding site at the Dlk3-Meg3 sub-TAD is around 105.55. My interpretation of Fig 5 is that the CTCF site is sitting in the middle of the early RT domain in the double mutant, rather than at the boundary.

iii. Similarly - "This notion is supported by our cellular model - with erased imprint and abolished Meg3 polycistron expression, where RT boundaries now align with CTCF binding sites found in WT cells (Fig. 5, arrows)." Line 360. Visually, the RT boundary does not align with the Meg3-DMR CTCF site indicated with an arrow in Fig 5.

5. "We conclude that imprinted asynchronous replication, only observable at two imprinted domains in stem cells in this study, attenuates or resolves during neural differentiation and these RT changes are not associated with concordant changes in TAD organization." Line 303.

a. RT domain boundaries and TAD boundaries do not always correspond (Pope et al. 2014) i.e. one RT domain can contain several TAD domains. RT and 3D is more linked at the A/B compartment level, however, this may be difficult to reconstruct using Capture Hi-C, as it generally requires information from long distance and chromosome wide interactions.

b. Papers that do look at the overlap between RT and TAD or A/B compartment boundaries tend to give a range to what is considered an overlap between boundaries i.e. depending on what is acceptable in the literature, it may be that the early RT domain in NPCs overlaps the larger paternal TAD. Then the interpretation would be that differentiated RT becomes concordant with paternal TAD organization.

Minor Comments

- "In agreement with our earlier study 51, NPC differentiation did not alter the methylation levels at ICRs (Fig. 6c)." Is the methylation maintained allelically too?

- More clarity for the BJ and JB lines would be welcomed, perhaps a schematic that shows which is maternal and which is paternal in each cell line and what the heterozygous SNPs are.

c. Bickmore, W. A., and A. D. Carothers. 1995. 'Factors affecting the timing and imprinting of replication on a mammalian chromosome', *J Cell Sci*, 108 (Pt 8): 2801-9.

d. Blaschke, K., K. T. Ebata, M. M. Karimi, J. A. Zepeda-Martinez, P. Goyal, S. Mahapatra, A. Tam, D. J. Laird, M. Hirst, A. Rao, M. C. Lorincz, and M. Ramalho-Santos. 2013. 'Vitamin C induces Tet-dependent DNA demethylation and a blastocyst-like state in ES cells', *Nature*, 500: 222-6.

e. Bonev, B., and G. Cavalli. 2016. 'Organization and function of the 3D genome', *Nat Rev Genet*, 17: 772.

f. Chakraborty, S., N. Wenzlitschke, M. J. Anderson, A. Eraso, M. Baudic, J. J. Thompson, A. A. Evans, L. M. Shatford-Adams, R. Chari, P. Awasthi, R. K. Dale, M. Lewandoski, T. J. Petros, and P. P. Rocha. 2025. 'Deletion of a single CTCF motif at the boundary of a chromatin domain with three FGF genes disrupts gene expression and embryonic development', *Dev Cell*.

g. Du, Q., G. C. Smith, P. L. Luu, J. M. Ferguson, N. J. Armstrong, C. E. Caldon, E. M. Campbell, S. S. Nair, E. Zotenko, C. M. Gould, M. Buckley, K. M. Chia, N. Portman, E. Lim, D. Kaczorowski, C. L. Chan, K. Barton, I. W. Deveson, M. A. Smith, J. E.

Powell, K. Skvortsova, C. Stirzaker, J. Achinger-Kawecka, and S. J. Clark. 2021. 'DNA methylation is required to maintain both DNA replication timing precision and 3D genome organization integrity', *Cell Rep*, 36: 109722.

h. Edwards, M. M., N. Wang, I. Sagi, S. Kinreich, N. Benvenisty, J. Gerhardt, D. Egli, and A. Koren. 2024. 'Parent-of-origin-specific DNA replication timing is confined to large imprinted regions', *Cell Rep*, 43: 114700.

i. Kota, S. K., D. Lleres, T. Bouschet, R. Hirasawa, A. Marchand, C. Begon-Pescia, I. Sanli, P. Arnaud, L. Journot, M. Girardot, and R. Feil. 2014. 'ICR noncoding RNA expression controls imprinting and DNA replication at the Dlk1-Dio3 domain', *Dev Cell*, 31: 19-33.

j. Pope, B. D., T. Ryba, V. Dileep, F. Yue, W. Wu, O. Denas, D. L. Vera, Y. Wang, R. S. Hansen, T. K. Canfield, R. E. Thurman, Y. Cheng, G. Gulsoy, J. H. Dennis, M. P. Snyder, J. A. Stamatoyannopoulos, J. Taylor, R. C. Hardison, T. Kahveci, B. Ren, and D. M. Gilbert. 2014. 'Topologically associating domains are stable units of replication-timing regulation', *Nature*, 515: 402-5.

k. Rivera-Mulia, J. C., A. Dimond, D. Vera, C. Trevilla-Garcia, T. Sasaki, J. Zimmerman, C. Dupont, J. Gribnau, P. Fraser, and D. M. Gilbert. 2018. 'Allele-specific control of replication timing and genome organization during development', *Genome Res*, 28: 800-11.

l. Sima, J., A. Chakraborty, V. Dileep, M. Michalski, K. N. Klein, N. P. Holcomb, J. L. Turner, M. T. Paulsen, J. C. Rivera-Mulia, C. Trevilla-Garcia, D. A. Bartlett, P. A. Zhao, B. K. Washburn, E. P. Nora, K. Kraft, S. Mundlos, B. G. Bruneau, M. Ljungman, P. Fraser, F. Ay, and D. M. Gilbert. 2019. 'Identifying cis Elements for Spatiotemporal Control of Mammalian DNA Replication', *Cell*, 176: 816-30 e18.

Version 1:

Reviewer comments:

Reviewer #1

(Remarks to the Author)

The authors have addressed most of my questions and the whole manuscript and data interpretation have improved substantially. However, I still have some comments that would like the authors to clarify in the text, without the need of further revision on my side.

In line 106, the authors state, "This correlated with faithful maintenance of DNA methylation levels at the domain's DMRs (Fig. 1f)." I do not understand what exactly "faithful maintenance" means. The same phrase is used elsewhere (e.g., line 113). Please clarify this phrase, or consider deleting it.

The authors generated a new deletion mutant ESC line in which the Meg lncRNA production is abolished without affecting its transcriptional initiation or the production of other parts of the polycistron. Although the results of this new experiment are informative, the mutant carries a large deletion and therefore does not fully resolve my concern that DNA sequence manipulation could have unintended effects on replication timing. I still believe that the drug-mediated inhibition of transcription would be worth trying, as it represents the simplest way to exclude this possibility (if complete transcriptional inhibition is toxic to cells, the authors might consider testing partial inhibition to examine its impact on replication timing). If this experiment is not feasible, the authors should provide a discussion of the possible unintended effects on replication timing.

In line 287, the authors state, "At the Dlk1-Dio3 domain, asynchronous replication thus associates with distinct allelic A/B compartments." I think this is an overstatement, as I do not see distinct differences in A/B compartments at this domain between androgenetic and pathenogenetic mESCs. While I do see some differences in this domain, a similar degree of differences is also observed at the 108-109 Mb region, where no asynchronous replication occurs.

Reviewer #2

(Remarks to the Author)

I am very happy with the author's response to my comments. This is an exciting advance in the imprinting field

Reviewer #3

(Remarks to the Author)

With this revision, Imaizumi and colleagues have improved the overall interpretation and presentation of their results and more precisely present their findings. Reflecting the changes the authors made to their title, this manuscript presents an in-depth study of how differential DNA methylation and/or lncRNA expression impacts asynchronous replication timing at the imprinted Dlk1-Dio3 locus. As said before, the major advance of this study over others are the use of genetic manipulation models to examine RT asynchrony via Capture Repli-seq, particularly for investigating the role of DNA methylation. However, these models and the additional CRISPR deletion models fall short in definitively showing that lncRNA expression independent of DNA methylation has a role in asynchronous replication at the Dlk1-Dio3 locus. Inclusion of dynamic intervention models for DNA methylation and/or expression would have greatly increased the novelty and mechanistic significance of this study. Major comments below.

1. Reviewer comment 3a: "...Deleting the promoter region of Meg3 could be unintentionally affecting TF/CTCF sites that directly affecting RT and 3D organisation regardless of expression changes (Chakraborty et al. 2025). The authors thus

cannot conclude that any consequences are due to only expression loss.”

a. Authors’ response to point 3a: “On a first note, we previously characterized CTCF binding and its importance for 3D structure at this domain, describing how CTCF does not bind the promoter of Meg3 but rather allele-specifically at its first intron (Liéres et al, 2019).”

b. It is not only CTCF, but other transcription factor binding sites may be deleted in the Meg3-pro^{-/-} model. For example, Sima et. al. 2019 found that deletion of cis-acting ERCE elements affected RT, but depletion of CTCF protein had no effect on RT, implying that something other than CTCF binding at these ERCEs is controlling RT.

2. Authors’ response to point 3a: “To address if, in the Zfp57^{-/-};Meg3-pro^{-/-} mESCs, the observed consequences for replication timing were due solely to the expression loss, we generated a new recombinant mESC line in which we removed the Meg3 lncRNA gene from the 220-kb ncRNA polycistron of the imprinted Dlk1-Dio3 domain, while keeping DNA methylation, CTCF binding at the first intron, and gene expression across the remainder of the polycistron unaltered. In this new mESC line, in which the lncRNA Meg3 was no longer expressed (shown in new figure 5), we find delayed replication timing across (non-overlapping) parts of the imprinted domain on the maternal chromosome. On the maternal allele, RT in this new cellular model mirrored the one in Zfp57^{-/-};Meg3-pro^{-/-} mESCs. This key finding excludes the possibility that the effect seen in the Zfp57^{-/-};Meg3-pro^{-/-} mESCs solely resulted from the deletion of (RT-regulatory elements within) the Meg3-promoter. Instead, it consolidates the suggested involvement of the lncRNA Meg3 (rather than that of the expression of the entire ncRNA polycistron). (see also our answer to point 5 of Reviewer 1).”

a. It was the concern of both Reviewer 1 (point 5) and this reviewer, that genetic disruption could confound the authors’ conclusion that it is expression loss that affects allelic RT. The ideal solution to this would have been to use a method that only disrupts the transcriptome and not the genome, as suggested by Reviewer 1 (also see comment below). If anything, an even larger genetic disruption, as in the Meg3-C1 deletion, would amplify the issue. For example, can the authors be sure that the Meg3-C1 deletion is not removing other enhancer-like elements that contribute to the organisation of the locus? Either with or without CTCF?

b. Comparing maternal RT in Fig. 4h to Fig. 5e, the Meg3-C1 deletion does not appear to exactly mirror Zfp57^{-/-};Meg3-pro^{-/-} mESCs. Specifically, the region between 109.75 and 110 Mb is early in Zfp57^{-/-};Meg3-pro^{-/-} mESCs and mid in Meg3-C1 deletion cell line. The authors point this out themselves in the figures. It is therefore possible that different genetic elements may be affected between the two deletions, leading to different results. In addition, the fact that the BJ and JB lines show differences in paternal allele RT, which the authors attribute to genetic differences between the strains (Line 147), further supports that genetic differences and thus also genetic disruptions can affect RT.

c. A cleaner solution could be to use siRNAs targeted to the Meg3 lncRNA. siRNAs would only disrupt RNA in a targeted manner, and not DNA. This method would deplete expression of all 3, Meg3-Rian-Mirg, as they are one transcript, so one would not be able to delineate between the 3, but differentiating between the 3 is not the focus of this manuscript anyway. Additionally, by removing just the RNA this would address the hypothesis the authors raise in the discussion, that the lncRNA itself could be contributing to the 3D organisation and sequestering replication factors to ensure early replication.

i. As a note, using siRNAs would not be too long a period to detect replication timing change in mESCs (siRNAs are often used for a max 7 days in rapidly dividing cells). The authors wrote the following in response to comment 5 from reviewer 1: “ α -Amanitin treatments of cells take >24h to achieve full transcriptional repression...which would be too long for RT studies on the rapidly dividing mESCs”. A length of time as short as 24hrs or even up to 7 days should not be a concern to the authors as the authors main cell models, Sh-1 and Zfp57^{-/-}, took days to weeks to generate, as they required colony selection. Similarly, the CRISPR models the authors generate also took more than 12 days according to their methods, “After 10–12 days of culture, colonies were transferred to 6-well plates to derive clonal mESC lines.”

3. Author response to point 5a: “As detailed in our response to reviewer 1 (point 7), we instead performed genome-wide Hi-C on parthenogenetic and androgenetic mESCs and determined the A/B compartment. Please, see our answer to reviewer 1 (point 7) for interpretation. These new data are presented in Figure 6a.”

a. It’s nice to see that the RT boundaries do seem to match A/B compartment boundaries, if not TAD boundaries. It is also interesting that RT and Hi-C differ for the parthenogenic line i.e. in the parthenogenic mESCs, the locus is clearly early replicating, but only very slightly A-compartment in the Hi-C. This suggests that this locus in general is not distinctly compartmentalized i.e. not ‘locked in’ to A or B, and this could be a reason why it can change RT readily between maternal and parental chromosomes, and upon genetic manipulation. It could be interesting to see if other imprinted loci are similar, with or without RT asynchrony.

b. It would be great to see a more zoomed in view of Fig. 6b. Does the A/B-compartment boundaries in the parthenogenic and androgenic lines help in any way to interpret the genomic location differences in RT changes between disrupting DNA methylation and disrupting the Meg3 lncRNA?

c. It is unfortunate that further analysis of RT and TADs from the capture Hi-C did not bring more clarity on this phenomenon. But have the authors looked at the genome wide Hi-C TADs? Do the TADs from genome wide Hi-C done in the parthenogenic and androgenic lines make sense with the RT boundaries? One benefit of looking at the genome-wide HiC is perhaps the capture Hi-C region missed regions of the genome where larger TAD structures that may interact with the Dlk1-dio3 locus.

4. Authors’ response: “In our mESC cultures, we systematically add ascorbic acid to the medium to prevent acquisition of abnormal DNA methylation at the IG-DMR differentially methylated region of the Dlk1-Dio3 domain. Since our cells were derived in the presence of multiple inhibitors and have been grown in the presence of ascorbic acid, they are considered naïve. To prevent aberrant methylation at the Dlk1-Dio3 domain we have to put the ascorbic acid, and longer incubations with ascorbic acid are not expected to lead to reductions in DNA methylation under these conditions....Once again, we

already have genetic systems that alter the DNA methylation at the imprinted DMRs, which in our view is a better approach.”

a. The suggestion was specifically to remove ascorbic acid, rather than add more. This was what was asked: “One possibility may be to assay allelic replication timing after removal of ascorbic acid from the culture media. The effect of ascorbic acid on DNA methylation in mESCs is reversible, whereby 5mC gradually returns over 9 days (Blaschke et al. 2013).” It would follow that the maternal allele could gain methylation at the IG-DMR over a course of 9 days. Using this as a timecourse would validate that DNA methylation at this DMR as an effect on allelic RT, using a transient and non-genome disruptive method. In other words, how would this the acquisition of abnormal DNA methylation upon removal of ascorbic acid impact the maternal locus?

b. It has already been shown that ascorbic acid can affect the methylation of the Dlk1-Dio3 locus (Stadtfeld et. al. 2012). The benefit of this is that unlike the Sh-1 cell line, the cells do not need to undergo clonal selection. This contrasts with the genetic models where cause and consequence are separated for much longer periods of time. Often weeks were required to generate the genetic models used in this study. This dynamic approach would better support the mechanism between DNA methylation and RT, as the cause and consequence can be followed at much closer timepoints and in sequence. There may be other reasons that removing ascorbic acid may not work, but the main point is that a transient DNA methylation disruption system would more clearly show a direct mechanistic relationship between allelic DNA methylation and allelic RT.

c. This is of lower priority than the lncRNA comment above.

Version 2:

Reviewer comments:

Reviewer #3

(Remarks to the Author)

I thank the authors for their comments. They have addressed my concerns.

REVIEWER COMMENTS

Reviewer #1 (Remarks to the Author):

This study by Imaizumi et al. utilizes uni-parental and hybrid mouse ES cell lines to examine the asynchronous replication timing of imprinted gene domains using genome-wide replication timing assays and Capture Repli-seq. The authors successfully delineate detailed asynchronous replication patterns across the *Dlk1-Dio3* and *Snrpn* domains, where lncRNA expression is regulated by DMR DNA methylation. The observed parent-specific replication timing was affected either by the DNA methylation status of the DMR or by a small deletion that abolished lncRNA expression within the domain. Overall, the experiments are well-designed and executed, and the obtained data are solid. However, the authors fail to provide mechanistic insights into why the local methylation status at the DMR has a domain-wide impact on replication timing. Additionally, it remains entirely unclear how lncRNA expression is involved in the early replication of the *Begain-Dlk1* region, given that the lncRNA coding region does not overlap with the *Begain-Dlk1* region. This study is interesting in that it identifies factors (DMR DNA methylation and lncRNA expression) that affect parent-specific replication timing patterns. However, an important question remains regarding how domain-wide asynchronous replication timing is established. Therefore, this study is considered preliminary.

We thank this reviewer for his/her careful assessment, and for commenting on the quality of the study-design and the obtained data. We agree that the study shows involvement of differential DNA methylation and of ncRNA expression, but that it does not provide a complete mechanistic explanation of how precisely these mediate the differential replication timing. Although not fully characterized, we nonetheless think our work identifies new regulatory mechanisms and provides accompanying mechanistic advances, whereas we think that further detailed characterization should be addressed in follow-up studies. Nevertheless, as a first step towards providing further insights, we generated a new recombinant mESC line in which we removed the *Meg3* lncRNA gene from the 220-kb ncRNA polycistron of the imprinted *Dlk1-Dio3* domain, while keeping DNA methylation and gene expression across the remainder of the polycistron unaltered. In this new mESC line, in which the lncRNA *Meg3* gene is no longer expressed (shown in new figure 5), we find delayed replication timing across (non-overlapping) parts of the imprinted domain on the maternal chromosome (see our below answer to point 5). This key finding consolidates the suggested involvement of the lncRNA *Meg3* (following the observed effects of the expression of the entire ncRNA polycistron).

Major comments:

1. The author mentioned that *Igf2-H19*, *Peg3/Zim*, and *RasGrf1* show potential minor asynchrony (line #85), while *Snrpn* is considered to exhibit putative asynchrony. However, in Extended Data Figure 1 and Figure 1g, the levels of asynchrony between *Igf2-H19*, *Peg3/Zim*, *RasGrf1*, and *Snrpn* appear quite similar. It is unclear how the author determined which regions were categorized as asynchronous or synchronous. This is because the author did not clearly explain how many imprinted regions were identified as asynchronous or which specific regions met the criteria for asynchrony (as mentioned in Figure 1 legend, asynchronous replication was determined using START-R software with the Mean method and a p-value threshold of 0.05). This information, such as the p-values, should be included, possibly alongside Figure 1d for clarification. Including statistical values might help readers better understand these distinctions.

We thank the reviewer for these helpful suggestions. In the Table (included in Figure 1) we now added p-value thresholds, which were obtained using the START-R software using the Mean method, as described before (Hadjadj, D. *et al.* 2020, *NAR Genom Bioinform.*, 19, 2: lqaa045). In this comparison between androgenetic and parthenogenetic mESCs, only for the *Dlk1-Dio3* and *Snrpn* imprinted domains p-values below 0.05 were obtained. We nevertheless considered in more detail several other imprinted loci, which showed higher p-values. The replication profiles at these loci, in the androgenetic versus parthenogenetic mESC, are shown in Extended data Fig. 1e. Based on this information, we further verified four of these loci in the hybrid mESCs, because they appeared to possibly have minor differences between the parental chromosomes. At these loci, allelic PCR assays were designed (using SNPs) at regions close to the germline DMRs, and we assessed in the early and late fractions whether there could be allelic biases. These were not detected, and our data therefore do not provide evidence for asynchronous replication at these additional loci. This is shown in Extended data Fig. 1 g, and described in the manuscript on page 8.

2. The authors mentioned in lines #89–91 regarding *Snrpn* that “This putative asynchrony between the parental chromosomes correlated with faithful maintenance of methylation levels at the domain’s gDMR (Fig. 1f).” However, it is unclear what message the authors intended to convey here. Does this imply that minor asynchrony regions, such as *Igf2-H19*, *Peg3/Zim*, or *RasGrf1*, do not show a similar correlation?

We thank the reviewer for pointing out this potential confusion in our writing. The reason we had mentioned about the faithful maintenance of differential DNA methylation at the *Snrpn* and *Dlk1-Dio3* domains at this point in the manuscript, is that this could be one of the features important for the replication asynchrony observed between the androgenetic and parthenogenetic mESCs. At several of the ‘minor, putative asynchrony’ domains, including at the *Igf2-H19* and *Kcnq1* domains there was also faithful maintenance of allelic DNA methylation at the ICR (Fig. 1f, Extended data Fig. 1g, Extended data fig. 2c). This showed that the imprinted DNA methylation does not systematically generate replication asynchrony at imprinted loci, and is insufficient on its own.

In the manuscript we rewrote the corresponding lines as follows (page 6): *‘At the imprinted Snrpn locus, a tendency towards late RT was apparent in both mono-parental lines. However, this gene replicated significantly earlier in the androgenetic than in the parthenogenetic mESCs (Fig. 1d,g). This correlated with faithful maintenance of DNA methylation at the domain’s gDMR (Fig. 1f), which suggested that differential methylation could potentially be involved.’*

3. The authors should elaborate on “Capture Repli-seq” in more detail, including the experimental scheme. Is it similar to the Repli-Capture-seq method described by Rivera-Mulia *et al.* in *Blood Advances* (2019)?

We thank the reviewer for directing us towards the study by Rivera-Mulia *et al.*, which we had missed. The Repli-Capture-seq methodology employed by Rivera-Mulia *et al.* (*Blood Advances* 2019) is different from the one we used: it captures representative 250-bp regions from each 10-kb window throughout the genome. In our approach, we captured specific genomic regions at ultra-high

resolution using many probes across these regions. This allowed us to generate high-resolution data (kb resolution) for these regions.

In the Results, page 7, we added a sentence to explain this to the readers: *‘Similar to the recent genome-wide Repli-Capture-Seq assay⁴², our approach relies on pools of custom oligonucleotides, but these were designed tiles, and thereby enriched at high resolution specific chromosomal regions.*

4. The section titled “LncRNA expression controls maternal-specific early replication at the Dlk1-Dio3 domain” is somewhat confusing. The authors show that *Zfp57*^{-/-};*Meg3*-*pro*^{-/-} mESCs no longer express *Meg3* (Fig. 4c). They then state that there are no RT changes at the *Meg3* region but significant changes upstream and downstream of *Meg3* (Fig. 4h). The authors conclude in line 194 that “DMR methylation, and not the expression of the *Meg3* polycistron, controls RT.” However, this statement seems contradictory to both the section title and the later conclusion in lines 200–202, which states, “The *Meg3* polycistron expression therefore appears to control the early RT on the maternal chromosome at this ‘*Begain-Dlk1* region.’” This section needs to be rewritten for better clarity and interpretation. In addition, the authors should highlight the regions where changes do or do not occur to help readers understand the data more easily. Furthermore, although the authors mention that no RT changes occur near the *Meg3* gene (Fig. 4h), I believe there may be subtle changes that could be observed if they carefully compares *Zfp57*^{-/-} mESCs (Fig. 4g) with *Zfp57*^{-/-};*Meg3*-*pro*^{-/-} mESCs (Fig. 4g). Further quantification of these potential changes should be done for more accurate interpretation.

To no longer create confusion, we describe the PCR-based data shown in Fig. 4g, but do not at this point in the text draw conclusions about mechanisms. Conclusions are now based primarily on the Capture Repli-seq data, shown in Fig. 4h, and follow on from the description of these data. Particularly, we write at the end of this section: *‘Combined, the above data evoke a model in which the paternal DMR controls differential RT in its surroundings on both the parental chromosomes, whereas the maternal expression of the large ncRNA polycistron in turn controls in cis early replication across proximal and distal parts of the asynchronous replication zone. We conclude that the asynchronous replication zone primarily depends on the action of ZFP57 (‘maintenance of allelic DMR methylation’) around Meg3 and on the expression of the Meg3 polycistron in the ‘Begain-Dlk1 region’ (strong effect) and the ‘Mirg region’ (weaker effect).*

On advice of the reviewer, we highlight the RT differences between *Zfp57*^{-/-} mESCs and *Zfp57*^{-/-};*Meg3*-*pro*^{-/-} mESCs (Fig. 4h). In panel h, we now indicate the boundaries of the proximal and distal regions controlled by the polycistron expression.

5. The conclusion regarding *Meg3* expression and RT changes at the *Begain-Dlk1* region is based on experiments involving mutant backgrounds and DNA sequence manipulation (such as the deletion of the promoter), which are known to sometimes affect RT. For example, “early replication control elements” (ERCEs) have been shown to cause RT changes, as demonstrated by Sima et al. (Cell, 2019). Although the authors deleted only ~200 bp near *Meg3* gene, this could possibly still have unintended effects on RT. An alternative method to inhibit transcription, such as α -Amanitin treatment, could help validate this conclusion without altering the genomic DNA. Furthermore, using *Ori*^{-/-} and *intron1*^{-/-} mESCs, in which *Meg3* expression is reduced, would provide additional control groups to validate these results. Since *Ori*^{-/-} mESCs exhibit approximately half the expression level of *Meg3*, if transcription

truly affects early RT, the authors might be able to observe corresponding changes in RT that are dependent on the level of transcription (i.e., dosage-dependent)

We thank the reviewer for these comments and agree that, based on the provided data, we could not exclude the possibility that the promoter deletion had unintended effects on RT. Since in the different mutant lines, RNA expression levels were altered across the entire ncRNA polycistron, it also remained difficult to be sure about the specific involvement of Meg3 lncRNA. The suggested inhibition of transcription with agents such as α -Amanitin could be an approach, but has its known complications as well. α -Amanitin treatments of cells take >24h to achieve full transcriptional repression (Nguyen, VT et al. 1996; *Nucl. Acids Res.* 24, 2924-2929; Letschert K et al. 2006; *Toxicological Sci.* 91, 140-149; Fenouil R et al. 2012; *Gen. Res.* 22, 2399-2408), which would be too long for RT studies on the rapidly dividing mESCs. In addition, genome-wide reductions in gene transcription generated by inhibitors of Pol-II might have indirect effects on RT, such as altered expression of proteins that control replication.

As an alternative approach, we chose to specifically target the production of the Meg lncRNA, without interfering with its transcription initiation or the production of other parts of the polycistron. For this, we generated a new mESC line in which we removed the *Meg3* lncRNA sequence from the maternally expressed 220-kb ncRNA polycistron of the imprinted *Dlk1-Dio3* domain, while keeping DNA methylation and gene expression across the remainder of the polycistron unaltered. In this new mESC line (called Δ *Meg3*-c1), in which Meg3 lncRNA was no longer expressed (shown in the new figure 5), there was delayed replication timing across (non-overlapping) parts of the domain on the maternal chromosome (see our below answer to point 5). This new finding, presented in Figure 5, consolidates the suggested involvement of the lncRNA Meg3 (as opposed to that of the expression of the entire ncRNA polycistron). It also consolidates the notion that altered replication timing in *Zfp57*^{-/-}; *Meg3*-*pro*^{-/-} does not solely result from unintended effects after Meg3-promoter manipulation.

In the manuscript, these new data are described on page 11.

6. To examine the effect of gene knockout or CRISPR-mediated genomic deletion on gene expression, the authors performed qPCR analysis only for the Meg3 gene. Since there are several genes within the asynchronously replicating region of the *Dlk1-Dio3* domain, it would be more appropriate to perform RNA-seq analysis to examine all possible transcripts within the domain.

To address this important point, as suggested, we performed RNA-seq on total RNAs and analysed the data in a parental chromosome-specific manner by filtering reads across SNPs between the C57BL/6J and JF1 genotypes. Expression levels were analysed as well, using total reads, with correction against genome-wide RNA expression. This is described in the Methods section. In figure 4, panel f shows the allelic RNA-seq profiles in WT (BJ cells), *Zfp57*^{-/-} mESCs and *Zfp57*^{-/-}; *Meg3*-*pro*^{-/-} mESCs across the polycistron (*Dlk1* expression is not detected in mESCs and therefore not shown).

These new RNA-seq data are described in the manuscript on page 10. They confirm that the *Zfp57* deletion had given rise to equal biallelic expression across the entire Meg3 polycistron, whereas the *Meg3* promoter deletion on the *Zfp57*^{-/-} genetic background led to an absence of reads across the entire polycistron, on both the parental chromosomes. RNA-seq was performed also on a novel mESC line in which we specifically deleted the *Meg3* gene. The obtained allelic profiles show absence of reads

across the deleted region only, whereas transcription is maintained over the downstream parts of the polycistron (data included in new Fig 5d).

7. While RT generally correlates well with A/B compartments, this is not always the case. It would be informative to include A/B compartment profiles alongside RT and TAD profiles for better comparison. For example, upon mESC differentiation into NPCs, the A/B compartments of the paternal *Dlk1-Dio3* domain may rearrange in parallel with changes in RT.

We agree that in published genome-wide studies, RT correlated well with A/B compartments at a lower resolution on a genome-wide scale. Determination of A/B compartments, identified from the eigenvector in the Hi-C matrix, is generally performed at resolutions below those used for TAD analysis (i.e., at 100kb-1Mb resolution, through the reduction of matrix resolution by grouping multiple windows together). For our study, where we created allele-specific Capture Hi-C maps at 10-kb resolution over an ~1 Mb window, calling an informative eigenvector is therefore not possible (as also commented by reviewer 3). To obtain genome-wide allele-specific Hi-C maps in hybrid cells at sufficient resolution would require a sequencing throughput (and associated cost) that we think goes beyond what can be expected for a question that is not directly at the core of this study.

Instead, as questions about A/B compartments were raised by all three reviewers, we identified A and B compartments in a parental chromosome specific manner by performing genome-wide Hi-C on parthenogenetic and androgenetic mESCs (described in the Methods).

In a new first panel added to Figure 6 we show mouse chromosome 12 as an example. We achieved an almost complete visible overlap between the A/B compartments and the (array-hybridization) RT profiles. Early replication domains co-localised with the A compartment, and late replication domains with the B compartment. Although less pronounced, a similar trend was observed at the *Dlk1-Dio3* imprinted domain, with only a weak link with the A compartment in the parthenogenetic mESCs (early replication), and a minor trends towards the B compartment in the androgenetic mESCs (late replication). This is described on page 13 of the manuscript.

At the imprinted *Snrpn* domain, which is rich in repeat elements, the obtained data were insufficiently dense for exploring the A/B compartment. Unfortunately, a similar approach could not be used for the NPCs either, since androgenetic and the parthenogenetic mESCs cannot be differentiated into neural cells (because of their aberrant patterns of imprinted gene expression).

8. In lines 235–236, the authors state, “The maternal chromosome adopted a TAD organization that was different from that on the paternal chromosome.” However, the subtitle claims, “Imprinted asynchronous replication seems not connected to parental allele-specific TAD organization.” This is again somewhat confusing. In general, while TAD structures differ between the two alleles, they do not always align precisely with RT profiles. So, it may be premature to conclude that RT is unrelated to allele-specific TAD structure. The differences in TAD structure could potentially contribute to RT, possibly in connection with A/B compartments, as discussed in comment #7.

We agree with the reviewer that we had not been clear enough in our conclusions, and that despite the non-overlap of the asynchronous RT domain with the parental chromosome-specific TADs, these

could potentially still contribute to the asynchronous replication. In this section, we now also added capture-HiC studies on the *Zfp57*^{-/-} mESCs –in which both the parental chromosomes showed early replication timing across the entire *Dlk1-Dio3* domain.

To better reflect the results in this section we changed its title into: *“Imprinted asynchronous replication and parental allele-specific TADs show distinct boundaries”*

9. In lines 245-247, it is stated that the upstream and downstream boundaries of the replication timing domain overlap with CTCF binding sites, but it is not clear how they determined these replication timing boundaries, nor where they are shown in Figures C and D (please indicate the position of the boundaries with arrows).

We thank the reviewer for raising this point. Replication domains are often defined as zones of constant replication timing that are terminated by Timing Transition Regions (TTR) corresponding to slopes of progressive change in replication timing (from early to later). The edges of TTR, that had been used to demarcate RT domain boundaries, have been shown to align with TAD border, with a certain degree of variability (+/- 150 kb; Pope *et al.* 2014). Assigning TRR boundaries for individual loci remains complex, particularly from 1-2kb resolutive capture Repli-seq (see also our answer to reviewer 3, point 5b). We’ve now clearly indicated the domain of asynchronous replication, based on visual inspection, and its boundaries (see Figs. 2e,f).

10. The title of this manuscript does not accurately represent the experimental results and is misleading, as the involvement of lncRNAs in the regulation of replication timing has only been suggested in a very limited number of imprinted gene regions.

We agree. To render it more specific, we changed the title as follows: *“DNA methylation and lncRNA control asynchronous DNA replication at specific imprinted gene domains”*

Minor comments:

1. The explanation for Fig. 2b is a bit confusing. Adding "top panel" and "bottom panel" to the text might help clarify which figures to refer to. Also, whether the PCR assays are allelic-PCR, qPCR, or Sanger sequencing should be clearly stated, as it is sometimes unclear which method is being used.

We thank the reviewer for these helpful suggestions. We changed the description of the *Meg3* data as follows: *“Both allele-specific PCR (top panels) and Sanger sequencing of PCR products (lower panels) showed early replication on the maternal and mostly late replication on the paternal chromosome, both in BJ and JB mESCs (Fig. 2c,d).”*

For *Dlk1*, we changed the text as follows: *“...we also performed amplification across an SNP at this developmental gene (Fig. 2b). In both JB and BJ cells, Sanger sequencing of PCR products showed early replication on the maternal and late replication on the paternal chromosome (Fig. 2c,d).”*

2. The sentence, “This asynchrony included ~450-kb of sequences upstream of *Dlk1*, the *Dlk1* gene itself, the entire 220-kb ncRNA polycistron, and *Rtl1* (Fig. 2a, e, f),” is unclear in terms of where to

focus. The lettering is also very small for the gene names. Using highlights or arrows would help the readers identify the relevant regions.

We simplified this sentence as follows: *'This large, continuous region of asynchrony includes the Dlk1 gene and the entire 220-kb ncRNA polycistron (Fig. 2a,e,f).* In Figs. 2e,f, we now indicate the extent of the asynchronous replication zone. In the figure panels, gene names are now put in larger fonts.

3. Typo in Extended Data Fig. 1d: "RT profiles at example imprinted domains and imprinted gene loci in AK2 and PR3 mESCs. Vertical rectangles indicate he extent of the imprinted loci." ("he" should be "the.").

Corrected.

4. The authors did not discuss why deleting the replication origin and intron-1 affected Meg3 expression.

In an earlier study (Kota *et al.*, *Dev. Cell* 2014), we reported that intron-1 deletion leads to full methylation at the *Meg3*-DMR, while keeping the methylation status of the IG-DMR unchanged. We verified that this was the case also in the batch of cells used in the current study, and show that also in these cells, the *Meg3*-DMR is hypermethylated (Extended data Fig. 6e). We now mention in the manuscript that this explains the loss of expression of the polycistron (*Meg3*, *Rian* and *Mirg*). Intron-1 is part of the *Meg3*-DMR and, presumably, key transcription factors are recruited to this region on the unmethylated maternal allele only.

The deleted region adjacent to the IG-DMR, at its extremity closest to *Meg3*, had been described by Aronson *et al.* as a 'transcriptional regulation element' (TRE) (Aronson BE *et al.* *Dev. Cell* 2021). Based on detailed studies on iPSC-derived neural cells, these authors showed that this region has enhancer activity. Our replication origin deletion overlaps the TRE and, not surprisingly, we find reduced expression of *Meg3*, *Rian* and *Mirg*. In the text, page 12, we now comment on the deleted region having a reported transcriptional regulatory role, and cite Aronson *et al.* 2021.

5. In Fig. 5, the meaning of the dashed and solid lines should be clearly explained.

Done.

6. In Fig. 6b, there is no scale bar, just a number?

We added the scale bar

7. In the Fig. 6b legend, "Nestin" and "Tubulin-B3" should not be italicized.

Corrected.

Reviewer #2 (Remarks to the Author):

Imprinted genes have proven to be an excellent model system for studying epigenetic regulation of gene expression and could therefore be a similar paradigm in studying epigenetic contribution to other DNA functions such as replication.

It has long been known (early 1990's) that some imprinted genes replicate asynchronously – indeed Howard Cedar was the first to show that fluorescent in-situ hybridisation with probes for the critical PWS region showed split signals. To date we still don't know the extent of this phenomenon or the underlying causal mechanisms. Here Imaizumi and colleagues from the Feil lab in Montpellier show that asynchronous allelic replication differences is quite subtle at most imprinted loci in mouse embryonic stem cells, similar to previous genome-wide reports (Rivera Mulia at al Genome research 2018) and in contrast to Edwards et al Parent-of-origin-specific DNA replication timing is confined to large imprinted regions - ScienceDirect). The exception is the Dlk1-Dio3 locus which shows a large difference in replication timing. The strength of this study is the use of reciprocal hybrid mESCs as well parthenogenic and androgenic derived ESCs, and congenic (CRISPR derived) lines in which DNA methylation is perturbed and/or expression of the lncRNA Meg3 congenic line is disrupted. These tools enable dissection of the contribution of imprinted DNA methylation and lncRNA expression to replication timing.

Data are presented well and the conclusions are supported.

We thank the Reviewer for these constructive comments.

1. Intuitively, based on what is well known about X-inactivation and overall high content of DNA methylation in late replicating regions, we'd expect that for imprinted genes the methylated allele would be the later replicating allele. From the heatmap in fig 1d, it seems that most of the imprinted loci studied were in the early replicating phases and not different between the mat and pat allele (using monoparental androgenic/parthenogenetic embryos). Exceptions include Herc3, and Snrp loci which replicate late, For the small number of imprinted gene loci (n=4) that do replicate asynchronously, three seem to have the methylated allele replicating late (Dlk1-Dio3, H19Igf2, Peg3) and one locus Rassgrf1 where it is the opposite – i.e the unmethylated allele that replicates late. In extended table 1 it is interesting that some loci have changes in replication timing after differentiation to neural cells – - This extended table should be in the main text. Given that the methylation patterns don't change upon differentiation, replication timing should also be correlated to expression changes during differentiation.

We thank the reviewer for raising these points. As concerns DNA methylation, we point out that at imprinted gene loci, the differential DNA methylation between the maternal and the paternal genome is not found across entire domains, but is confined to the differentially methylated regions (DMRs), including the essential germ-line transmitted 'Imprinting Control Regions' (one for each imprinted domain). These germ-line DMRs, listed in the Table in Figure 1d, are only several kilobases in size, and most of them correspond to CpG islands. At imprinted gene domains, therefore, there is not a situation that one parental chromosome is methylated and the other not. Instead, the differential methylation is confined to the small fraction of the domain that is covered by these DMRs (see Fig. 2a). Moreover, both methylated and unmethylated DMRs can be present on a single parental allele, with the opposite

pattern present on the other parental allele. This situation seems different from that on the X chromosome, or in late replicating domains, which the reviewer mentioned about. This may explain why at most imprinted gene domains we observed that the methylated DMR allele was not associated with late replication.

We now added two sentences to the Introduction to explain imprinted DMRs (on page 3): *'Imprinted gene domains provide an attractive RT paradigm. They are all controlled by parental DNA methylation imprints at differentially methylated regions (DMRs), which are up to several kilobases in size, but there is no ubiquitous canonical mechanism through which these mediate the allelic expression of imprinted genes.'* This also addresses the below point 3.

To be more conclusive about the mentioned imprinted gene loci for which the genome-wide array-based replication data suggested a possible asynchrony between the parental genomes, we included p-value thresholds into the Table shown in Figure 1d. This showed significance for the *Dlk1-Dio3* and *Snrpn* domains only. For a number of additional loci at which there could nevertheless be asynchrony, we performed locus-specific allelic RT studies in the hybrid mESCs (qPCR). See our answer to Reviewer 1 for additional detail (point 1).

Both at the *Dlk1-Dio3* and *Snrpn* domains, the loss of the asynchronous replication upon differentiation into NPCs is associated with changes in gene expression. At *Dlk1-Dio3*, we previously showed that the *Dlk1* gene is activated on the paternal chromosome predominantly during neural differentiation (Kota et al. *Dev. Cell* 2014). Also the *Snrpn* polycistron becomes activated during neural differentiation, on the paternal chromosome only. We do not exclude a role for these developmental gene expression patterns in the RT changes that occur during neural differentiation.

We refer to this in the Discussion (page 19): *'On the paternal chromosome, the DMRs are methylated, and the associated late replication is less well understood, but could be linked to the low levels of gene expression on this parental chromosome, at least in mESCs³⁴. Irrespective of the precise downstream mechanism(s), our studies provide the first demonstration that DNA methylation functionally controls imprinted asynchronous replication.'*

2. Do the differences between cell cycle length of ESC and that of differentiated cells have an effect on early and late replication ?

This is an interesting idea that we cannot address, unfortunately, and that goes beyond the scope of the current study. We note, however, that also random asynchronous replication has been reported to resolve upon differentiation of ESCs into NPCs (Rivera-Mulia et al., 2018). The question remains as to whether developmental differences in RT indicate that a particular length of cell cycle would be required, or alternatively, whether specific pluripotency-associated factors could be involved in the asynchronous replication in mESCs.

3. Also there should be some mention in the discussion that different imprinted genes are regulated through different mechanisms and that there is no ubiquitous canonical mechanism at play at all the ICRs (some are boundaries, others are promoters etc).

We agree with the reviewer that this is important to point out, particularly because asynchronous replication is detected at two of the imprinted domains only. In the Introduction we added a line to clarify this point: *'Imprinted gene domains provide an attractive RT paradigm. They are all controlled by parental DNA methylation imprints at differentially methylated regions (DMRs), which are up to several kilobases in size, but there is no ubiquitous canonical mechanism through which DMRs mediate the allelic expression of imprinted genes.'*

4. In most cases gene expression is completed before replication starts but there are exceptions. Would it be possible to examine when transcription of the lncRNA (Meg3) other imprinted genes are completed in the cell cycle? This is further important because actively transcribed genes are early replicating, long transcripts could result in replicative stress.

This is an interesting idea, which is however complicated to address. Rather than focusing on the role of transcription at the large Meg3-Rian-Mirg polycistron, we chose for the revision to specifically explore the role of the lncRNA Meg3. We generated a new mESC line in which we removed the *Meg3* gene from the 220-kb ncRNA polycistron of the imprinted *Dlk1-Dio3* domain, while keeping DNA methylation and gene expression across the remainder of the polycistron unaltered. In this new mESC line, in which Meg3 lncRNA was no longer expressed (shown in the new figure 5), we find delayed replication across (non-overlapping) parts of the domain on the maternal chromosome (see our below answer to point 5). This finding consolidates the suggested involvement of the lncRNA Meg3. For more detail, also see our answer to Reviewer 1, point 5.

5. More discussion is required highlighting the sensitivity of the assay compared to other methods for genome-wide replication timing.

To evaluate the accuracy of our Repli-chip method, in an earlier study we compared it to the previously established six-fraction Repli-seq protocol (Hansen *et al.*, 2010, PNAS USA 107, 139-144), and we published the outcome of this comparison (Hadjadj D. *et al.*, 2020, NAR Genom Bioinform. 2(2):lqaa045). In this comparison both approaches were applied to human K562 cells and mouse ES46C cells. The Repli-seq data, comprising six cell cycle fractions (G1, S1, S2, S3, S4, and G2) were retrieved from the UCSC Genome Browser. We concluded that our two-fraction Repli-chip protocol, when optimised with appropriate cell-sorting parameters, produces results comparable and with similar sensitivity to those obtained with the six-fraction Repli-seq method. Currently, it is also more cost-effective and time-efficient.

Concerning the comparison of cell lines, we used the Mean method option in the START-R software. Briefly, the Mean method compares the means of log ratio intensities (Early/Late) obtained in two different experiments. Mean is calculated for a sliding window of 30 successive probes corresponding to a 300 kb genomic domain, which is consistent with the size of replication-timing domains already described (Rivera-Mulia and Gilbert, 2016). The overlapping parameter, defining the number of probes overlapping successive windows was initially set to 15. After the calculation of nominal and adjusted p-values (t-tests for mean comparison), we choose the p-value thresholds (here pval=0.05) to distinguish significant differences between two conditions. The p-value adjustment method is chosen among a list of classical procedures (such as Bonferroni in our case). Building on this, we measured the

same RT differences in our studies using the Repli-Chip, Repli-qPCR (using SNPs) and Capture Repli-seq methods. Finally, as a reminder, the chips used have an average of one probe every 13 kb, with a maximum resolution of one probe every 250 bp at certain loci. In the case of the *Dlk1-Dio3* region, the resolution is approximately one probe every 500 bp on average.

In the Results, on page 5, we put a sentence to mention about our earlier comparison between RT methods: '*Previously, we reported that this approach gives similar results as multi-fraction Rep-seq*³⁴.'

6. The putative replication origins should be mapped for all the imprinted loci. It would be interesting to see how many replication origins fall within DMRs.

We thank the reviewer for raising this interesting point. We analysed the genome-wide data from the Cayrou *et al.* study to see whether other imprinted DMRs (listed in Fig. 1d) could comprise replication origins in mESCs as well. This analyses showed that at six other imprinted loci (*Igf2r*, *Grb10*, *Rasgrf1*, *Cdh15*, *Peg1* and *Gnas*) the germline DMRs comprised mutiple replication origins. As opposed to the *Dlk1-Dio3* domain, however, at these loci there was no replication asynchrony. In general, our study does not link the presence of replication origins at imprinted DMRs to asynchronous RT.

In the Results section, on page 12, we put a line to emphasise this point : *At several other imprinted loci (Igf2r, Grb10, Rasgrf1, Cdh15, Peg1 and Gnas), the germline DMRs (Fig. 1d) comprise multiple replication origins in mESCs (data from 52). As opposed to the Dlk1-Dio3 domain, however, at these loci there was no replication asynchrony. In general, our study does not link putative replication origin activity at imprinted DMRs to asynchronous RT.*

7. Does the chromatin conformation change in the *Znf57^{-/-}* cells? Although the *Meg3* was only transiently deleted, would we predict that the HiC conformation would change when the lncRNA is no longer transcribed – if so then the altered replication pattern may well correlate with the Hi-C TAD structure.

We thank the reviewer for raising this question. We have now generated Hi-C in the *Zfp57^{-/-}* cells (Figure 6d), which shows that upon loss of imprinting the paternal chromosome adopted the 3D conformation of the wild-type maternal chromosome. This did not result into reinforced overlap between the boundaries of replication timing domains and those of 3D-chromatin domains.

8. Rather than correlating TAD structure, it would be useful to highlight A and B compartments in relation to replication domains.

We thank the reviewer for raising this point, which was also raised by the two other reviewers. As also mentioned by reviewer 3, and as detailed in our response to reviewer 1 (point 7), our Capture Hi-C does not cover sufficiently large genomic intervals to reliably call A/B compartments in hybrid cells. Instead, we therefore performed genome-wide Hi-C on parthenogenetic and androgenetic mESCs and determined the A/B compartment. Please, see our answer to reviewer 1 (point 7) for interpretation. These new data are presented in Figure 6a.

9. It would be nice to include a model figure bringing the data together and explaining the contribution of Meg3 transcription and DMRs to asynchronous replication before and after differentiation.

We added such a figure (Fig. 8), to present a summary model that shows the allelic RT status in embryonic stem and differentiated cells, and explains the contributions of differential DNA methylation and Meg3 lncRNA (in ESCs).

Fonts on figures need to be increased

In the figures, we increased the font sizes where needed.

Reviewer #3 (Remarks to the Author):

Key results

The authors describe asynchronous replication timing at two parentally imprinted loci, *Dlk1-Dio3* and *Snrpn*, and related the asynchrony to differential allelic parental DNA methylation and expression of lncRNA polycistrons. The authors conclude that differential DNA methylation functionally controls imprinted asynchronous replication.

Significance & Validity

The authors and others have established that expression, DNA methylation, histone modifications and 3D organization are associated with allelic asynchronous replication timing (Du et al. 2021; Rivera-Mulia et al. 2018; Kota et al. 2014; Edwards et al. 2024; Bickmore and Carothers 1995). Some of these studies including the authors have already shown that manipulation of DNA methylation, histone marks or expression can affect allelic asynchronous replication timing, including imprinted regions (Du et al. 2021; Kota et al. 2014; Bickmore and Carothers 1995). The main advance in this study is that the authors use genetic manipulation models to connect allelic replication timing to DNA methylation, 3D organization and expression of the same allele. However, I have major concerns over the some of their methods and the conclusions the authors derive from the results. These are further described below.

We thank the reviewer for these constructive comments, and for mentioning about these published studies on asynchronous replication timing, most of which we cited already.

Despite the interest of the mentioned studies, we argue that as concerns imprinted gene domains in the mouse, our study is novel. The genome-wide replication study by Edwards et al. 2024 concerned human ESCs, and we cite this study in the Discussion to highlight similarities with our mouse findings. Kota et al. 2014 and Bickmore and Carothers 1995 used DNA FISH to explore specific imprinted and non-imprinted loci, but this technology does not directly assess replication timing (it explores the frequency of chromatid separation: formation of 'doublets'). The closest to our work is the Rivera-Muller et al. 2018 study, in which the authors performed Rep-seq on mouse hybrid ESCs of a different genotype than ours. We already cited this high-resolution allele-specific study in the Introduction, and mentioned that it did not report replication asynchrony based on the parental origin of the allele (i.e., at imprinted gene domains), but highlights the importance of genetic polymorphisms. The apparent lack of asynchronous replication at imprinted domains in this study may relate to the cell lines or passages studied, or to the way these authors cultured their cells (prolonged ESC culture can lead to loss of imprinting in certain media).

We also thank the reviewer for mentioning about the Du et al. 2021 study by the laboratory of Susan Clark. We cite this interesting study in the Discussion. It explored genome-wide 3D genome organization, gene expression and replication timing in a human cell line (HCT116). In their detailed multi-omics comparisons, the authors reported that the majority of the identified loci with allelic replication timing in individual cells did not correspond to known imprinted gene loci. Particularly, of the 35 genes found to display allele-specific replication timing (Table S1 in Du et al. 2021), only one is

imprinted (DGCR6) and another one might be imprinted (PRIM2). In addition, this study was performed on cancer cells, and concerned humans, whereas our study explored RT in the mouse, in non-transformed cell lines.

To us, the main interest of the Du et al. 2021 study is that it explored what happens with RT domains when levels of DNA methylation are globally reduced (i.e. in HCT116 cells knock-out for both DNMT1 and DNMT3B). Interestingly, they found that this led to a reduced precision of DNA replication timing, but led to marked RT alterations at some loci only.

We comment on this interesting finding in the Discussion, on pages 19-20: *‘In human colorectal cancer cells double-KO for DNMT1 and DNMT3A—which induced strongly reduced DNA methylation levels—there was a reduced precision of RT, with more variability between cells, but only 3% of gene loci showed shifts in replication timing* ⁶⁹.

Major comments

1. Result section “DNA methylation is essential for imprinted asynchronous replication.”

a. The authors themselves show that DNA methylation is not essential at all imprinted loci, and in their cell systems, methylation is specifically relevant to Dlk1-Dio3 and Snrpn. A more nuanced writing of their results is warranted.

We agree with the Reviewer and changed the title of this section to *‘Differential DNA methylation is essential for asynchronous replication at the Dlk1-Dio3 and Snrpn domains’*

b. The authors use two clonally derived cell lines, Sh-1 and Zfp57^{-/-} lines, which have either methylation on both alleles or no methylation at all, to address the importance of DNA methylation for allelic replication timing at the Dlk1-Dio3 locus. As these lines are clonally derived, it is hard to conclude that there is a direct causative effect between DNA methylation and allelic replication timing, even though there definitely appears to be an association. In my view this could be improved through use of a more transient disruption of DNA methylation in their BJ and JB cell systems. One possibility may be to assay allelic replication timing after removal of ascorbic acid from the culture media. The effect of ascorbic acid on DNA methylation in mESCs is reversible, whereby 5mC gradually returns over 9 days (Blaschke et al. 2013). Even better, a timecourse here could definitively prove causality if allelic asynchrony of RT gradually increases/decreases with gradual change in DNA methylation after ascorbic acid removal. An alternative is 5-azacytidine, however, 5-aza may have confounding side effects on mESC identity, I am not so familiar with the literature here.

We thank the reviewer for his/her thoughts and for making these suggestions. As explained to reviewer 2 (answer to point 1), at imprinted gene domains the differential DNA methylation does not cover large regions, but is confined to the DMRs, which cover several kilobases only. At the *Dlk1-Dio3* domain, the DMRs are already aberrant in their methylation, with full methylation in the Sh-1 mESCs, and a complete loss of methylation in the *Zfp57^{-/-}* mESCs. In our mESC cultures, we systematically add ascorbic acid to the medium to prevent acquisition of abnormal DNA methylation at the IG-DMR differentially methylated region of the *Dlk1-Dio3* domain.

Since our cells were derived in the presence of multiple inhibitors and have been grown in the presence of ascorbic acid, they are considered naïve. To prevent aberrant methylation at the *Dlk1-Dio3* domain we have to put the ascorbic acid, and longer incubations with ascorbic acid are not expected to lead to reductions in DNA methylation under these conditions.

As mentioned by the reviewer, incubation with 5-azacytidine is another way to induce global hypomethylation, but its effects are slow and pleiotropic, with extensive cell death and major changes in gene expression. Observed effects on replication timing will be difficult to explain and are likely indirect. Once again, we already have genetic systems that alter the DNA methylation at the imprinted DMRs, which in our view is a better approach.

2. Is it possible to add allelic PCR for replication timing or Capture Repli-seq for the *Snrpn* locus? As the *Snrpn* locus is the only other example of what the authors describe, it would be useful to see the *Snrpn* replication timing and changes in more detail.

We agree it would have been nice to provide a detailed view on the RT at the *Snrpn* domain in the hybrid mESCs. As mentioned in the text, the density of repeated sequences in this imprinted domain, covering the large majority of the genomic interval, prevented us from using Capture Repli-seq. As an alternative approach, we now developed an allelic PCR amplification assay, at the paternally expressed *Ipw* gene. We find that at this central gene ('region 3'), there is no asynchronous DNA replication (Extended data Fig. 1f). This suggests that the asynchronous replication does not comprise the entire imprinted domain, which agrees with the array-based data on the mono-parental mESCs with clear asynchrony along *Snrpn*, but not at proximal and distal regions.

3. Result section "LncRNA expression controls maternal-specific early replication at the *Dlk1-Dio3* domain."

a. "A transient CRISPR-Cas9 approach was used to delete a 198-bp region directly upstream of the *Meg3* TSS, to generate *Zfp57*^{-/-};*Meg3*-*pro*^{-/-} mESCs (Fig. 4a,b)." line 186. The authors use CRISPR to induce deletions in the *Dlk1-Dio3* locus at the TSS of *Meg3* and then examine replication timing and 3D organisation. The purpose of the deletion is to change expression of the lncRNA. However, it is known that replication timing and 3D organization is highly influenced by promoter and enhancer regions due to binding of transcription factors like CTCF (Sima et al. 2019; Bonev and Cavalli 2016). Deleting the promoter region of *Meg3* could be unintentionally affecting TF/CTCF sites that directly affecting RT and 3D organisation regardless of expression changes (Chakraborty et al. 2025). The authors thus cannot conclude that any consequences are due to only expression loss.

We agree with the reviewer's comment that deletions may remove sequences with unrecognized regulatory functions. On a first note, we previously characterized CTCF binding and its importance for 3D structure at this domain, describing how CTCF does not bind the promoter of *Meg3* but rather allele-specifically at its first intron (Llères et al, 2019).

To address if, in the *Zfp57*^{-/-};*Meg3*-*pro*^{-/-} mESCs, the observed consequences for replication timing were due solely to the expression loss, we generated a new recombinant mESC line in which we

removed the *Meg3* lncRNA gene from the 220-kb ncRNA polycistron of the imprinted *Dlk1-Dio3* domain, while keeping DNA methylation, CTCF binding at the first intron, and gene expression across the remainder of the polycistron unaltered. In this new mESC line, in which the lncRNA *Meg3* was no longer expressed (shown in new figure 5), we find delayed replication timing across (non-overlapping) parts of the imprinted domain on the maternal chromosome. On the maternal allele, RT in this new cellular model mirrored the one in *Zfp57^{-/-};Meg3^{-pro}^{-/-}* mESCs. This key finding excludes the possibility that the effect seen in the *Zfp57^{-/-};Meg3^{-pro}^{-/-}* mESCs solely resulted from the deletion of (RT-regulatory elements within) the *Meg3*-promoter. Instead, it consolidates the suggested involvement of the lncRNA *Meg3* (rather than that of the expression of the entire ncRNA polycistron). (see also our answer to point 5 of Reviewer 1).

b. Have the authors investigated potential transcription factor binding motifs in the deleted region? Even if CTCF itself is not disrupted, other TFs can affect where and how CTCF binds, potentially which CTCFs interact with each other which then can determine 3D organisation.

Besides the binding of CTCF, assessed by ChIP-seq, we did not further explore transcription factor binding. We presume that in the mESC lines in which one or both DMRs of the *Dlk1-Dio3* domain were methylated, with no longer promoter activity at the ncRNA polycistron, there could be reduced recruitment of TFs to the *Meg3* DMR. In the novel mESC line in which we deleted *Meg3* exon 2-10 (line d*Meg3*-C1), importantly, there was no change in the methylation status and the *Meg3* promoter remained active (i.e., the remainder of the polycistron remained expressed normally). This allowed us to draw conclusions specifically about *Meg3* lncRNA independently from possible effects of altered TF binding.

c. The text says the *Meg3* TSS deletion is 198bp (line 186), however, Fig 4b says 259bp. What is the discrepancy here?

We corrected this mistake. The size of this deletion is 198 bp.

4. Result section “Imprinted asynchronous replication seems not connected to parental allele-specific TAD organization.” In general, I am concerned about the authors interpretation of their results.

We agree with the reviewer that the title of this section did not interpret well the results. We changed the section title as follows: ‘*Imprinted asynchronous replication and parental chromosome-specific TADs show distinct boundaries*’.

a. “Similarly, neither on the maternal nor on the paternal chromosome, we could observe apparent co-localization between CTCF binding and RT domain boundaries (Fig. 5a-c).” line 241. There is a non-allelic CTCF site at 109.75 (Fig. 5a) that does appear to be at the rightmost boundary of both the maternal and paternal TADS. However, this does not appear in either the maternal or paternal CTCF tracks. Could this be missing due to lack of SNPs in this region to distinguish reads allelically?

The reviewer is right: a CTCF binding site delineates the right-most boundary of the maternal-specific TAD, but it cannot be assigned to either parental chromosome due to the lack of SNPs in this region. We have now clarified this in the revised manuscript. We also now provided an easier visualization of non-assigned (non-allelic) and allelic CTCF ChIP-seq results.

b. “Here, the upstream boundary of the early-replicating RT domain over the IG-DMR coincided with a CTCF binding site that demarcates the boundary of the *Dlk1-Meg3* sub-TAD on the maternal chromosome in WT cells (Fig. 5b,d).” line 245.

i. Can the authors please indicate the localisation of the IG-DMR in Fig 5?

Done.

ii. As far as I can tell, the upstream boundary of the early replicating RT domain over IG-DMR is right between 109.25 and 109.50, whereas the CTCF binding site at the *Dlk3-Meg3* sub-TAD is around 105.55. My interpretation of Fig 5 is that the CTCF site is sitting in the middle of the early RT domain in the double mutant, rather than at the boundary.

In our previous study in which we explored sub-TAD organization at the *Dlk1-Dio3* domain (Llères et al, 2019), we identified the maternal *Dlk1-Dio3* sub-TAD and its CTCF binding sites. This sub-TAD is demarcated by bi-allelic CTCF binding at the upstream (5'-) boundary (located between 109.25 and 109.50) and maternal-specific CTCF binding at the downstream (3'-) boundary (located around 105.55).

To better highlight the relevant CTCF binding sites, including the upstream biallelic CTCF sites that coincide with the early-replicating RT domain, we have added additional shading in Figure 6 (former Figure 5).

On page 13, we added a line to clarify this point: “Specifically, a maternal-specific sub-domain forms with boundaries, as defined by local minima in the insulation score⁵⁸, that aligned with a CTCF binding site at the *Meg3-DMR* on one side, and on the other side with a CTCF binding site downstream of *Mirg* that could not be assigned to either parental chromosome due to the lack of discriminative SNP (Fig. 6c).

iii. Similarly - “This notion is supported by our cellular model with erased imprint and abolished *Meg3* polycistron expression, where RT boundaries now align with CTCF binding sites found in WT cells (Fig. 5, arrows).” Line 360. Visually, the RT boundary does not align with the *Meg3-DMR* CTCF site indicated with an arrow in Fig 5.

As mentioned above, we added additional shading in the new Fig. 6 and named certain CTCF binding sites to better highlight their relevance.

5. “We conclude that imprinted asynchronous replication, only observable at two imprinted domains in stem cells in this study, attenuates or resolves during neural differentiation and these RT changes are not associated with concordant changes in TAD organisation.” Line 303.

a. RT domain boundaries and TAD boundaries do not always correspond (Pope et al. 2014) i.e. one RT domain can contain several TAD domains. RT and 3D is more linked at the A/B compartment level,

however, this may be difficult to reconstruct using Capture Hi-C, as it generally requires information from long distance and chromosome wide interactions.

We thank the reviewer for this comment, including the notion that Capture Hi-C is not well suited to determine A/B compartments. Questions about A/B compartments were also raised by the other reviewers, we therefore tried to follow up on them. As detailed in our response to reviewer 1 (point 7), we instead performed genome-wide Hi-C on parthenogenetic and androgenetic mESCs and determined the A/B compartment. Please, see our answer to reviewer 1 (point 7) for interpretation. These new data are presented in Figure 6a.

b. Papers that do look at the overlap between RT and TAD or A/B compartment boundaries tend to give a range to what is considered an overlap between boundaries i.e. depending on what is acceptable in the literature, it may be that the early RT domain in NPCs overlaps the larger paternal TAD. Then the interpretation would be that differentiated RT becomes concordant with paternal TAD organisation.

We thank the reviewer for this valuable comment, which made us revisit the manuscript by Pope et al. (Nature 2014). Upon re-reading, we realized an imprecision in our interpretation of the overlap between TAD boundaries and RT domains/ Timing Transition Regions (TTRs). Rather than overlap between TAD boundaries and centre of TTRs, Pope et al reports that, on average, TAD boundaries coincide with the boundaries of TTRs, particularly at the side of the early RT domains. This agrees with the further correct remark by the reviewer that a certain degree of variability is observed for individual boundaries (in a range of approximately +/- 150 kb for the large majority of boundaries?). We have added the following changes to the manuscript:

- In the Introduction we've added a sentence to provide a more precise description of overlap between TAD boundaries and TTRs (together with a mention of A/B compartments as well).

- In the Results section, we have modified our interpretations based on the overlap between TAD and TRR boundaries, while incorporating the notion of the range of TRR boundaries as well. This with the complications that (i) assigning TRR boundaries for individual loci remains approximate (particularly from Capture Repli-seq) and (ii) the sub-TADs that are present within the domain are in the order of a 150-300 kb in size, which is the same order of size as the reported variability for TRR boundaries. Modifications to the interpretation have been incorporated throughout the Results section. For our studies in mESCs, we find that distances between TRR and TAD boundaries are generally >100 kb. Although within the accepted range, this is a large distance relative to the size of the sub-TADs and early RT domains at the locus. In NPCs, the reviewer is correct that the paternal sub-TAD and RT domain (up to the TRRs on both sides) show an improved overlap, relative to the acceptable range. We discuss the possibility that RT and TAD organization on the paternal chromosome may have become concordant, but we remain cautious due to the reported variation at individual loci.

Minor Comments

- "In agreement with our earlier study 51, NPC differentiation did not alter the methylation levels at ICRs (Fig. 6c)." Is the methylation maintained allelically too?

Yes, it does. Also in the mESC-derived NPCs, the DNA methylation is present on the paternal chromosome only. This is shown in Extended data Fig. 7d.

- More clarity for the BJ and JB lines would be welcomed, perhaps a schematic that shows which is maternal and which is paternal in each cell line and what the heterozygous SNPs are.

In the BJ lines, the maternal genome is C57BL/6J and the paternal genome has the JF1 genotype. Conversely, in the JB lines, the maternal genome is JF1 and the paternal genome C57BL/6J. This we now explain in the beginning of the Results section: *'Particularly, we studied line 'BJ', in which the maternal genome is C57BL/6J and the paternal genome JF1, and line 'JB', which has the reciprocal genotype'*.

To provide clarity about the heterozygous SNPs used for allelic discrimination, between the C57BL6/6 genome and the JF1 genome, in the Supplementary Information we systematically indicate for each SNP used for PCR, which nucleotide is diagnostic for C57BL6/J, and which one for JF1 genome.

Reviewer #1 (Remarks to the Author):

The authors have addressed most of my questions and the whole manuscript and data interpretation have improved substantially. However, I still have some comments that would like the authors to clarify in the text, without the need of further revision on my side.

We thank the reviewer for the evaluation of our manuscript. As indicated below, we made the requested textual modifications.

In line 106, the authors state, “This correlated with faithful maintenance of DNA methylation levels at the domain’s DMRs (Fig. 1f).” I do not understand what exactly “faithful maintenance” means. The same phrase is used elsewhere (e.g., line 113). Please clarify this phrase, or consider deleting it.

We changed this sentence as follows: “*This correlated with maintenance of normal DNA methylation levels at the domain’s DMRs (Fig. 1f)*”. In the two other instances, we now write ‘maintenance’ instead of ‘faithful maintenance’

The authors generated a new deletion mutant ESC line in which the Meg lncRNA production is abolished without affecting its transcriptional initiation or the production of other parts of the polycistron. Although the results of this new experiment are informative, the mutant carries a large deletion and therefore does not fully resolve my concern that DNA sequence manipulation could have unintended effects on replication timing. I still believe that the drug-mediated inhibition of transcription would be worth trying, as it represents the simplest way to exclude this possibility (if complete transcriptional inhibition is toxic to cells, the authors might consider testing partial inhibition to examine its impact on replication timing). If this experiment is not feasible, the authors should provide a discussion of the possible unintended effects on replication timing.

In the Discussion, on page 18, we added a paragraph to mention that the different deletions that induced loss of the Meg3 lncRNA expression may have generated unintended side effects:

“One limitation of exploring deletions that ablate Meg3 lncRNA expression (Δ Meg3-C1 and Zfp57^{-/-};Meg3-pro^{-/-}) is that these may have comprised regulatory sequences that influence RT. This could be relevant particularly for the Δ Meg3-C1 in which we deleted a large region, from exon 2 until exon 10 of Meg3. However, since these two independent, non-overlapping deletions both led to delayed replication timing within the imprinted domain, the most parsimonious model is that the lncRNA expression contributes to the early replication on the maternal chromosome (Fig. 7).”

In our revision, we did not attempt to determine whether locus-specific RT profiles become altered as a consequence of inhibition of transcription. As mentioned in our previous rebuttal, RNA-PolIII inhibition studies do not allow the distinction between direct (transcription at the locus) or indirect (altered expression of factors that influence replication or Meg3 transcription) effects. Moreover, as explained, it is also complicated to achieve efficient and timely inhibition within the nuclei of all cells, which may be particularly challenging for the Meg3 lncRNA, which is known for its

DNA-association and complex higher-order folding due to extensive hairpin formation (Sanli et al. *Cell reports* 2018; Sherpa et al. *Nucl. Acids Res* 2018; Uroda et al. *Mol Cell* 2019; Farhadova et al. *Nucl. Acids Res* 2024).

Generally, the relationship between transcriptional activity and replication timing is complex and not necessarily causal. While a general correlation exists between early replication and transcriptional activity across metazoans (Gilbert, 1986; Goldman et al., 1984; Hatton et al., 1988; Vouzas & Gilbert, 2021), this association primarily reflects the replication of constitutively expressed housekeeping genes in all cell types (Rivera-Mulia et al., 2015). Developmentally regulated genes display a much weaker correlation between RT and transcription. Furthermore, certain monoallelically expressed and asynchronously replicating genes demonstrate no evident correlation between their transcriptional state and RT (Heskett et al., 2022). These observations suggest that transcription and RT are coordinated but can be uncoupled and that RT regulation likely involves higher-order chromatin organization rather than transcriptional output alone.

Studies that have manipulated transcription experimentally to test its effect on RT have produced variable and sometimes contradictory results. For instance, targeting strong transcriptional activators to the promoters of late-replicating genes in mouse embryonic stem cells resulted in transcriptional activation and accelerated RT (Brueckner et al., 2020; Therizols et al., 2014). In contrast, other studies have shown that histone acetylation alone, without any detectable transcriptional induction, can also advance RT (Goren et al., 2008). Furthermore, the degree and type of transcription appear to matter. In chicken DT-40 cells, the insertion of active promoters or high levels of transcription of long transcripts advanced RT; short or low-level transcription did not (Blin et al., 2019; Hassan-Zadeh et al., 2012; Brossas et al., 2020). These heterogeneous outcomes suggest that the relationship between transcription and RT is context-dependent and influenced by factors such as chromatin environment, transcript length, and transcriptional strength.

Using global RNA polymerase II inhibitors, such as α -amanitin, DRB, or triptolide, would profoundly alter the transcriptional landscape and chromatin architecture genome-wide. These treatments can alter gene expression, chromatin accessibility, nuclear organization, and replication origin activity. Consequently, any observed changes in RT under these conditions would be difficult to interpret because they could result from indirect or pleiotropic effects rather than a direct influence of transcription on the replication program of the locus under study. Thus, while drug inhibition could reveal general dependencies between transcription and replication, it would not provide mechanistic insight into the local relationship between transcriptional activity and RT at a specific genomic region.

We hope that the reviewer agrees that our added comment about the potential removal of regulatory elements provides sufficient caution for the reader.

In line 287, the authors state, "At the Dlk1-Dio3 domain, asynchronous replication thus associates with distinct allelic A/B compartments." I think this is an overstatement, as I do not see distinct differences in A/B compartments at this

domain between androgenetic and pathenogenetic mESCs. While I do see some differences in this domain, a similar degree of differences is also observed at the 108-109 Mb region, where no asynchronous replication occurs.

We agree with the reviewer that this observation should have been presented more carefully. In the text, we tuned-down this sentence as follows: “At the Dlk1-Dio3 domain, asynchronous replication thus associates with minor differences in A/B compartment organization”.

Reviewer #2 (Remarks to the Author):

I am very happy with the author's response to my comments. This is an exciting advance in the imprinting field

We thank the reviewer for the constructive evaluation of our manuscript and for sharing his/her excitement about this work.

Reviewer #3 (Remarks to the Author):

With this revision, Imaizumi and colleagues have improved the overall interpretation and presentation of their results and more precisely present their findings. Reflecting the changes the authors made to their title, this manuscript presents an in-depth study of how differential DNA methylation and/or lncRNA expression impacts asynchronous replication timing at the imprinted Dlk1-Dio3 locus. As said before, the major advance of this study over others are the use of genetic manipulation models to examine RT asynchrony via Capture Repli-seq, particularly for investigating the role of DNA methylation. However, these models and the additional CRISPR deletion models fall short in definitively showing that lncRNA expression independent of DNA methylation has a role in asynchronous replication at the Dlk1-Dio3 locus. Inclusion of dynamic intervention models for DNA methylation and/or expression would have greatly increased the novelty and mechanistic significance of this study. Major comments below.

We thank the reviewer for the constructive evaluation, including the conclusion that we improved the interpretation and presentation of our results.

1. Reviewer comment 3a: “...Deleting the promoter region of Meg3 could be unintentionally affecting TF/CTCF sites that directly affecting RT and 3D organisation regardless of expression changes (Chakraborty et al. 2025). The authors thus cannot conclude that any consequences are due to only expression loss.”

- Authors' response to point 3a: “On a first note, we previously characterized CTCF binding and its importance for 3D structure at this domain, describing how CTCF

does not bind the promoter of *Meg3* but rather allele-specifically at its first intron (Llères et al, 2019).”

>> It is not only CTCF, but other transcription factor binding sites may be deleted in the *Meg3*-pro^{-/-} model. For example, Sima et. al. 2019 found that deletion of cis-acting ERCE elements affected RT, but depletion of CTCF protein had no effect on RT, implying that something other than CTCF binding at these ERCEs is controlling RT.

We thank the reviewer for providing additional detail. When reading the initial remark, we were not sure to which manuscript by Chakraborty et al., the reviewer was referring. The study from the Rocha group, which reports the deletion of a CTCF site (<https://pubmed.ncbi.nlm.nih.gov/40015278/>), does not include data on replication timing. In an effort to reply to the reviewer, we therefore referred to our own work on CTCF binding at the domain instead.

We also thank the reviewer for pointing out the Sima *et al.* 2019 study, and for highlighting its finding that CTCF binding at ERCEs may not be relevant for controlling RT. Indeed, we had reported on the importance of CTCF binding to intron-1 of *Meg3* for chromatin organisation and gene expression at the locus (Llères et al., 2019). However, the deletion of the *Meg3* gene in the *Meg3*-C1 mESCs does not comprise this intronic region (we deleted exon 2 until exon 10), and neither does the promoter deletion. In the manuscript we therefore chose to not draw conclusions about how CTCF binding might influence RT. We do agree that these deletions may have ablated other key TF binding sites that could for instance be involved in enhancer functions. See the continuation of this discussion below the next comment from the reviewer.

- Authors' response to point 3a: “To address if, in the *Zfp57*^{-/-};*Meg3*-pro^{-/-} mESCs, the observed consequences for replication timing were due solely to the expression loss, we generated a new recombinant mESC line in which we removed the *Meg3* lncRNA gene from the 220-kb ncRNA polycistron of the imprinted *Dlk1*-*Dio3* domain, while keeping DNA methylation, CTCF binding at the first intron, and gene expression across the remainder of the polycistron unaltered. In this new mESC line, in which the lncRNA *Meg3* was no longer expressed (shown in new figure 5), we find delayed replication timing across (non-overlapping) parts of the imprinted domain on the maternal chromosome. On the maternal allele, RT in this new cellular model mirrored the one in *Zfp57*^{-/-};*Meg3*-pro^{-/-} mESCs. This key finding excludes the possibility that the effect seen in the *Zfp57*^{-/-};*Meg3*-pro^{-/-} mESCs solely resulted from the deletion of (RT-regulatory elements within) the *Meg3*-promoter. Instead, it consolidates the suggested involvement of the lncRNA *Meg3* (rather than that of the expression of the entire ncRNA polycistron). (see also our answer to point 5 of Reviewer 1).”

>> It was the concern of both Reviewer 1 (point 5) and this reviewer, that genetic disruption could confound the authors' conclusion that it is expression loss that affects allelic RT. The ideal solution to this would have been to use a method that only disrupts the transcriptome and not the genome, as suggested by Reviewer 1 (also see comment below). If anything, an even larger genetic disruption, as in the

Meg3-C1 deletion, would amplify the issue. For example, can the authors be sure that the Meg3-C1 deletion is not removing other enhancer-like elements that contribute to the organisation of the locus? Either with or without CTCF?

We thank this reviewer for raising this important issue, which we now address in the Discussion (see below). Indeed, we can't exclude that the Meg3-C1 deletion (exon 2 until exon 10) does not remove enhancer-like elements that may influence the organisation of the locus. To provide the reviewer with additional insights, we compiled the below reviewer figure, which shows published ATAC-seq data for the *Meg3* locus in E14 mESCs, combined with our own (unpublished) ChIP-seq data on H3K27ac (a histone modification that is enriched at enhancer elements) in both hybrid mESC lines (BJ and JB) that were used in this study. Indeed, we observe a single peak of accessible chromatin in the deleted region, in intron-8 of *Meg3*, whose deletion in the Δ *Meg3*-C1 cells could potentially have influenced RT. However, this site of chromatin accessibility does not show H3K27ac enrichment and is therefore unlikely to function as an enhancer (see provided figure below).

Importantly, the early replication timing on the maternal chromosome was also affected by the small, non-overlapping promoter deletion (see yellow highlight in the provided reviewer figure). Since two independent deletions affecting the expression of the lncRNA polycistron both led to delayed replication timing at proximal and distal regions, the most parsimonious model is that the lncRNA expression contributes to the early replication at the *Dlk1-Dio3* domain on the maternal chromosome. As discussed, it remains to be determined how precisely this is regulated and which other factors could be involved.

To nonetheless discuss this potential limitation of our study, as raised by reviewer 1 as well, we have added the following paragraph to the Discussion, on page 18:

“One limitation of exploring deletions that ablate Meg3 lncRNA expression (dMeg3-C1 and Zfp57^{-/-};Meg3-pro^{-/-}) is that these may have comprised key regulatory sequences that influence RT. This could be relevant particularly for the Meg3-C1 in which we deleted a large region, from exon 2 until exon 10 of Meg3. However, since these two independent, non-overlapping deletions both led to delayed replication timing within the imprinted domain, the most parsimonious model is that lncRNA expression contributes to the early replication on the maternal chromosome.”

Reviewer Figure 1:

H3K27ac ChIP-seq in BJ and JB mESCs used in this study, and reanalyzed ATAC-seq in E14 mESCs (origin: <https://pubmed.ncbi.nlm.nih.gov/32690000/>). The gRNAs used for Meg3-promoter deletion and for generating the Delta-C1 line are positioned above. The yellow highlights indicate the span of the two different deletions.

Comparing maternal RT in Fig. 4h to Fig. 5e, the Meg3-C1 deletion does not appear to exactly mirror *Zfp57*^{-/-};*Meg3*-*pro*^{-/-} mESCs. Specifically, the region between 109.75 and 110 Mb is early in *Zfp57*^{-/-};*Meg3*-*pro*^{-/-} mESCs and Mid in Meg3-C1 deletion cell line. The authors point this out themselves in the figures. It is therefore possible that different genetic elements may be affected between the two deletions, leading to different results. In addition, the fact that the BJ and JB lines show differences in paternal allele RT, which the authors attribute to genetic differences between the strains (Line 147), further supports that genetic differences and thus also genetic disruptions can affect RT.

We thank the reviewer for commenting on RT differences in the Δ *Meg3*-C1 deletion line versus the *Zfp57*^{-/-};*Meg3*-*pro*^{-/-} mESCs. This is an important point, and the mentioned differences may hint at additional factors that influence the RT. However, it is complicated to directly compare these two lines. In the *Zfp57*^{-/-};*Meg3*-*pro*^{-/-} cells, there is loss of RNA expression across the entire 220-kb polycistron and, in addition, there is partial DNA methylation at the Meg3 DMR (due to a partial regain of biallelic methylation). In the *Meg3*-C1 cells, the methylation status of the DMRs is as in the WT cells, and only the expression of *Meg3* lncRNA is lost, while the remainder of the polycistron remains expressed normally. In addition, as pointed out by the reviewer, another difference between these two cell lines is that they are of reciprocal genotype (JB versus BJ), which may have influenced results, particularly in the distal part of the domain (see Figure 2e,f)..

We apologise, but it was precisely for these confounding reasons that we did not comment on differences between the *Meg3-C1* and *Zfp57^{-/-};Meg3-pro^{-/-}* mESCs. Instead, we compared the *Meg3-C1* cells with the WT mESCs from which they derived (BJ genotype), so that we could conclude specifically about the importance of Meg lncRNA expression only (Figure 5). The *Zfp57^{-/-};Meg3-pro^{-/-}* cells were compared with the *Zfp57^{-/-}* cells (Figure 4) from which they derived (JB genotype), so that we could conclude specifically about the induced loss of polycistron expression. As the reviewer, we noted a stronger effect of the Meg3 deletion in the distal portion of the domain (downstream of the polycistron, Fig. 5e), compared to that of loss of expression of the entire polycistron (Fig. 4h). The available data do not allow us to draw conclusions about this difference, but we do not exclude the possibility that genetic differences between the two lines (BJ versus JB genotype) may have influenced the results.

To be transparent about this limitation, we have added a sentence to the Discussion (page 18) to emphasize the likely contribution of genotype differences when comparing different cell lines that are of reciprocal genotype (BJ versus JB):

“Another limitation is that the different deletion lines in our study were not all of the same hybrid genotype. As highlighted by the minor differences observed between the BJ and JB mESCs (Figure 2e,f), also this may have influenced the observed effects on replication timing.”

>> A cleaner solution could be to use siRNAs targeted to the Meg3 lncRNA. siRNAs would only disrupt RNA in a targeted manner, and not DNA. This method would deplete expression of all 3, Meg3-Rian-Mirg, as they are one transcript, so one would not be able to delineate between the 3, but differentiating between the 3 is not the focus of this manuscript anyway. Additionally, by removing just the RNA this would address the hypothesis the authors raise in the discussion, that the lncRNA itself could be contributing to the 3D organisation and sequestering replication factors to ensure early replication.

i. As a note, using siRNAs would not be too long a period to detect replication timing change in mESCs (siRNAs are often used for a max 7 days in rapidly dividing cells). The authors wrote the following in response to comment 5 from reviewer 1: “ α -Amanitin treatments of cells take >24h to achieve full transcriptional repression...which would be too long for RT studies on the rapidly dividing mESCs”. A length of time as short as 24hrs or even up to 7 days should not be a concern to the authors as the authors main cell models, Sh-1 and *Zfp57^{-/-}*, took days to weeks to generate, as they required colony selection. Similarly, the CRISPR models the authors generate also took more than 12 days according to their methods, “After 10–12 days of culture, colonies were transferred to 6-well plates to derive clonal mESC lines.”

In our revision, we did not attempt to determine whether the locus-specific RT profile at *Dlk1-Dio3* could become altered as a consequence of inhibition of transcription with α -Amanitin or similar other agents. In such RNA-PolIII inhibition studies it is difficult to conclude whether observed locus-specific effects are a direct consequence of altered transcription at the locus of interest, or an indirect effect of

altered expression of factors that influence replication or *Meg3* transcription. Practically, as commented on before, it is also complicated to achieve efficient transcriptional inhibition of a nuclear lncRNA, in all cells and in a timely manner. Please, see also our reply to reviewer 1.

Based on our earlier studies, and from research by others, it is likely that the lncRNA *Meg3* is highly structured and chromatin associated (Sanli et al. *Cell reports* 2018; Sherpa et al. *Nucl. Acids Res* 2018; Uroda et al. *Mol Cell* 2019; Farhadova et al. *Nucl. Acids Res* 2024). Instead of the challenging and nuclear siRNA experiments with highly uncertain outcomes, we therefore opted to generate the additional genetic model in which we specifically targeted the *Meg3* gene. Although we agree that the idea of using siRNAs is interesting, we hope that the reviewer agrees that the more detailed description of limitations due to potential perturbations of regulatory elements is sufficient.

3. Author response to point 5a: “As detailed in our response to reviewer 1 (point 7), we instead performed genome-wide Hi-C on parthenogenetic and androgenetic mESCs and determined the A/B compartment. Please, see our answer to reviewer 1 (point 7) for interpretation. These new data are presented in Figure 6a.”

>> It's nice to see that the RT boundaries do seem to match A/B compartment boundaries, if not TAD boundaries. It is also interesting that RT and Hi-C differ for the parthenogenic line i.e. in the parthenogenic mESCs, the locus is clearly early replicating, but only very slightly A-compartment in the Hi-C. This suggests that this locus in general is not distinctly compartmentalized i.e. not 'locked in' to A or B, and this could be a reason why it can change RT readily between maternal and parental chromosomes, and upon genetic manipulation. It could be interesting to see if other imprinted loci are similar, with or without RT asynchrony.

We thank the reviewer for this interesting suggestion. Below, we made a figure for the reviewer that shows the A/B compartment [i.e., Eigenvector values, commonly used to identify A (positive sign) and B compartments (negative sign)] at the main imprinted domains in the parthenogenetic versus the androgenetic mESCs, based on our low-resolution Hi-C data.

Reviewer Figure 2:

Replication timing (RT) and Hi-C eigenvector comparison in parthenogenetic and androgenetic mESCs, across 13 well-characterized imprinted domains. The genes are indicated underneath. The grey highlights indicate genes annotated as imprinted (in one tissue at least).

We were intrigued to see that there is a minor, but inconsistent, trend where small occurrences in RT asynchrony at several other imprinted domain coincide with minor differences between A/B compartments (*Peg10*, *Peg3-Zim2*, *Snrpn-Ube3a*, *Igf2-H19*

Kcnq1, Grb10). This observation may thus be considered as an orthogonal validation for the existence of RT asynchrony at imprinted domains. However, given that the differences are minor, are not consistent at all loci, and more generally that the focus of our manuscript is on RT regulation at the *Dlk1-Dio3* domain, we think this analysis is outside of the scope of the manuscript.

>> It would be great to see a more zoomed in view of Fig. 6b. Does the A/B-compartment boundaries in the parthenogenic and androgenic lines help in any way to interpret the genomic location differences in RT changes between disrupting DNA methylation and disrupting the Meg3 lncRNA?

We thank the reviewer for this helpful suggestion. We have added a new track in Figure 6c (below the insulation score) that shows the zoom of the A/B compartment eigenvector scores at the Dlk1-Dio3 domain. Despite their lower resolution, the largest difference in Eigenvector between the parental chromosomes is observed where RT assynrony is most pronounced as well.

We prefer not to use these data from the WT monoparental mESCs for the interpretation of RT domain differences in cells with DNA methylation or Meg3 lncRNA perturbations, as each of these lines carries their particular changes, including changes in allelism of CTCF binding.

>> It is unfortunate that further analysis of RT and TADs from the capture Hi-C did not bring more clarity on this phenomenon. But have the authors looked at the genome wide Hi-C TADs? Do the TADs from genome wide Hi-C done in the parthenogenic and androgenic lines make sense with the RT boundaries? One benefit of looking at the genome-wide HiC is perhaps the capture Hi-C region missed regions of the genome where larger TAD structures that may interact with the *Dlk1-dio3* locus.

With all due respect for the reviewer, we are unfortunately not sure what he/she means by “larger TAD structures that may interact with the *Dlk1-Dio3* locus”. TADs are defined as insulated domains with depleted contacts with the outside. The larger structures that do interact over larger distances, albeit at relatively low frequency and probably in a heterogeneous manner within the cell population, are the A/B compartments, which are characterized by preferred mutually exclusive homotypic interactions among either euchromatic/gene-rich/non-lamina contacting regions or heterochromatic/gene-poor/lamina contacting domains.

As Figure 6b shows for a 5 Mb window, the Hi-C Eigenvector “drops” considerably on both parental chromosomes (as inferred from our Hi-C data in mono-parental lines) over the *Dlk1-Dio3* domain, with a more pronounced effect on the paternal allele. This is quite different from RT, where late replication is only observed on the paternal allele. In contrast, on the maternal allele a quite rare configuration is present, with early RT signal coinciding with Hi-C Eigenvector signal that hovers close to 0. This later may be due to both the low gene density and their mostly inactive state, with the exception of the actively transcribed Meg3 polycistron. We don't know what causes this disconnection between RT and A/B-compartments, but it's attractive to

speculate that it's exactly the Meg3 lncRNA and its retention at the locus that may uncouple them.

We consider the discussion about how genome-wide RT relates to genome-wide TADs and/or A/B compartments out of the scope of this study, and therefore prefer not to discuss this in the manuscript.

4. Authors' response: "In our mESC cultures, we systematically add ascorbic acid to the medium to prevent acquisition of abnormal DNA methylation at the IG-DMR differentially methylated region of the Dlk1-Dio3 domain. Since our cells were derived in the presence of multiple inhibitors and have been grown in the presence of ascorbic acid, they are considered naïve. To prevent aberrant methylation at the Dlk1-Dio3 domain we have to put the ascorbic acid, and longer incubations with ascorbic acid are not expected to lead to reductions in DNA methylation under these conditions....Once again, we already have genetic systems that alter the DNA methylation at the imprinted DMRs, which in our view is a better approach."

>> The suggestion was specifically to remove ascorbic acid, rather than add more. This was what was asked: "One possibility may be to assay allelic replication timing after removal of ascorbic acid from the culture media. The effect of ascorbic acid on DNA methylation in mESCs is reversible, whereby 5mC gradually returns over 9 days (Blaschke et al. 2013)." It would follow that the maternal allele could gain methylation at the IG-DMR over a course of 9 days. Using this as a timecourse would validate that DNA methylation at this DMR as an effect on allelic RT, using a transient and non-genome disruptive method. In other words, how would this the acquisition of abnormal DNA methylation upon removal of ascorbic acid impact the maternal locus?

It has already been shown that ascorbic acid can affect the methylation of the Dlk1-Dio3 locus (Stadtfeld et. al. 2012). The benefit of this is that unlike the Sh-1 cell line, the cells do not need to undergo clonal selection. This contrasts with the genetic models where cause and consequence are separated for much longer periods of time. Often weeks were required to generate the genetic models used in this study. This dynamic approach would better support the mechanism between DNA methylation and RT, as the cause and consequence can be followed at much closer timepoints and in sequence. There may be other reasons that removing ascorbic acid may not work, but the main point is that a transient DNA methylation disruption system would more clearly show a direct mechanistic relationship between allelic DNA methylation and allelic RT.

This is of lower priority than the lncRNA comment above.

We thank the reviewer for the further thoughts on ascorbic acid. As observed by Stadtfeld and his colleagues, and other groups in the field, the DMRs of the *Dlk1-Dio3* domain have a tendency to gain aberrant *de novo* DNA methylation on their unmethylated maternal allele upon prolonged culture of mESCs. This occurs in a stochastic and non-timed manner. It is precisely to prevent this aberrant situation that the addition of ascorbic acid to the culture medium is important.

As opposed to the genome as a whole, imprinted DMRs are exceptional in that their DNA methylation status is not reversible, with the exception of the active setting of imprinted status in the germ cells. This remarkable stability allows these regions to stably maintain their allelic methylation status throughout development -including through the pre-implantation waves of genome-wide demethylation and remethylation- and this mediates the allelic expression of the imprinted genes. If on the other hand, the imprinted status is lost, as for instance is the case for imprinting disorders associated with the human *DLK1-DIO3* domain, this is stably maintained as well. Similarly, in an in-vitro cell culture context, the removal of ascorbic acid, followed by its re-addition, will therefore not only have unpredictable effects, but will also not result in a return to allele-specific differences afterwards, and certainly not in a parent-of-origin-dependent manner.